# High-latitude biomes and rock weathering mediate climate–carbon cycle feedbacks on eccentricity timescales

David De Vleeschouwer [1✉], Anna Joy Drury [1,2], Maximilian Vahlenkamp [1], Fiona Rochholz [1,3], Diederik Liebrand[1] & Heiko Pälike [1]

The International Ocean Discovery Programme (IODP) and its predecessors generated a treasure trove of Cenozoic climate and carbon cycle dynamics. Yet, it remains unclear how climate and carbon cycle interacted under changing geologic boundary conditions. Here, we present the carbon isotope ($\delta^{13}C$) megasplice, documenting deep-ocean $\delta^{13}C$ evolution since 35 million years ago (Ma). We juxtapose the $\delta^{13}C$ megasplice with its $\delta^{18}O$ counterpart and determine their phase-difference on ~100-kyr eccentricity timescales. This analysis reveals that 2.4-Myr eccentricity cycles modulate the $\delta^{13}C$-$\delta^{18}O$ phase relationship throughout the Oligo-Miocene (34-6 Ma), potentially through changes in continental weathering. At 6 Ma, a striking switch from in-phase to anti-phase behaviour occurs, signalling a reorganization of the climate-carbon cycle system. We hypothesize that this transition is consistent with Arctic cooling: Prior to 6 Ma, low-latitude continental carbon reservoirs expanded during astronomically-forced cool spells. After 6 Ma, however, continental carbon reservoirs contract rather than expand during cold periods due to competing effects between Arctic biomes (ice, tundra, taiga). We conclude that, on geologic timescales, System Earth experienced state-dependent modes of climate–carbon cycle interaction.

[1] MARUM - Center for Marine and Environmental Sciences, University of Bremen, Klagenfurterstraße 2-4, 28359 Bremen, Germany. [2] Present address: Department of Earth Sciences, University College London, Gower Street, London WC1E 6BT, UK. [3] Present address: Research Group for Earth Observation, Pädagogische Hochschule Heidelberg, Czernyring 22/10-12, 69120 Heidelberg, Germany. ✉email: ddevleeschouwer@marum.de

The $\delta^{13}C_{benthic}$ of deep-sea benthic foraminifera reflects the dissolved inorganic carbon composition ($\delta^{13}C_{DIC}$) of the surrounding waters during calcification. High-resolution $\delta^{13}C_{benthic}$ time-series from deep-sea sedimentary archives thus reveal how deep-sea $\delta^{13}C_{DIC}$ varied through geologic time[1,2]. This is important, as past $\delta^{13}C_{DIC}$ changes can illuminate rearrangements between other carbon reservoirs that interact with the deep-ocean reservoir. However, the $\delta^{13}C_{benthic}$ signal represents a composite of several processes, complicating the disentanglement of past changes in the partitioning of carbon between ocean, atmosphere and terrestrial biosphere. For example, an enigmatic negative $\delta^{13}C$ excursion at 56 Ma (Palaeocene–Eocene thermal maximum) represents a rapid carbon transfer from an isotopically negative reservoir to the atmosphere–ocean system on $10^4$-year timescales[3]. On longer ($10^4$–$10^7$ years) timescales, the burial of organic carbon in terrestrial environments (e.g. as coal) causes marine $\delta^{13}C_{DIC}$ to become more positive[4]. Conversely, when the terrestrial biomass shrinks, marine $\delta^{13}C_{DIC}$ turns more negative as the marine $^{12}C$ reservoir grows relative to the continental one (net $^{12}C$ flux from continents to oceans). Sediments and lithosphere also exhibit carbon exchange with the oceans: organic matter accumulation is an important $^{12}C$ sink that leads to more positive marine $\delta^{13}C_{DIC}$, whereas volcanic outgassing provides slightly depleted carbon to the ocean–atmosphere system. Changes in ocean-based photosynthesis or respiration also result in changing deep-ocean $\delta^{13}C_{DIC}$: A stronger biological pump, with organic matter respiration at depth, causes deep-sea $\delta^{13}C_{DIC}$ to turn more negative and induces a steepened surface-to-deep carbon isotopic gradient ($\Delta\delta^{13}C_{DIC}$). Ocean overturning and mixing further complicate the deep-ocean $\delta^{13}C_{DIC}$ pattern.

The oxygen isotope composition of benthic foraminifera, $\delta^{18}O_{benthic}$, is a well-established proxy for the evolution of deep-sea temperatures and the extent of the continental cryosphere. When combined, $\delta^{13}C_{benthic}$ and $\delta^{18}O_{benthic}$ provide insights in climate–carbon cycle interactions through geologic history. Indeed, climate and carbon cycle are profoundly coupled systems, as many carbon-cycle processes are affected either directly by changes in temperature (e.g. soil respiration, net primary productivity, $CO_2$ solubility) or by variables that covary with temperature (e.g. precipitation, weathering). Thereby, the carbon cycle controls the amount of effective greenhouse gases ($CO_2$ and $CH_4$) in the atmosphere, regulates Earth's global temperature, and hence closes the climate–carbon cycle feedback loop. However, to date, the evolution and pivotal points of carbon cycle–climate feedback mechanisms throughout the Cenozoic Icehouse are poorly understood. This is chiefly due to a lack of continuous high-resolution benthic isotope records that span the last 35 Myr of Earth's history.

We introduce a 35-Myr-long $\delta^{13}C_{benthic}$ megasplice of nine orbitally resolved $\delta^{13}C_{benthic}$ records (2.8-kyr average time resolution). In contrast to existing compilations that gather and/or average time-equivalent data across sites[1,2], the megasplice solely comprises data from a single site at any point in time. The $\delta^{13}C_{benthic}$ megasplice provides a time-continuous geological history of carbon-cycle dynamics over the last 35 Myr (Fig. 1c) by combining records from different ocean basins. Its construction is similar to the construction of its $\delta^{18}O_{benthic}$ counterpart[5], where million-year scale isotopic divergence between ocean basins is compensated by shifting individual records to the Pacific Ocean $\delta^{13}C_{benthic}$ long-term trend[2] (see "Methods" section). Two important tectonic ocean gateways reconfigurations are held responsible for the steepening Atlantic–Pacific $\delta^{13}C$ gradient since the middle Miocene[2]: deepening of sills surrounding the Nordic Seas (Fram Strait, FS on Fig. 2) and the shoaling of the Panama Isthmus (Central American Seaway, CAS on Fig. 2). The long-timescale adjustment between ocean basins makes a negligible contribution to the Milankovitch band (20–405 kyr), and hence, does not introduce short-term artefacts that might impact our astronomical cycle and phase interpretations. Moreover, the removal of smooth inter-basinal $\delta^{13}C_{benthic}$ trends from individual isotopic time-series reveals similar astronomical rhythms in $\delta^{13}C_{benthic}$ time-series from different basins (Supplementary Fig. 1). This highlights an important assumption for this work: Deep-sea $\delta^{13}C_{benthic}$ records from different basins pick up a strong global astronomical $\delta^{13}C_{benthic}$ rhythm, despite distinct basin-specific $\delta^{13}C_{DIC}$ values. Supplementary Fig. 1 corroborates this assumption for the Pleistocene, when maximum oceanic $\delta^{13}C_{DIC}$ heterogeneity occurs, as well as for the Tortonian (Late Miocene), when rapid inter-basin carbon isotopic divergence unfolds.

The connection between Earth's orbital eccentricity and the global carbon cycle is apparent from the $\delta^{13}C_{benthic}$ wavelet spectrogram, especially in response to the 405-kyr eccentricity cycle (Fig. 1b). This consistent link is remarkable because the $\delta^{18}O_{benthic}$ megasplice[5] loses its 405-kyr eccentricity imprint after the Miocene climatic optimum (16–14 Ma), indicating a fundamental divergence in global climate and carbon cycle response to astronomical forcing (Earth System response to astronomical forcing). Different authors reported that, after 14 Ma, the cryosphere ($\delta^{18}O_{benthic}$) became particularly sensitive to obliquity forcing[5,6], most likely in connection to Antarctic ice sheet expansion[7]. A strengthened obliquity imprint is also observed in $\delta^{13}C_{benthic}$ records[8]. However, at the rhythm of the 405-kyr eccentricity cycles, both proxies exhibit fundamentally different responses. The strong imprint of 405-kyr eccentricity cycles in $\delta^{13}C_{benthic}$ is likely related to the amplification of a non-linear carbon-cycle response to eccentricity cycles, facilitated by the ~65 kyr residence time of carbon in the oceans as a whole (and assuming zero net carbon exchange between the atmosphere and surface ocean)[8–12]. Yet, it remains an open question as to why the 405-kyr eccentricity imprint fades from $\delta^{18}O_{benthic}$ after 14 Ma, but not the 100-kyr eccentricity imprint. For more details on this so-called 405-kyr problem[13], we refer to de Boer et al.[8].

In this manuscript, the changing interactions between the climate–cryosphere system ($\delta^{18}O_{benthic}$), global carbon cycle ($\delta^{13}C_{benthic}$) and astronomical Milankovitch forcing is investigated through time-evolutive $\delta^{13}C$–$\delta^{18}O$ phase analysis on 100-kyr eccentricity time-scales (90–135 kyr frequency range). We use 1.2-Myr-wide sliding windows and take 0.2-Myr steps between computations to describe in-phase versus anti-phase behaviour, and to identify leads and lags between $\delta^{13}C$ and $\delta^{18}O$ (see "Methods" section). On 100-kyr eccentricity time-scales, $\delta^{13}C$ and $\delta^{18}O$ exhibit subtle oscillations around the zero-phase from 35 until ~6 Ma (Intervals I and II). After 6 Ma, the in-phase behaviour abruptly inverts to anti-phase behaviour (Interval III) (Figs. 2 and 3a).

## Results and discussion

**High-latitude biome dynamics as phase-switch trigger.** The abrupt shift from in-phase to anti-phase behaviour around 6 Ma in the Pacific and shortly after in the Atlantic Ocean is the most striking observation in Fig. 2, and has been comprehensively described by Kirtland Turner[14]. Her approach was to combine cross wavelet analysis with the average 100-kyr phase for individual $\delta^{13}C$ and $\delta^{18}O$ time-series, each a few million years long. Several of the ocean drilling sites used in that study also appear in the $\delta^{13}C$ and $\delta^{18}O$ megasplices because they constitute the best available orbitally resolved benthic isotope records for their specific time intervals. Contrary to the discrete *site-by-site* approach, we exploit the time-continuous nature of the megasplice by adopting a sliding window approach (see "Methods" section).

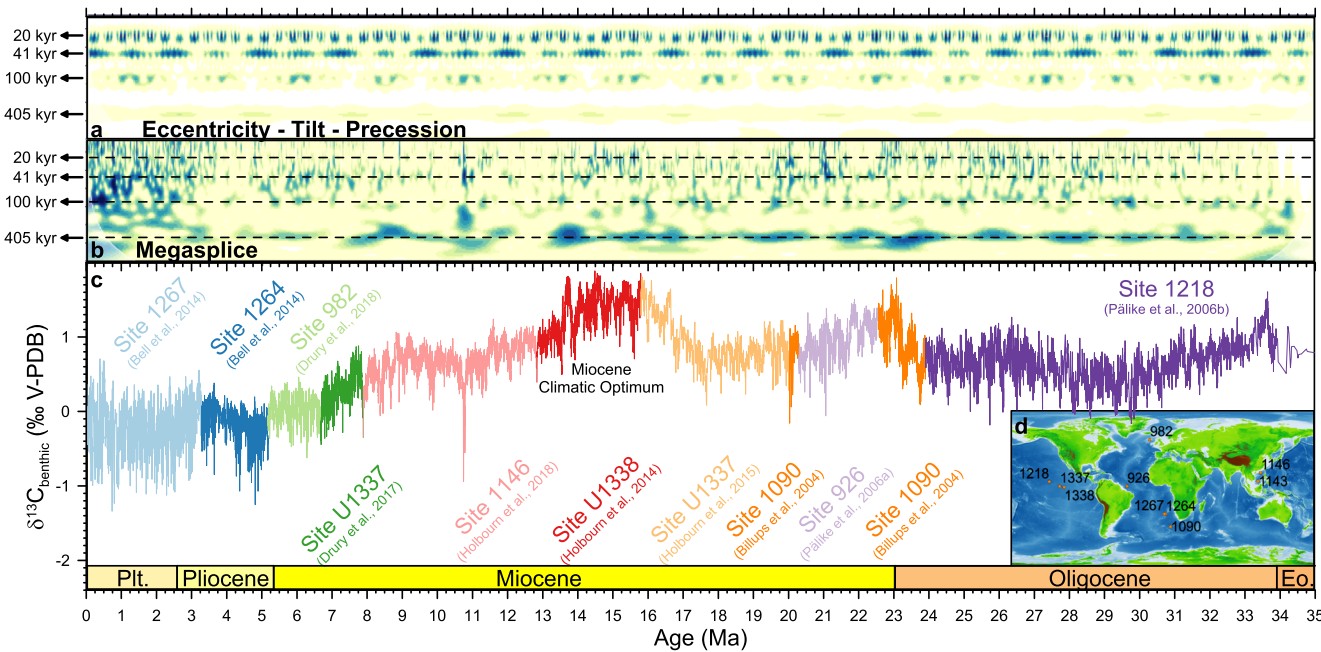

**Fig. 1 Benthic δ¹³C megasplice.** Wavelet spectrograms of **a** the astronomical rhythms (eccentricity–tilt–precession[58] composite) and **b** the benthic δ¹³C megasplice visualize the strong response of carbon cycle dynamics to 405-kyr long eccentricity. **c**, **d** The megasplice consists of nine globally distributed benthic carbon isotope records, all retrieved by the IODP and its predecessors: Site 926[75], 982[19,76], 1090[77], 1146[7,21], 1218[10], 1264[18], 1267[18], U1337[20,70] and U1338[78]. World map after Amante and Eakins[79].

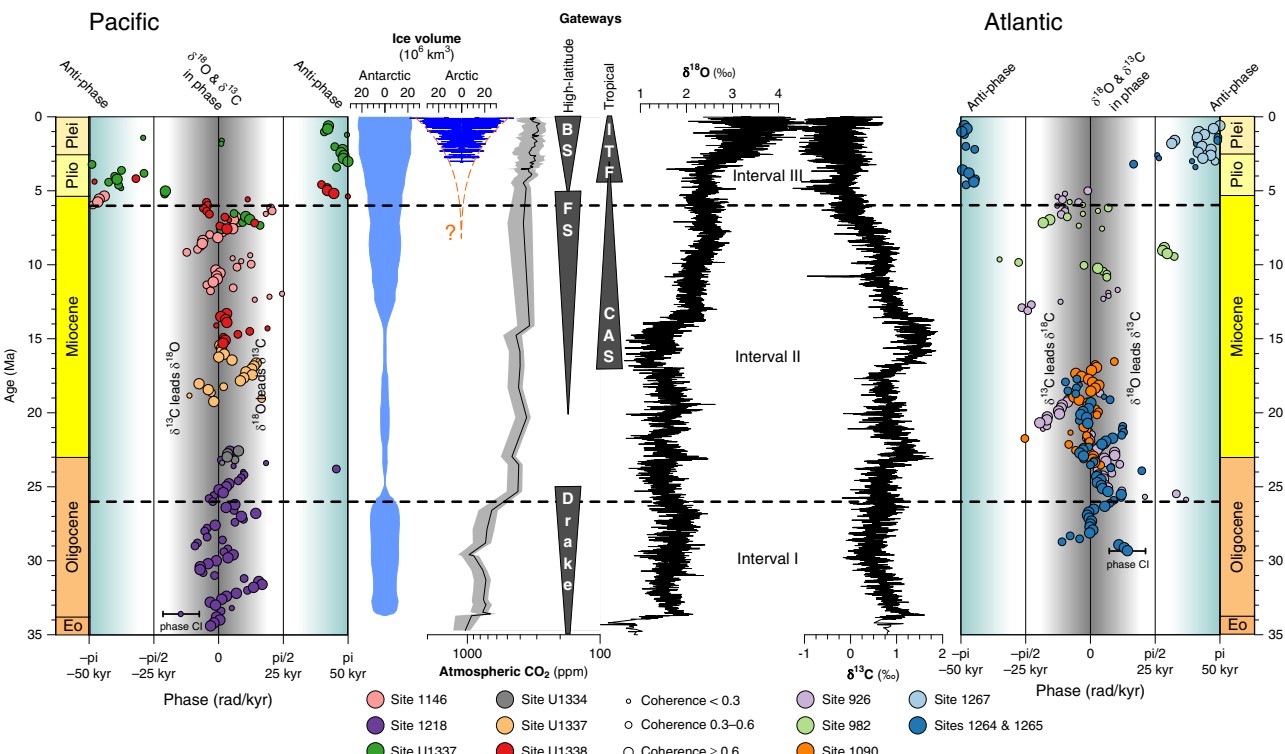

**Fig. 2 Phase-relationship between δ¹³C and δ¹⁸O on 100-kyr eccentricity time-scales.** On eccentricity time-scales, δ¹³C and δ¹⁸O exhibit cyclic in-phase behaviour from 35 till ~6 Ma in the Pacific, as well as in the Atlantic Ocean (Intervals I and II). Around 6 Ma in the Pacific (Interval III), and somewhat later in the Atlantic, we observe an abrupt shift from in-phase to anti-phase behaviour. The plotted uncertainty bar indicates a typical confidence interval (CI) on the phase estimate of ±0.44 rad. Uncertainty on the phase difference between δ¹³C and δ¹⁸O is inversely proportional to coherence and overall ranges between ±0.25 and ±0.59 rad. Antarctic and Arctic ice volume reconstructions are from Oerlemans[63] and Berger et al. [80], respectively. The earlier onset of NH glaciation suggested by this study is indicated in red. Atmospheric $CO_2$ reconstruction from Zhang et al. [81]. Grey triangles indicate opening and closing of important ocean gateways. BS, Bering Strait; FS, Fram Strait; Drake, Drake Passage; CAS, Central American Seaway; ITF, Indonesian Throughflow. All Sites that constitute the δ¹³C megasplice are included here (references in Fig. 1) and are complemented by additional orbital-resolution benthic isotope data from Sites 926[82], 982[83], 1264/1265[15,49], U1334[50,84], U1337[17] and U1338[6,85] (data as time-series in Supplementary Fig. 3).

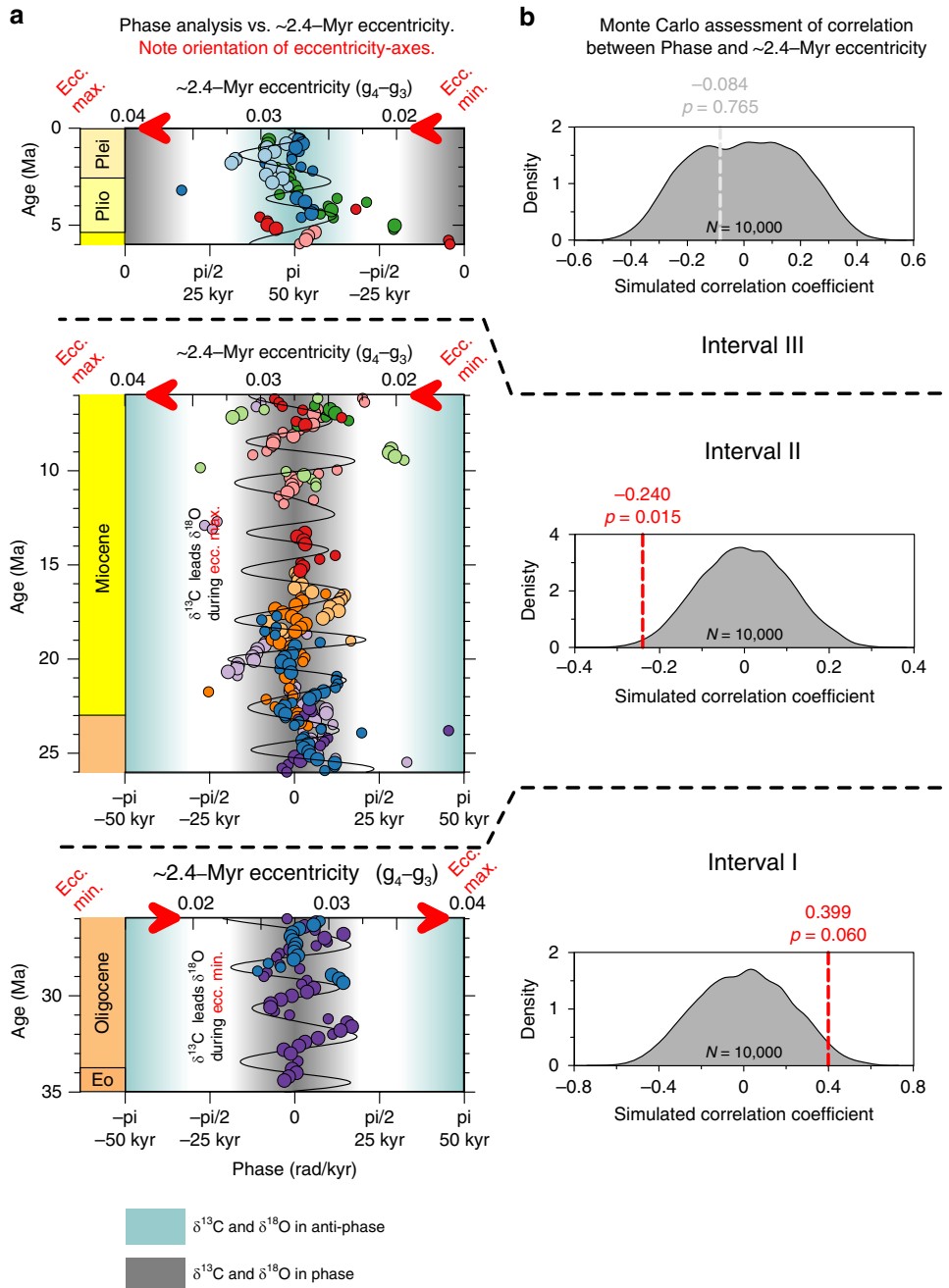

**Fig. 3 Long 2.4-Myr eccentricity ($g_4-g_3$) modulates Oligocene and Miocene leads and lags between $\delta^{13}$C and $\delta^{18}$O on 100-kyr eccentricity time-scales. a** Visual comparison of phase analysis results and 2.4-Myr eccentricity cycles, only considering phase-results with coherence >0.3 (see Fig. 2). **b** Prior to 26 Ma (Interval I), long 2.4-Myr eccentricity minima correlate with time-intervals when $\delta^{13}$C leads $\delta^{18}$O (positive Spearman correlation coefficient of 0.399 between phase and eccentricity). Between 26 and 6 Ma (Interval II), long 2.4-Myr eccentricity minima correlate with time intervals when $\delta^{18}$O leads $\delta^{13}$C (negative Spearman correlation coefficient of −0.240). After 6 Ma (Interval III), the correlation between 2.4-Myr eccentricity and phase ceases ($R = -0.084$).

Our approach allows for a narrower determination of the exact timing of the phase switch, and for the differentiation between ocean basins (Fig. 2), but is fundamentally congruent with earlier work[14]: Prior to the global phase switch, 100-kyr cycles in $\delta^{13}$C and $\delta^{18}$O were in-phase[15], with simultaneously occurring minima in both isotope systems (hyperthermal-style, sensu Kirtland Turner[14]). After 6 Ma, an anti-phase relationship arises on eccentricity timescales, with minima in $\delta^{13}$C corresponding to maxima in $\delta^{18}$O series[16] (glacial-style, sensu Kirtland Turner[14]). The anti-phase $\delta^{13}$C–$\delta^{18}$O behaviour in Interval III (<6 Ma) holds for all time-scales from obliquity[17–19] to the million-year

trends in $\delta^{13}$C and $\delta^{18}$O (Fig. 2). This is not the case prior to 6 Ma (Intervals I and II): then, 100-kyr cycles in $\delta^{13}$C and $\delta^{18}$O were in-phase, but long-term trends were occasionally out-of-phase. For example, in the multi-million-year run-up to the Miocene Climatic Optimum, $\delta^{18}O_{benthic}$ becomes more negative and $\delta^{13}C_{benthic}$ shifts towards more positive values (hence, anti-phased behaviour on million-year timescales between 20 and 15 Ma, Site U1337[20]).

Here, we present the in-phase hypothesis, providing a mechanistic interpretation for climate–carbon cycle dynamics during Intervals I and II (prior to 6 Ma). Our hypothesis starts

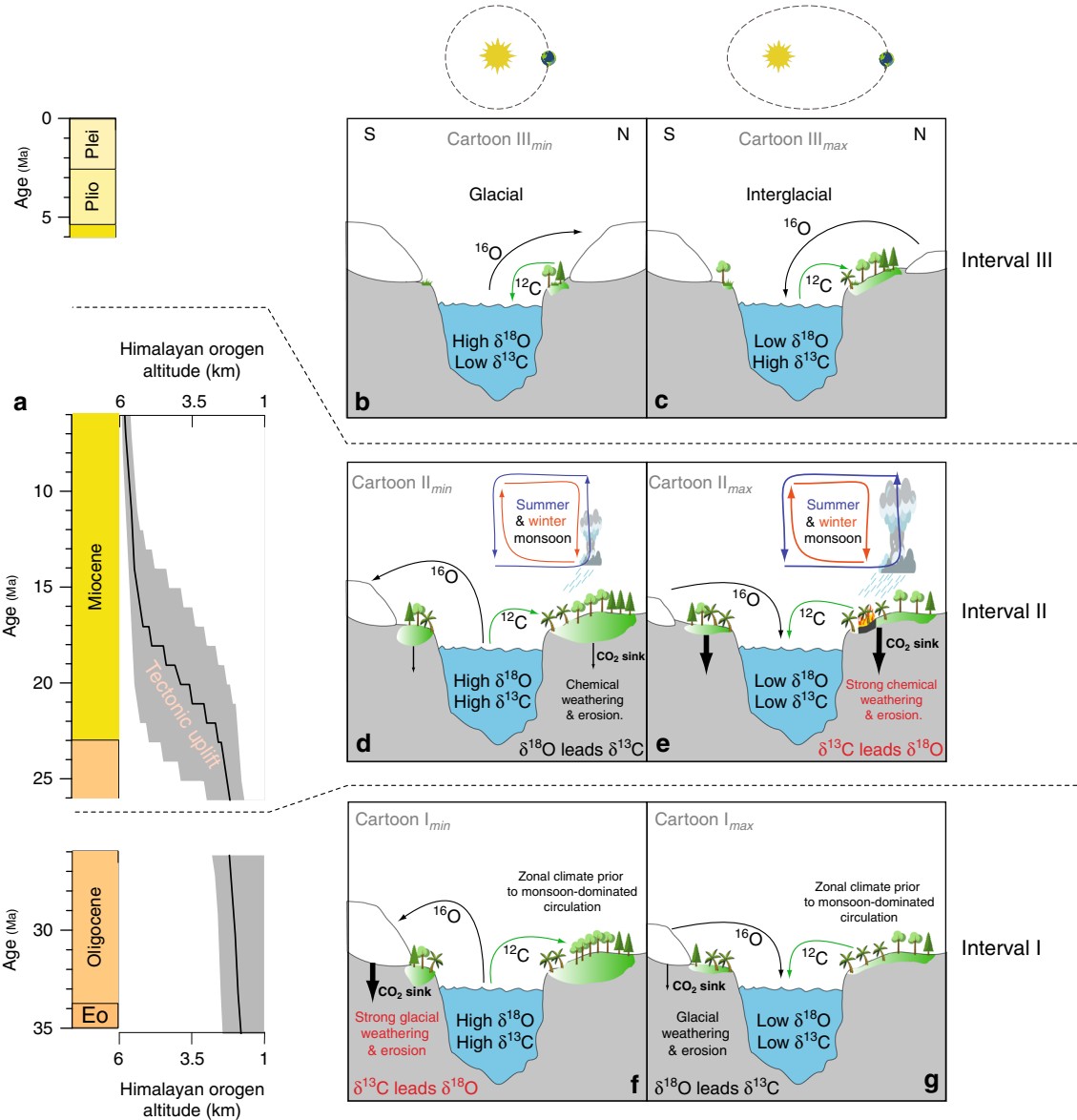

**Fig. 4 Cartoons illustrating the proposed concepts for the observed phasing patterns between δ$^{13}$C and δ$^{18}$O. a** Geologic time-scale of the last 35 Ma, and evolution of the Himalayan Orogen (after Ding et al.[64]). **b, c** The anti-phase behaviour in Interval III is primarily ascribed to high-latitude biome dynamics. **d–g** During Intervals I and II, when in-phase behaviour is observed, continental rock weathering forms the key mechanism in our hypotheses, with the carbon cycle (δ$^{13}$C) leading the climate–cryosphere system (δ$^{18}$O) when the continental rock weathering $CO_2$ sink is enhanced. This occurs during eccentricity minima (strong glacial activity) during Interval I, but during eccentricity maxima (strong monsoonal circulation) during Interval II.

from the established finding that eccentricity minima correspond to cooler climate spells of the relatively warm Miocene[5,21] (hence, δ$^{18}$O$_{benthic}$ ↑). These cooler periods coincide with climate conditions that remained relatively stable for tens of thousands of years, because precession-forced seasonal insolation extremes are truncated during eccentricity minima. Such prolonged stable climate conditions allowed for the expansion of continental carbon reservoirs, especially in the low latitudes, where precession amplitude directly controls insolation. For example, under eccentricity minima, deep subtropical soils[22] would be less affected by seasonal extremes (e.g. wildfires, hydrological stress by strong monsoons, Fig. 4d–g), and could store more isotopically light carbon ($^{12}$C). As a result of the more effective $^{12}$C storage on land, the average δ$^{13}$C composition of marine DIC would increase, which conforms with the relatively positive δ$^{13}$C$_{benthic}$

observed at times of low eccentricity during Intervals I and II (δ$^{13}$C$_{benthic}$ ↑, Fig. 4d–g). This hypothesis explains the in-phase δ$^{13}$C–δ$^{18}$O interactions prior to 6 Ma, while also accommodating anti-phase δ$^{13}$C–δ$^{18}$O behaviour on million-year timescales (e.g. between 20 and 15 Ma). This is because the key-factor in the in-phase hypothesis is the absence of seasonal extremes during eccentricity minima, which remain independent of longer-term cooling or warming trends of the late Cenozoic. Importantly, this hypothesis is compatible with the observed phase shift documented at the 405-kyr eccentricity rhythm (same methodology as for the 100-kyr focused analysis, Supplementary Fig. 8), also occurring around 6 Ma.

Conversely, during Interval III, we propose that eccentricity minima resulted in cold and dry climate conditions (hence, δ$^{18}$O$_{benthic}$ ↑), a shrinkage of high-latitude $^{12}$C-enriched continental

carbon reservoirs and a net carbon transfer towards the deep ocean (hence, $\delta^{13}C_{benthic}$ ↓)[14,23]. In other words, the response of the continental carbon reservoir to eccentricity minima changes from an expansion (in-phase hypothesis, >6 Ma, Fig. 4d–g) to a contraction (anti-phase hypothesis, <6 Ma, Fig. 4b, c). This switch in carbon cycle dynamics chiefly coincides with the late Miocene global cooling[24] and with the establishment of a persistent and dynamic ice sheet in eastern Greenland[25–27], although the exact timing of the latter remains subject of discussion. Yet, the advent of (ephemeral or otherwise) continental ice and substantial cooling in the northern hemisphere (NH)[24] could have intensified the areal competition between different high-latitude continental carbon reservoirs (polar desert, tundra, peatland and wetland, boreal forests). Global carbon cycle dynamics could be substantially affected by such competition, as polar desert and tundra ecosystems have significantly smaller carbon storage capacities compared to boreal forests[28]. Herbert et al.[24] high-lighted a late Miocene cooling interval around 6 Ma and suggested transient NH glaciation at such an early age. Their claim is supported by several marine records of ice-rafted debris from offshore Norway, Greenland, Iceland, and Alaska[26,29–35], suggesting that there were (at least) terminal marine-based glaciers at this time. Ocean Drilling Programme Site 918 is of particular importance in this context, as this marine core displays alternations between diamictites, ice-rafted debris, and marine sediments free of dropstones, from the late Miocene onward, and all throughout the Pliocene[26]. The direct indications for small NH continental ice caps in the late Miocene (ice-rafted debris and diamictites) are in unison with a model-based search for the threshold value for Cenozoic bipolar glaciation: a simulation of NH ice sheets at late Miocene levels of 420 ppm $CO_2$ shows the appearance of continental ice broadly in the same places as described in the geological study of Larsen et al.[26] (Fig. 3c in DeConto et al.[36]). Furthermore, different authors found a late Miocene increase in carbon cycle sensitivity to obliquity forcing (see Figs. 6c, 7f in Drury et al.[6,19], respectively), which we interpret as indirect evidence for a latitudinal shift of the decisive climate–carbon cycle feedback mechanisms towards the high latitudes, where obliquity has a greater relative impact on insolation variability. We advocate high-latitude biomes competing for areal extent to constitute that critical high-latitude dynamic after 6 Ma (Fig. 4b, c).

The lines of previously published evidence for late Miocene polar cooling, in combination with our phase-shift, suggest that from 6 Ma onwards, eccentricity minima are sufficiently cold for NH carbon-poor biomes to advance at the expense of high-latitude biomes with a higher carbon storage capacity. A net flux of $^{12}C$-enriched organic matter thus occurs from the land to the ocean when Earth's orbit dampens NH summer insolation (Fig. 4b, Fig. 1 in Brovkin et al.[37]). Such astronomical configurations likely correspond to what Herbert et al.[24] call overshoots in cooling and might have allowed for transient late Miocene glaciation of (eastern) Greenland. With glaciation, a net flux of $^{16}O$-enriched $H_2O$ occurs from the ocean to the continent. The oceans thus become enriched in $^{18}O$ and $^{12}C$ during cold phases after 6 Ma, which is in accord with the observed anti-phased behaviour of $\delta^{13}C$ and $\delta^{18}O$. The Arctic warming during the Pliocene[24,38] seems insufficient to reverse this dynamic, with Pliocene sea surface temperatures (SST) in the NH high latitudes (>50°N) failing to return to the comparably warm Tortonian conditions (8–12 Ma in Fig. 2a of Herbert et al.[24]). The warmest intervals of the Pliocene, even when they were characterized by little or no NH continental ice, thus stayed with anti-phased climate–carbon interactions, primarily driven by the areal competition between tundra and boreal forest. It is important to note that the reported phase-switch unfolded during the global

rise of C4-dominated ecosystems and more widespread arid conditions[39,40]. The upsurge of grasslands and shrublands made continents more susceptible for erosion, and thus magnified the $^{13}C$-depleted carbon fluxes between the continental and ocean reservoirs[24,41], whenever this continental carbon reservoir expanded or shrank. The rise of the C4 plants thus possibly amplified the observed phase-switch.

The key to our preferred hypothesis lies with the relative areal fraction between high-latitude continental carbon reservoirs, each with a different carbon storage capacity per unit area. However, the idea of a persistent ice sheet on Greenland since the late Miocene remains controversial. Paleobotanical data suggest boreal forests up to 80°N for the late Miocene, with mean annual temperatures up to 6 °C warmer than the pre-industrial[42]. Moreover, the boreal realm is thought to remain extensive throughout most of the Pliocene[43]. The botanical data thus place an important question mark over the idea of eccentricity-paced expansion/contraction cycles of the Arctic cryosphere since the Late Miocene. On the other hand, we emphasize that these botanical data from Canada, Alaska, and Siberia are not necessarily at odds with a continuous—but volatile—Arctic cryosphere (polar desert and tundra) over the past 6 Myr, considering the current and past[38] large longitudinal climate variability along the Arctic circle.

Although we favour the hypothesized areal competition hypothesis, we cannot currently rule out an alternative hypothesis for the observed phase-switch. Ocean ventilation is known to contribute to the anti-phase behaviour after 6 Ma, with the augmented storage of isotopically negative metabolic $CO_2$ in the deep ocean during glacials[44,45]. Notably the more sluggish production of North Atlantic Deep Water (NADW) during Pleistocene glacial intervals ($\delta^{18}O_{benthic}$↑) resulted in poor ocean ventilation ($\delta^{13}C_{benthic}$↓)[46], as an anti-phase relationship. During the Miocene, on the contrary, ocean ventilation is thought to respond inversely to astronomically forced cool and warm spells: Holbourn et al.[47] demonstrate that mid-Miocene intervals of global cooling ($\delta^{18}O_{benthic}$↑) coincide with a substantial improvement in the ventilation of the deep Pacific ($\delta^{13}C_{benthic}$↑). Hence, the observed mid-Miocene in-phase $\delta^{13}C$–$\delta^{18}O$ behaviour could also be explained through fluctuations in ocean ventilation. However, exactly why ocean ventilation was vigorous during cold periods in the mid-Miocene, but sluggish during cold periods in the Pleistocene is poorly understood: It remains unclear which modification(s) of Earth's boundary conditions could have been responsible for reversing the response of ocean ventilation to astronomical forcing. We note that our preferred areal competition hypothesis is compatible with an ocean ventilation mechanism. In fact, the $^{12}C$-enriched carbon removed from the continental carbon reservoirs could be efficiently stored in the deep-ocean during cold spells after 6 Ma, and during relatively warm periods prior to 6 Ma. However, we consider the areal competition among high-latitude biomes the most parsimonious explanation as the magnitude of the continental biomass isotopic signal is likely to be significantly larger than a mere ventilation signal[48]: A relatively small shrinkage of the $^{12}C$-enriched continental carbon reservoirs in the NH high-latitudes can cause a notable shift in deep-sea $\delta^{13}C_{DIC}$. However, a much larger surge in primary productivity and respiration is needed to explain a deep-sea $\delta^{13}C_{DIC}$ fluctuation of equal size.

**Weathering locus modulates Oligo–Miocene leads and lags.** To better understand the climate–carbon cycle dynamics during Intervals I and II (prior to 6 Ma), we examine the leads and lags between $\delta^{13}C$ and $\delta^{18}O$ (Figs. 2 and 3a). Their overall in-phase behaviour at 100-kyr time scales is characterized by an average lag

of $\delta^{13}C$ compared to $\delta^{18}O$ of 1.9 kyr (0.12 rad) in Interval I, and 2.5 kyr (0.16 rad) in Interval II. This small but noticeable phase lag of $\delta^{13}C$ has been observed before[10,49,50] and explained by the ~65 kyr residence time of carbon in the ocean[9,51]. The large size of the ocean carbon reservoir, in combination with a non-linear climate response to precessional forcing, causes the response time of the carbon cycle to eccentricity to be several thousand years slower than that of the climate–cryosphere system.

The evolutive phase analyses reveal previously unobserved periodic oscillations around the zero-phase, which are shown in Figs. 2 and 3a. Intervals during which $\delta^{18}O$ leads $\delta^{13}C$ (positive phase) alternate with periods where the opposite is the case with a multi-million-year wavelength. This is an intriguing observation, as several previous studies speculated on carbon-cycle sensitivity to very long (~2.4 Myr) eccentricity cycles[52–55]. Here, we find the oscillations around the zero-phase to be consistent with long-term 2.4-Myr eccentricity forcing. A Monte-Carlo-based nonparametric method 'surrogateCor'[56,57] (see "Methods" section) ensures that the observed oscillations are non-random: $\delta^{13}C$ periodically leads, rather than lags $\delta^{18}O$, concurrent with extremes in the 2.4-Myr eccentricity cycle (caused by resonance between Mars' and Earth's orbits, i.e. $g_4-g_3$ term in Laskar et al.[58]; Fig. 3a). Intriguingly though, in Interval I (35–26 Ma) $\delta^{13}C$ leads during long-term eccentricity minima ($p$-value = 0.060; Fig. 3b), whereas in Interval II (26–6 Ma) $\delta^{13}C$ leads during long-term eccentricity maxima ($p$-value = 0.015; Fig. 3b).

We endeavour to conceptually explain the observed correlations between $\delta^{13}C$–$\delta^{18}O$ phase-relationships and long-term eccentricity forcing by looking towards carbonate and silicate rock weathering. Continental weathering processes are a dominant sink for atmospheric $CO_2$. On eccentricity time-scales, small variations in carbon flux through rock weathering can have a significant impact on the size of the atmospheric carbon reservoir and thus global climate[59] (Supplementary Fig. 7). Rock weathering and climate perpetually interact as weathering itself is also a function of climate (e.g. temperature). Yet, when a non-temperature-determined factor enhances rock weathering (e.g. tectonic uplift), an invigorated carbon cycle expands its control on climate and thereby reduces its time-lag relative to the climate system on orbital time-scales. In this conceptual model, the time-lag of the carbon cycle compared to the climate system decreases or even switches sign to a carbon-cycle lead, when continental weathering becomes more important within the global carbon cycle for a reason other than $CO_2$-induced temperature change. Translated to our benthic isotope proxies, weathering maxima would allow $\delta^{13}C$ to reduce its time-lag with respect to $\delta^{18}O$, or even to start leading $\delta^{18}O$, when continental weathering is enhanced. We do not claim that continental weathering directly controls the marine $\delta^{13}C_{DIC}$ signal. We rather advocate that continental weathering acts as a mechanism through which the carbon cycle (approximated by $\delta^{13}C_{benthic}$) can drive the climate–cryosphere system (approximated by $\delta^{18}O_{benthic}$), thereby ultimately reversing their common phase relationship.

We thus interpret $\delta^{13}C$ leads over $\delta^{18}O$ as an indicator for weathering maxima. Our phase-analysis shows that they occurred during 2.4-Myr minima in Interval I, yet during 2.4-Myr maxima in Interval II (Fig. 3). In the next two paragraphs, we propose that this change in the climate carbon-cycle response to astronomical forcing is associated with a shift in the locus of the major continental weathering $CO_2$ sink, from glacial-induced (35–26 Ma) to monsoon-driven (26–6 Ma).

In the Oligocene climate state of Interval I, with a large unipolar ice sheet on Antarctica, long-term 2.4-Myr eccentricity minima allowed for relatively stable and cool climate conditions to prevail over several 10,000 years[5]. Such an orbital configuration would have favoured ice sheet growth and glacial

weathering[60] (Fig. 4f). An early Oligocene modelling study by Pollard et al.[61,62] shows that—on the global scale—$CO_2$ consumption through silicate weathering could increase with cooling global climate conditions. This finding illustrates that, potentially, the Oligocene could have experienced a positive silicate weathering feedback loop. Building on the outcomes from these modelling studies, we suggest that the Oligocene climate state could support a system in which $CO_2$ is more efficiently removed from the atmosphere during the colder intervals associated with eccentricity-minima. Accordingly, the in-phase hypothesis for Interval I (35–26 Ma) stipulates that an initial astronomically induced global cooling (e.g. eccentricity minima) could have resulted in more silicate weathering, primarily in the higher latitudes in the form of glacial weathering, and a more efficient $CO_2$ sink. This, in turn, leads to more cooling. Not only does this scenario describe a positive weathering feedback mechanism, it also represents a mechanistic chain in which the carbon cycle leads climate. Therewith, this scenario provides a plausible explanation for the observation in Interval I that $\delta^{13}C$ leads $\delta^{18}O$ during 2.4-Myr eccentricity minima. Beyond the 2.4-Myr eccentricity minima, the weathering $CO_2$ sink would have been too weak to stimulate the carbon cycle sufficiently for it to lead the climate system. Hence, beyond the 2.4 Myr eccentricity minima, we observe the natural phase-lag prevailing, with $\delta^{18}O$ leading $\delta^{13}C$ by a few thousand years on ~100-kyr timescales.

Throughout Interval II (26–6 Ma), $\delta^{13}C$ leads $\delta^{18}O$ for short-lived intervals that correspond to 2.4-Myr eccentricity maxima. This is a remarkable observation, given the long duration and marked climate variability of Interval II[1]. To understand the differences to Interval I, one has to bear in mind that continental (Antarctic) ice volume was lower[63] and that rapid uplift of the Himalayan orogen[64] occurred during Interval II. Together, these two factors facilitate an intensified hydrological cycle and a monsoon-dominated climate state[65], which contrasts with the more zonal global climate organization of Interval I[65,66]. We suggest that this climatic restructuring induced a latitudinal shift of the main weathering locus. High-latitude glacial weathering would have dominated during Interval I, whereas during Interval II, the dominant weathering processes probably related to wet/dry variations at low latitudes. Monsoonal dynamics are thus key to explain the $\delta^{13}C$ leads during 2.4-Myr eccentricity maxima in our hypothesis for Interval II. The characteristic precession-driven wet/dry variability of a monsoonal climate system is amplified under eccentricity maxima, which in turn would enable stronger chemical weathering (Fig. 4e). During eccentricity minima, on the other hand, seasonal insolation extremes are subdued for a prolonged period of time and chemical weathering in the low latitudes remains modest (Fig. 4d). Since eccentricity minima are typically associated with cooler conditions than eccentricity maxima, the rock weathering $CO_2$ sink here acts as a negative feedback mechanism: The most pronounced eccentricity maxima, which occur during 2.4-Myr eccentricity maxima, would have triggered a greater low-latitude hydrological cycle response, enhanced chemical weathering, an efficient $CO_2$ sink and a situation where the global carbon cycle leads climate. This leading role is depicted by slightly negative $\delta^{13}C$–$\delta^{18}O$ phase differences in our proxy time-series (Fig. 3a). These Interval II interactions between eccentricity, monsoonal intensity and weathering response have been numerically modelled and comprehensively described by Ma et al.[67] for the Miocene. It should be noted though, that these authors emphasize the role of enhanced $CaCO_3$ burial in tropical shallow seas during eccentricity maxima, resulting in a depletion of marine DIC $\delta^{13}C$, whereas our cartoons in Fig. 4 display the net $^{12}C$ fluxes between the continental biosphere and the oceans.

To explain the suggested shift in weathering regime from the high (glacial) to the low (monsoonal) latitudes, we suggest that Earth underwent a transition from an Oligocene (Interval I) climate state with distinct climate belts and a fully glaciated Antarctica to a Miocene (Interval II) monsoon-dominated climate state with stronger north–south atmospheric circulation, promoted by a main uplift phase of the Himalayan orogen unfolding around that time[64] (Fig. 4a). The observed change in phase-behaviour from Interval I to Interval II is synchronous with enhanced Himalayan mountain building[68], but also coincides with a significant shrinkage of the Antarctic cryosphere (Fig. 2) and the advent of a relatively warm part of the late Cenozoic ice age. Which of these three changes in Earth's boundary conditions was decisive in modifying the weathering regime between both intervals remains unclear. The concepts presented here will require testing with Earth System models that can simulate the temporal and spatial distribution of weathering, using time-varying orbits, and under different boundary conditions: with and without a Himalayan mountain range and with different amounts of continental ice on Antarctica.

**Concluding remarks**. Our analysis of the leads and lags between the climate and the carbon cycle results in the delineation of three marked intervals of distinct climate–carbon cycle interactions.

The youngest time interval (Interval III; 0–6 Ma) corresponds to an Earth that is cooler than the earlier Cenozoic, especially in the NH high-latitudes. Under these conditions, deep-sea $\delta^{13}C$ and $\delta^{18}O$ show clear anti-phased behaviour on 100-kyr time-scales. Our anti-phase hypothesis explains this observation by suggesting contractions of Arctic continental carbon reservoirs during astronomically induced cool intervals. Whether ice sheets have been persistent in the Arctic over the last 6 Myr remains an open question. But even without ice, shifts in the areal distribution of high-latitude biomes may have been sufficient to cause significant continental carbon storage fluctuations. Still, continental ice could be a powerful part of the equation, given its virtually negligible carbon storage capacity. In addition to the proposed mechanism, ocean ventilation dynamics may have contributed to the anti-phased $\delta^{13}C$–$\delta^{18}O$ behaviour of the last 6 Myr, albeit with smaller amplitude.

Prior to 6 Ma (Intervals I and II), we find in-phase $\delta^{13}C$–$\delta^{18}O$ behaviour. Interestingly, ~2.4 Myr eccentricity cycles modulate the observed $\delta^{13}C$–$\delta^{18}O$ phase oscillations around the zero-phase: During Interval I (35–26 Ma), $\delta^{13}C$ leads $\delta^{18}O$ in ~2.4 Myr cycle minima, whereas during Interval II (26–6 Ma), $\delta^{13}C$ leads $\delta^{18}O$ in ~2.4 Myr eccentricity maxima. To explore which mechanisms enable ~2.4 Myr eccentricity to influence the leads and lags in System Earth, we suggest a conceptual model where the main weathering locus shifts from the high latitudes in Interval I (in-phase glacial-weathering hypothesis) to the low latitudes in Interval II (in-phase monsoonal hypothesis).

Further modelling work is needed to test our concepts. However, this work explicitly shows that the interaction between climate and carbon cycle has changed profoundly following large changes in continental ice volume, tectonic uplift and atmospheric circulation patterns (e.g. monsoons). Our findings provide new evidence and ideas that will direct further research toward a process-level understanding of how changing planetary boundary conditions influenced climate–carbon cycle interactions and Earth System Sensitivity[69] in the past.

## Methods
**Megasplice construction**. The nine $\delta^{13}C_{benthic}$ records that are used in the megasplice (Fig. 1c) were selected with the aim of maximizing time-resolution throughout. All of the individual records were originally published with an astronomically calibrated age–depth model (Supplementary Table 1) and are

characterized by sedimentation rates (typically 2–4 cm/kyr) that are sufficiently high such that precession cycles (i.e. the shortest astronomical cycle) are not blurred by bioturbation. We first scrutinized and then adopted all of those original age–depth models, except for Site U1146. At that site, the published age–depth model is not compatible with the preceding and succeeding records. We revised the 1146 age–depth model by tuning characteristic $\delta^{13}C_{benthic}$ cycles in the late Miocene to the same obliquity cycles in the La2004 solution as proposed in Drury et al.[70] and Zeeden et al.[71]. This revision implies a maximum change in the Site 1146 age–depth relationships of three obliquity cycles and is documented in Supplementary Table 2. We also implemented a slight revision of the 1146 age model in the middle Miocene to optimize the match between the U1337 and U1338 records. With this revision, we ensure that all $\delta^{13}C_{benthic}$ records in the megasplice are mutually consistent.

The disadvantage of the megasplice, compared to $\delta^{13}C_{benthic}$ compilations, is that data from single sites do not only reflect whole-ocean conditions, but also contain a regional and local component. To account for this regional component, we consider different isotope trends in individual ocean basins (Supplementary Table 1), using the largest basin (the Pacific Ocean) as the reference. All records in the megasplice are shifted towards the general trend of the Pacific Ocean as shown in Fig. 6 of Cramer et al.[2]. This procedure allows for the comparison of globally distributed $\delta^{13}C_{benthic}$ at orbital resolution, as illustrated in Supplementary Figs. 1–3. The final step in the construction of the megasplice consists of the establishment of the most suitable positions to transition between records, so as to ensure smooth transitions and to maximize temporal resolution (Supplementary Fig. 4). The result, the megasplice, contains 12,493 $\delta^{13}C_{benthic}$ measurements with an average temporal resolution of 2.80 kyr.

**Phase analysis**. We calculated the phase relationship between $\delta^{13}C$ and $\delta^{18}O$ from all studied sites with cross-spectral fast Fourier transform analysis, as implemented in the *crossSpectrum* function, which is part of the *IRISSeismic* package for R[72,73]. Cross spectra were calculated using a sliding window approach, where 1.2-Myr-wide windows were moved through the individual datasets with steps of 0.2 Myr. To determine the phase relationship between $\delta^{13}C$ and $\delta^{18}O$ on 100-kyr eccentricity time-scales, we considered the frequency range between 90 and 135 kyr. Within this range, we selected the phase from the frequency with the highest coherency (Supplementary Fig. 6). Since $\delta^{13}C$ and $\delta^{18}O$ are always compared within individual sites, this technique is impervious to minor inaccuracies in age models (Supplementary Table 3, Supplementary Fig. 5).

We also determine phase relationships between $\delta^{13}C$ and $\delta^{18}O$ on 405-kyr eccentricity time-scales, using the same numerical recipe considering periodicities between 370 and 435 kyr. We increased the window-size from 1.2 to 1.8 Myr. This window-size is a compromise between a large-enough window to have enough 405-kyr cycles per analysis, and a small-enough window that still allows for the analysis of the connection between phase and the 2.4-Myr cycle (Supplementary Fig. 8).

**Correlation analysis with surrogateCor**. We observed intriguing oscillations in the $\delta^{13}C$–$\delta^{18}O$ phase series presented in Fig. 2 and wanted to enquire whether there was a correlation between these oscillations and 2.4-Myr eccentricity. But, both time series are serially correlated (nonzero autocorrelation), which implies that we cannot use classical statistics to estimate the significance of the correlation. Instead, we use a nonparametric method to estimate the statistical significance of the correlation between phase-oscillations and 2.4-Myr eccentricity, developed by Ebisuzaki[56]. This method is implemented in the function surrogateCor[57], which is part of the astrochron package for R[73,74]. The eccentricity solution was resampled on the sample grid of the phase-relationship data, using piecewise linear interpolation. Subsequently, the Spearman Rank correlation coefficients are calculated. Next, the surrogateCor function carries out the same correlation analyses for 10,000 Monte Carlo simulations, using phase-randomized surrogates. The surrogates are subject to the same interpolation process and compensate for the auto-correlation that characterize both time-series[56]. The correlation coefficient of the data is then compared to the distribution of correlation coefficients obtained from the surrogates and allows for the determination of a p-value for the correlation.

## Data availability
The $\delta^{13}C$ megasplice can be retrieved from https://doi.pangaea.de/10.1594/PANGAEA.918140. The individual benthic carbon isotope records that make up the megasplice and/or support the findings of this study are available through PANGAEA and through the National Climatic Data Center with the DOI identifiers and data links as listed in Supplementary Table 1.

## Code availability
R-scripts for megasplice construction, phase-analysis and correlation analysis are available at https://github.com/dadevlee/megasplice_d13C.

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

## Acknowledgements

This research used samples and data provided by the International Ocean Discovery Programme (IODP). Figure 4 was created using clipart designed by Freepik. This research is part of ERC Consolidator Grant EarthSequencing awarded to H.P. DDV is currently funded through the Cluster of Excellence: The Ocean Floor–Earth's Uncharted Interface.

## Author contributions

D.D.V. and H.P. designed research. D.D.V., A.J.D., M.V. and D.L. verified age models of individual isotope records and constructed the $\delta^{13}C_{benthic}$ megasplice. D.D.V., A.J.D., M.V., F.R., D.L. and H.P. contributed to data interpretation. D.D.V. wrote the manuscript with input from A.J.D., M.V., F.R., D.L. and H.P.

## Funding

## Competing interests

The authors declare no competing interests.
