## [Peer Review File · Nature Communications]

Reviewers' comments:

Reviewer #2 (Remarks to the Author):

Thirty-five million years of changing climate – carbon cycle dynamics

David De Vleeschouwer, Anna Joy Drury, Maximilian Vahlenkamp, Fiona Rochholz, Diederik Liebrand, Heiko Pälike

This paper investigates the changing high frequency orbital interactions between the climate cryosphere system and global carbon cycle under evolving geological boundary conditions and through the major climate developments of the past 35 Myrs.

It involves a synthesis of deep sea $\delta^{13}\text{C}$ and $\delta^{18}\text{O}$ data but is different from earlier data compilations of Zachos et al., 2001, 2008; Cramer et al., 2009, in that the focal record is a continuous composite, comprised of segments from different deep sea sites around the globe (splice), whereas the former studies were based on compilations of all amassed data. The mega splice approach allows the authors to perform long-term time series analysis and carefully inspect the evolving global carbon system ($\delta^{13}\text{C}$) response to orbital forcing.

The paper is well written, the figures of a high quality and the mega splice is unique for this time interval, making it a valuable resource for paleoclimate analysis on its own.

The main conclusion that the 2.4-Myr eccentricity modulates the in-phase relationship between $\delta^{13}\text{C}$ and $\delta^{18}\text{O}$ during the Oligo-Miocene (34-6 Ma), potentially related to changes in continental weathering, is the important foundation here.

The more tangible and accessible finding that the authors point to is what they describe as 'a striking switch from in-phase to anti-phase behaviour (in the high-frequency orbital responses) at 6 Ma', and which they attribute to the 'onset of Arctic glaciation' and the emergence of bipolar ice sheets; the interpretation here is that northern ice sheet expansion enabled eccentricity to exert its control on the carbon system in a different way, i.e. massively impacting carbon cycling by changing the size of continental carbon reservoirs.

In coming to this conclusion, the authors build directly on the work of Turner, 2014 (appropriately cited). Turner does herself identify this astronomical shift, i.e. of the high-frequency relationship between $\delta^{13}\text{C}$ and $\delta^{18}\text{O}$, using a spliced record of sorts, however, she does so in a more general way for the Plio-Pleistocene interval, and comparing the late Neogene to the much warmer early Paleogene, and more generally suggesting it might be associated with the onset of Northern hemisphere glaciation around 5 Ma.

The work and conclusions presented in the present contribution are potentially of high impact, that will represent an important advance of significance to specialists within the fields of palaeoceanography and climatology, and in predicting further climate responses in a warming deglaciating world.

However, before this can happen there are some unclear aspects and areas where the unique elements need to be clarified, including:

-How the Mega splice presented here differs from the approach of Turner, 2014 (many of the sites involved in both are the same).

- The treatment of the Pacific records in the late Neogene, i.e. the practice of 'adjusting' isotopic records. This would benefit from a more clarification to ensure its appropriateness for high resolution time series analysis. i.e. can you be sure that the adjustment approach does not introduce high frequency artefacts impacting interpretation of the change in orbital phasings; after all the time when this starts to matter (~7 Ma), as the Panama seaway beginning closing, global

circulation path of surface and deep waters was gradually imposed, and the strong isotopic gradients between the North Pacific bottom waters developed, largely coincide.

-Are the authors really trying to say that Northern hemisphere glaciation begin at 6 Ma, and how does this differ in practice with say Herbert et al., 2016's estimate of/evidence for 7- 5.4 Ma?

-If the important conclusion is a firmer timing of the onset of Arctic glaciation what is meant by Arctic glaciation? Is it only glaciation on Greenland or other parts of the Arctic? This needs to be better linked with existing geological evidence. Should we expect a "threshold response" of the phasing even when the inferred Arctic ice sheets are only tiny, i.e. 6 Ma, as shown in the hypothesized 'ghost' range of northern ice sheets in Fig. 2.

Expansion on the above points and other comments

- The introduction does a good job of emphasizing how the results and treatment of data are different from early syntheses of Zachos et al., 2001 and Cramer et al., 2009.
- I'm not sure that the title matches the conclusions of the paper as it does not allude to the main conclusions in anyway. Too general and rather flat.
- One weakness in the current paper is that the difference in approach compared to Turner, 2014, is not totally obvious. Turner used what she calls 'multiple high-resolution snapshots throughout the Cenozoic', including many of the same ODP sites as used here. Here it is called a 'continuous' megasplice, but Turners time windows are also contiguous. Please emphasize the differences and developments in summary form in the main text.
- Inter basin d13C correction: Making this splice is challenging due to the need to switch back and forth between ocean basins because no single record, or single basin record exists. This becomes an increasing issue from the Miocene when the closing of Panama changes circulation pathways (decreased mixing of Atlantic and Pacific water masses), and Pacific and Atlantic benthic d13C diverges (Cramer et al., 2009). The authors have overcome this by applying a 'basin correction', using the Pacific as the reference basin, an approach based on the work of Cramer et al., 2009, and following De Vleeschouwer et al., 2017. Can the authors elaborate a little more on the process of correction as it is not adequately shown or interrogated.

For the last 10 million years (through which the northern ice sheets are inferred to have developed), Fig. 1 shows just the single 'corrected' curve, which for this time interval is based on Site 1267, 1264, 982, all Atlantic Sites. However, the time series analyses that lead to the conclusions of different isotopic phasings from 6 Ma as I read it are conducted on the non-adjusted separate Atlantic and Pacific segments ...correct? Certainly, the far left and far right hand panels of Fig. 2 show Pacific and Atlantic data. However, I am confused as to what sites from the Pacific and Atlantic extend through the all-important 0-10 Ma period, which is where we need to focus to look for the describe system shift.

Extended data Fig. 1 goes some way towards this by comparing an Atlantic and a Pacific record, Site 1267 and 1143 respectively, however, this only shows the overlap through 0-2.6 Ma, whereas main Fig. 2 implies additional Atlantic - Pacific site comparisons exist that we are not shown. The caption for Fig. 2 implies that through 0-10 Ma, the green and red colour coding of the symbols for the Pacific comprises Sites 1338, 1337, 1146, and light blue and grey blue for the Atlantic comprises 1267, 1264, 962, 982. These sites appear as segments in the Mega splice (Fig. 1) but it is not clearly discussed that other segments from these sites become important for the analyses in Fig. 2. To this end it would be good to see a second version of Fig. 1 (Extended data), for example that shows the 'raw' uncorrected isotopic records examined, not just the segments in the splice but the other bits in Fig. 2. That aren't properly treated anywhere.

Finally still on this point, Fig. 2 right hand panel shows that for the Pacific records spanning 10-5 Ma there are not continuous, instead they jump to different sites at the exact time of the inferred 6 Ma system shift. This is problematic and needs some discussion.

- According to the analyses and arguments presented, the signal of northern hemisphere glaciating is rather subtle at first, i.e. a change in the high frequency orbital behaviour of the carbon cycle linked to size of the terrestrial soil C reservoir, rather than some obvious state change in the benthic d18O or d13C. Please clarify why there would be no clearer change in d18O due to ice growth response at 6.0?
- The interbasin d13C adjustment factor used in this paper (shown in Exte. Data Fig. 1) and derived Cramer et al., 2009 (Fig. 6) implies that the correction is smooth, so no differences in d13C response to orbital forcing between basins. How sure of this are you? Emphasize better that there is convincing evidence that there is a strong global d13C heartbeat picked up by benthic d13C, so similar orbital forcing detected despite local differences in basinal d13C (references needed there). This has to be the starting point. Could be done around lines 31-32, and or lines 47-58.
- In the paper by Cramer et al., 2009 (Fig. 6) the inter basin d13C divergence begins at around 12-13 Ma, thus I would like to see the d13C of Site 1267 and Site 1143 compared over this longer duration, including the raw data before the Correction is applied to Site 1267. In this figure before the correction and over a longer period, encompassing the Miocene divergence.
- Lines 59-60 says that this study explores Milankovitch forcing under evolving geological boundary conditions. To me this includes ice sheets, (sea level), ocean gateways, and CO₂. Some of these are represented in Fig. 2, but major tectonic ocean gateway changes are not. Should the Panama Seaway closure and Arctic deep water passage opening (Fram Strait) etc not be annotated in here somewhere, with connected discussion? This is important since both have been connected the timing of onset of northern glaciation at various times. It would be good to mark these tectonic closures and uncertainty on Fig. 2.
- What scale of glaciation and what should you call it? Herbert et al., 2016, a proxy-rich study of this same interval finds evidence for transient glaciations in the Northern Hemisphere and with a steepening of the pole-to-equator temperature gradient from 7-5.4 Ma. Vleeschower et al., call it Arctic glaciation? Some resolution on this please? What exactly is meant by this. Transient glaciations on Greenland? If you want to call it 'Arctic glaciation', how does the fossil data look for the rest of the Arctic?
- What do the sedimentation rates look like for all the sites in the megasplice in relation to interpreting signals, and their potential blurring by bioturbation?
- Fig. 1. Make it clear in the caption that panel A is the orbital template while B is the analysis of the mega splice. IODP does not need to be followed by the word 'program' since that is already in the acronym.
- Ext Data fig. 2. Clarify whether this is the Pacific corrected d13C or not. Show from where the Pacific data correction is applied, with bars/arrows or something.
- Ext Data fig. 5: In the caption, explain what the different coloured symbols represent?

Reviewer #3 (Remarks to the Author):

Review of d13C megasplice

As an overview, I admire the effort and ambition of producing the d13C megasplice and contrasting it with the evolution of climate as recorded by the d18O proxy. These compilations stimulate both further empirical research (testing hypotheses with new data) and conceptual work. They are typically widely cited for good reason. The team here has the right combination of stratigraphic skills and paleoclimatic expertise to produce a significant piece of work.

I find the current version to be too immature to publish. It relies too much on a finding already made by Kirtland Turner (2014)- the phase change at ~ 6 Ma without adding enough solid material either in a numerical/statistical or conceptual framework to stand apart. I'd urge the authors to dig deeper. There are several obvious targets that fall out of their compilation that would add new fodder. For example, they could follow the evolution of variance in d18O and d13C in their 1.2 Myr windows. This should give information on whether variance scales similarly between the cryosphere and whatever is represented by d13C or not. They could also do moving window correlation analyses to assess coupling (correlation) and scaling (slope) between the proxies. It also seems to me that by focusing uniquely on ~100 kyr rhythms they are neglecting the role of tilt and precession, which are almost certainly expressed in the megasplince time series.

I would urge a very complete rewrite and also omitting the final "Back to the Future" section, which is simply not compelling.

Abstract:

Wouldn't it be simpler to say that the "2.4-Myr eccentricity modulates the phase relationship between..."- the use of "in-phase" is hard to understand because it stands for one phasing (e.g. zero phase lag), not a modulated phase.

The statement "We hypothesize that Arctic glaciation and the emergence of bipolar ice sheets enabled eccentricity to exert a major influence on the size of continental carbon reservoirs." Is problematic on 2 fronts:

First, (and this comment will continue through the manuscript), I would advise the authors not to attribute a direct role to Eccentricity. It is possible that a direct and important link between Eccentricity and climate will emerge, but no one has identified it yet based on any physical model of climate. What is more likely is that behaviors (frequency spectra, phasing) with ~Eccentricity" wavelengths observed in the past arise through non-linearities in response to the precessional cycle.

A second problem is that most of the Pliocene (5.35-2.58 Ma) was above the threshold for Arctic glaciation so I would downplay the interpretation of bipolar ice sheets as a prerequisite for the ~6 Ma phase shift. I'm basing this on abundant Pliocene evidence from the circum-Arctic region, including palynology, the absence of ice rafting, as well as warm temperatures recorded in the North Atlantic and North Pacific, and the generally depleted benthic d18O values, which do not allow for much or any build-up of ice except for very short-lived events- too short lived to account for a general regime shift at 6 Ma.

A general problem is that "climate-carbon cycle interaction" is left very undefined. It would be nice if the authors can state some specific mechanisms (beyond simply invoking weathering changes, for example), as well as which way they would drive coupling of d18O and d13C observed. Granted, the authors do not have a full model and it's very hard to uniquely interpret the d13C signal, but I think they should suggest some isotopic scenarios that could be modeled, rather than simply presenting statistical associations.

Introduction:

It would be good to remind the reader of what the different components that could influence the d13C composition of the ocean are: changes in the terrestrial biosphere (less = negative signal to the ocean, citations to examples...?), burial of organic matter in the ocean (more = negative), ocean-atmosphere partitioning of CO2, long-term changes in CO2 sources and sinks to the atmosphere, etc. The 13C isotope system is way more complex than for d18O, which is why it's harder to understand.

I'd like the authors to provide the rationale for constructing the megasplince from single sites. I

believe they do this to choose optimal records from a sampling/stratigraphic point of view but the reader may not understand this. And, unlike the case for $\delta^{18}\text{O}$, we expect significant glacial-interglacial shifts in $\delta^{13}\text{C}$ at single sites that might reflect basin-basin fractionation rather than global DIC isotopic changes. We know that for the Pleistocene at least, the ^{13}C gradients between the Atlantic and Pacific are dynamic. Can the authors satisfy us that, at the orbital scale, the isotopic changes they document are in fact global?

I would also think some comments are in order on age dating and sample resolution. I have a lot of confidence in this group, but a summary statement of how sample data were aligned to the geological time scale would be useful.

"Hence, the global climate and global carbon cycle response to astronomical forcing, and specifically the response to eccentricity strongly differ from that point onwards." As alluded to above, it is very unlikely that the climate and carbon cycle respond linearly and/or directly to eccentricity. "405 kyr forcing" is also a dangerous phrase in the same light, as "forcing" implies a direct cause-effect.

The statement

"This divergence in $\delta^{13}\text{C}_{\text{benthic}}$ and $\delta^{18}\text{O}_{\text{benthic}}$ response to 405-kyr eccentricity forcing has been attributed to a weakening in the coupling between cryosphere and carbon cycle from the mid-Miocene onwards, most likely in connection to the expansion of Antarctica to its near-modern extent⁴⁻⁷"

Seems very naïve and neglects the complexity of what $\delta^{13}\text{C}$ actually means. We know from the late Pleistocene that the cryosphere and atmospheric CO_2 levels are EXTRAORDINARILY coupled. The point is that the $\delta^{13}\text{C}$ proxy represents multiple aspects of the carbon cycle, and not CO_2 level itself. I also think that a number of high-resolution studies, such as that of Holbourn and colleagues, would dispute the lack of $\delta^{18}\text{O}/\delta^{13}\text{C}$ coupling after the middle Miocene.

I'd like to know more about how the authors do their phase analysis. Since the "100 kyr" cycle is in fact a composite of multiple terms with two main components at ~123 and 97 kyr, one worries about spurious phase assignment. The authors state that they "selected the phase from the frequency with the highest coherency". This approach may be a bit arbitrary- it will depend to begin with on how high the coherencies are (is there good evidence for phase locking?) and then on the dispersion of coherencies within the 90-135 kyr band).

I would back up and look at how compelling the evidence is for continuous 100 kyr components matching the predicted eccentricity terms. I've seen cases where there is power at say 85 kyr or 145 kyr that is pseudo-100 kyr, but where there isn't good evidence of the expected 123 and 97 kyr components when subjected to statistical tests. The phase analysis only makes sense if one has evidence for the underlying eccentricity-related components in each window of the analysis. Figure 1 supports the common presence of ~100 kyr components but more details would help substantiate the use of the 90-135 kyr band.

A new result relative to the Kirtland Turner work is the look at oscillations around zero phase and the idea that these oscillations tie to the envelope of the long eccentricity cycles. I was glad to see that the authors do provide a statistical test to back up their proposed correlation.

I found that the section proposing a conceptual explanation for lead-lag relationships was hard to follow. In particular:

"But the response time of the climate system to eccentricity forcing increases when continental weathering becomes more important within the global carbon cycle. In such situations, global climate variability is strongly driven by the weathering CO_2 sink, in addition to direct Milankovitch forcing and other climate forcing factors³⁴. We therefore hypothesize that the carbon cycle expands its driving influence on climate during weathering maxima, thereby allowing $\delta^{13}\text{C}$ to lead $\delta^{18}\text{O}$."

One would think that there would be first order relationships therefore between: $\delta^{18}\text{O}$ (a rough proxy for the temperature-weathering effect) and $\delta^{13}\text{C}$ (a recorder of the weathering cycle?). But none is proposed- we only get a set of assertions that do not link to the observables. In particular, the logic of how weathering fluxes would shift the marine $\delta^{13}\text{C}$ inventory is not made explicit. One can try to infer it from the Figure 3 caption, but I find it opaque. It's also hard to understand the reasoning behind linking the response time of the climate system to eccentricity and to weathering. A mechanism would need to be asymmetrical (non-linear) to generate long-term modulations of any importance, since the eccentricity cycles return to a similar baseline ~ 100 kyr (I realize they do not strictly return to the same baseline, but close to it).

As to what would drive changes in weathering intensity, it's not clear that the authors have a hypothesis. If weathering is to lead $\delta^{18}\text{O}$, then there needs to be a process to initiate it separate from the cryosphere. Figure 3 suggests that the mechanism lies in the hydrological cycle. If so, aren't there implications that one would see signals at precession, which controls low latitude hydrology? These signals might not be resolved well, but they should be looked for.

I'm not sure I would characterize the interval from 26-6 Ma as one climatic state (Termed Interval II here). In fact, it encompasses significant late Oligocene/early Miocene ice ages, the partial deglaciation of the Antarctic (~ 17.5 -14 Ma) and then the refrigeration of the Antarctic. Given these large climatic swings, the absence of phase changes is all the more enigmatic, and suggests that the authors interpretations are simplistic.

I'm puzzled by the beginning of the "Back to the Future" section:

"Our analysis of leads and lags between the climate and the carbon cycle constitutes the first evidence for its regulation by continental ice volume and eccentricity."

The authors must not mean what they wrote- there is a huge literature on both topics. I think they didn't clearly articulate what they intend here. Furthermore, their previous analysis has not made clear that ice volume "regulates" carbon, in fact they contended the opposite in some cases.

Extended Figure 1: change to a light colored from black font for site identification- very difficult to read

Answers to reviewers.

Reviewer #2

This paper investigates the changing high frequency orbital interactions between the climate cryosphere system and global carbon cycle under evolving geological boundary conditions and through the major climate developments of the past 35 Myrs.

It involves a synthesis of deep sea $\delta^{13}\text{C}$ and $\delta^{18}\text{O}$ data but is different from earlier data compilations of Zachos et al., 2001, 2008; Cramer et al., 2009, in that the focal record is a continuous composite, comprised of segments from different deep sea sites around the globe (splice), whereas the former studies were based on compilations of all amassed data. The mega splice approach allows the authors to perform long-term time series analysis and carefully inspect the evolving global carbon system ($\delta^{13}\text{C}$) response to orbital forcing.

The paper is well written, the figures of a high quality and the mega splice is unique for this time interval, making it a valuable resource for paleoclimate analysis on its own.

The main conclusion that the 2.4-Myr eccentricity modulates the in-phase relationship between $\delta^{13}\text{C}$ and $\delta^{18}\text{O}$ during the Oligo-Miocene (34-6 Ma), potentially related to changes in continental weathering, is the important foundation here.

The more tangible and accessible finding that the authors point to is what they describe as 'a striking switch from in-phase to anti-phase behaviour (in the high-frequency orbital responses) at 6 Ma', and which they attribute to the 'onset of Arctic glaciation' and the emergence of bipolar ice sheets; the interpretation here is that northern ice sheet expansion enabled eccentricity to exert its control on the carbon system in a different way, i.e. massively impacting carbon cycling by changing the size of continental carbon reservoirs.

In coming to this conclusion, the authors build directly on the work of Turner, 2014 (appropriately cited). Turner does herself identify this astronomical shift, i.e. of the high-frequency relationship between $\delta^{13}\text{C}$ and $\delta^{18}\text{O}$, using a spliced record of sorts, however, she does so in a more general way for the Plio-Pleistocene interval, and comparing the late Neogene to the much warmer early Paleogene, and more generally suggesting it might be associated with the onset of Northern hemisphere glaciation around 5 Ma.

The work and conclusions presented in the present contribution are potentially of high impact, that will represent an important advance of significance to specialists within the fields of palaeoceanography and climatology, and in predicting further climate responses in a warming deglaciating world.

We thank Reviewer #2 for this clear and accurate summary of our work.

However, before this can happen there are some unclear aspects and areas where the unique elements need to be clarified, including:

Reviewer #2, Comment 1

How the Mega splice presented here differs from the approach of Turner, 2014 (many of the sites involved in both are the same).

We added the following text to the “Late Miocene” section of our manuscript:

Kirtland-Turner's approach is to calculate the average phase for individual $\delta^{13}\text{C}$ and $\delta^{18}\text{O}$ time-series, each a few million years long (Figure 2 in Kirtland-Turner11). For the last 35 Myr, Kirtland-Turner uses time-series from seven different ocean drilling sites, several of which also appear in the $\delta^{13}\text{C}$ and $\delta^{18}\text{O}$ megasplices. These records are, after all, the best available orbitally-resolved benthic isotope records for their specific time intervals. Contrary to the discrete site-by-site approach of Kirtland-Turner, we exploit the time-continuous nature of the megasplice and adopt a sliding window approach (see above and Methods). Our approach allows for a narrower determination of the exact timing of the phase switch, yet the fundamental outcome is congruent between Kirtland-Turner11 and this study.

Reviewer #2, Comment 2

The treatment of the Pacific records in the late Neogene, i.e. the practice of ‘adjusting’ isotopic records. This would benefit from a more clarification to ensure its appropriateness for high resolution time series analysis. i.e. can you be sure that the adjustment approach does not introduce high frequency artefacts impacting interpretation of the change in orbital phasings; after all the time when this starts to matter (~7 Ma), as the Panama seaway beginning closing, global circulation path of surface and deep waters was gradually imposed, and the strong isotopic gradients between the North Pacific bottom waters developed, largely coincide.

The reviewer correctly states that changes in the configuration of major seaways (Panama, Indonesian Throughflow, Drake passage, Fram Strait, ...) are responsible for fundamental changes in global circulation. This makes that, in the late Neogene, the $\delta^{13}\text{C}$ gradient between the Pacific and Atlantic significantly steepened (see left panel, which shows the adjustment applied to N-Atlantic IODP Sites for “basin-correction”). The low-frequency nature of this adjustment, however, makes that it is safe to say that the **basin-adjustment approach does not introduce high-frequency artefacts that might jeopardize any of our interpretations.**

To further illustrate this point, the right panel shows the power spectrum of the N-Atlantic – Pacific $\delta^{13}\text{C}$ gradient. There is negligible power at frequencies > 0.25 cycles/Myr (logarithmic y-axis, x-axis in cycles/Myr). Therefore, one can confidently say that this adjustment does not influence any of the Milankovitch frequencies.

In the revised manuscript, we emphasize this point in the description of megasplice construction (2nd paragraph) and with additional supplementary figures (Extended Data Figures 1 and 2).

Reviewer #2, Comment 3

Are the authors really trying to say that Northern hemisphere glaciation begin at 6 Ma, and how does this differ in practice with say Herbert et al., 2016's estimate of/evidence for 7- 5.4 Ma?

Our results are in line with Late Miocene the SST study by Herbert et al. (2016), the orbital imprint studies by Drury et al. (2016, 2018) and several lines of physical evidence by Ice Rafted Debris in the North Atlantic and North Pacific (e.g. Larsen et al, 1994, Krissek et al, 1995, Wolf-Welling et al., 1996, Fronval et al., 1995, John et al., 2002).

All these lines of evidence suggest Late Miocene global cooling and/or transient glaciation in the Northern Hemisphere. Herbert et al. refers to these transient glaciations as “overshoots” in cooling. **Our work now demonstrates that these overshoots more than likely correspond to orbital configurations that buffer or mute NH summer insolation (e.g. obliquity minima and eccentricity minima).**

In the revised version of the manuscript, we significantly expanded the discussion of how our new results match with earlier research. Therewith, we want to make clear that Late Miocene transient NH glaciations were sufficiently large to reduce the size of the continental biosphere C reservoir.

Reviewer #2, Comment 4

If the important conclusion is a firmer timing of the onset of Arctic glaciation what is meant by Arctic glaciation? Is it only glaciation on Greenland or other parts of the Arctic? This needs to be better linked with existing geological evidence. Should we expect a “threshold response” of the phasing even when the inferred Arctic ice sheets are only tiny, i.e. 6 Ma, as shown in the hypothesized ‘ghost’ range of northern ice sheets in Fig. 2.

Our understanding of “Arctic glaciation” is now better stipulated in the manuscript. We explicitly refer to marine records of diamicrites and Ice rafted debris in the North Atlantic and North Pacific of late Miocene age. We interpret our results in the context of the Larsen et al. (1994) hypothesis, which states that **glaciation began in southeast Greenland in the Late Miocene because of the combination of high precipitation and high topography in the area.** We also make the link with the modelling work by Rob DeConto and colleagues (2008), who show the appearance of small ice sheets in good agreement with the description / interpretation by Larsen et al. (1994) at 1.5 x PAL CO₂.

Reviewer #2, Comment 5

Expansion on the above points and other comments

The introduction does a good job of emphasizing how the results and treatment of data are different from early syntheses of Zachos et al., 2001 and Cramer et al., 2009.

Thank you. These sentences (in the Introduction and in the Methods section) remained unchanged.

Reviewer #2, Comment 6

I'm not sure that the title matches the conclusions of the paper is it does not allude to the main conclusions in anyway. Too general and rather flat.

Updated to a title that better matches with our main conclusion.

Reviewer #2, Comment 7

One weakness in the current paper is that the difference in approach compared to Turner, 2014, is not totally obvious. Turner used what she calls 'multiple high-resolution snapshots throughout the Cenozoic', including many of the same ODP sites as used here. Here it is called a 'continuous' megasplice, but Turners time windows are also continuous. Please emphasize the differences and developments in summary form in the main text.

See answer to Reviewer #2, Comment 1

Reviewer #2, Comment 8

Inter basin d13C correction: Making this splice is challenging due to the need to switch back and forth between ocean basins because no single record, or single basin record exists. This becomes an increasing issue from the Miocene when the closing of Panama changes circulation pathways (decreased mixing of Atlantic and Pacific water masses), and Pacific and Atlantic benthic d13C diverges (Cramer et al., 2009). The authors have overcome this by applying a 'basin correction', using the Pacific as the reference basin, an approach based on the work of Cramer et al., 2009, and following De Vleeschouwer et al., 2017. Can the authors elaborate a little more on the process of correction as it is not adequately shown or interrogated.

We drafted a new figure (Extended Data Figure 2) that shows the $\delta^{13}\text{C}$ **megasplice before basin correction** is applied. It thus shows the raw basin-specific isotopic signals. The black line on the figure represents the general $\delta^{13}\text{C}$ trend for the Pacific Ocean, as calculated by Cramer et al., 2009. The Atlantic $\delta^{13}\text{C}$ records in the megasplice are shifted towards this general trend during basin correction to obtain the *smooth* time-continuous $\delta^{13}\text{C}$ megasplice. Extended Data Figure 2 is our best attempt at making this visually clear.

At the same time, this new figure is helpful in demonstrating that the Pacific $\delta^{13}\text{C}$ trend by Cramer et al. (2009) is of low-frequency and therefore does not introduce any features that might interfere with our analyses in the Milankovitch band (Reviewer #2, Comment 2).

Reviewer #2, Comment 9

For the last 10 million years (through which the northern ice sheets are inferred to have developed), Fig. 1 shows just the single 'corrected' curve, which for this time interval is based on Site 1267, 1264, 982, all Atlantic Sites. However, the time series analyses that lead to the conclusions of different isotopic phasings from 6 Ma as I read it are conducted on the non-adjusted separate Atlantic and Pacific segments ...correct?

We hope to have addressed the first part of Comment 9 with the new Extended Data Figure 2. The reviewer is correct that phase-analysis is conducted on the **non-basin-corrected** (and non-spliced, thus **full time-domain** of the original datasets) segments. For clarity's sake, the basin correction only consists of a shift of the absolute $\delta^{13}\text{C}$ values (Extended Data Figure 2). This shift has no effect whatsoever on the results of $\delta^{18}\text{O} - \delta^{13}\text{C}$ phase analysis in Figure 2. Phase analysis results on the 100-kyr time-scale are identical whether one works with the adjusted or non-adjusted isotopic values.

Certainly, the far left and far right hand panels of Fig. 2 show Pacific and Atlantic data. However, I am confused as to what sites from the Pacific and Atlantic extend through the all-important 0-10 Ma period, which is where we need to focus to look for the describe system shift.

We added a legend to Figure 2 to make it readily clear which result correspond to which site.

Reviewer #2, Comment 10

Extended data Fig. 1 goes some way towards this by comparing an Atlantic and a Pacific record, Site 1267 and 1143 respectively, however, this only shows the overlap through 0-2.6 Ma, whereas main Fig. 2 implies additional Atlantic - Pacific site comparisons exist that we are not shown. The caption for Fig. 2 implies that through 0-10 Ma, the green and red colour coding of the symbols for the Pacific comprises Sites 1338, 1337, 1146, and light blue and grey blue for the Atlantic comprises 1267, 1264, 962, 982. These sites appear as segments in the Mega splice (Fig. 1) but it is not clearly discussed that other segments from these sites become important for the analyses in Fig. 2. To this end it would be good to see a second version of Fig. 1 (Extended data), for example that shows the 'raw' uncorrected isotopic records examined, not just the segments in the splice but the other bits in Fig. 2. That aren't properly treated anywhere.

We address this issue in a new Extended Data Figure 3. This figure shows **all raw uncorrected $\delta^{18}\text{O}$ and $\delta^{13}\text{C}$ records** selected for phase analysis.

Reviewer #2, Comment 11

Finally still on this point, Fig. 2 right hand panel shows that for the Pacific records spanning 10-5 Ma there are not continuous, instead they jump to different sites at the exact time of the inferred 6 Ma system shift. This is problematic and needs some discussion.

It is indeed unfortunate that the 926 and 982 records end just prior to the switch, while the Site 1264 record starts only after the switch. However, we do not consider this problematic as the whole idea behind the megasplice is, that one can build a single time-continuous record that holds an astronomical imprint in benthic $\delta^{13}\text{C}$ and $\delta^{18}\text{O}$ that is representative of the global Earth system response to astronomical forcing ("**a global carbon cycle astronomical heartbeat**"). This concept is substantiated by the inter-ocean comparison of Pleistocene (!!) $\delta^{13}\text{C}$ signals in Extended Data Figure 1 and by the inter-site correlation between the different megasplice records in Extended Data Figure 4.

Importantly, in the Pacific, the records from Sites 1337, 1338 and 1146 do span the crucial 10 – 5 Ma interval. These three sites all exhibit a simultaneous switch from in-phase to anti-phase behaviour at 6 Ma. This pattern agrees with what we observe in the Atlantic, albeit with 926 & 982 data before the switch and 1264 data after the switch.

Reviewer #2, Comment 12

According to the analyses and arguments presented, the signal of northern hemisphere glaciating is rather subtle at first, i.e. a change in the high frequency orbital behaviour of the carbon cycle linked to size of the terrestrial soil C reservoir, rather than some obvious state change in the benthic $\delta^{18}\text{O}$ or $\delta^{13}\text{C}$. Please clarify why there would be no clearer change in $\delta^{18}\text{O}$ due to ice growth response at 6.0?

In line with our answer to Reviewer #2 Comment 3, we now discuss our results in the light of earlier evidence (direct and indirect) for a Late Miocene expansion of the NH cryosphere in much more detail. In summary, our observations are in line with the interpretations of Herbert et al., who make a case for **transient glaciations during "cooling overshoots" in the Late Miocene**. Our results specify that these cooling overshoots (characterized by multiple late Miocene positive $\delta^{18}\text{O}$ excursions) correspond to astronomical configurations that dampen northern hemisphere summer insolation, like for example 100-kyr eccentricity minima.

Reviewer #2, Comment 13

The interbasin d13C adjustment factor used in this paper (shown in Extended Data Fig. 1) and derived from Cramer et al., 2009 (Fig. 6) implies that the correction is smooth, so no differences in d13C response to orbital forcing between basins. How sure of this are you? Emphasize better that there is convincing evidence that there is a strong global d13C heartbeat picked up by benthic d13C, so similar orbital forcing detected despite local differences in basinal d13C (references needed there). This has to be the starting point. Could be done around lines 31-32, and or lines 47-58.

We agree with the reviewer that this is a fundamental starting assumption of the megasplice. The assumption of “a global d13C heartbeat” is now explicitly stated in the second paragraph of the manuscript. We point the reader to Extended Data Figure 1 to illustrate the validity of this assumption. Indeed, this figure shows **a consistent astronomical signature between the Pacific and the Atlantic** for the Pleistocene. The fact that the assumption seems valid for the Pleistocene is of high relevance, as this is the Epoch of the Cenozoic with the highest ocean heterogeneity.

Reviewer #2, Comment 14

In the paper by Cramer et al., 2009 (Fig. 6) the inter basin d13C divergence begins at around 12-13 Ma, thus I would like to see the d13C of Site 1267 and Site 1143 compared over this longer duration, including the raw data before the Correction is applied to Site 1267. In this figure before the correction and over a longer period, encompassing the Miocene divergence.

Unfortunately, the 1267 and 1143 records do not extend back to the Miocene at orbital resolution. Instead, we generated a new figure panel (Extended Data Figure 1B) that shows the “global d13C heartbeat” at Sites 982 (N. Atlantic) and 1146 (Pacific) during the Late Miocene time of rapid inter-basin d13C divergence.

Reviewer #2, Comment 15

Lines 59-60 says that this study explores Milankovitch forcing under evolving geological boundary conditions. To me this includes ice sheets, (sea level), ocean gateways, and CO₂. Some of these are represented in Fig. 2, but major tectonic ocean gateway changes are not. Should the Panama Seaway closure and Arctic deep water passage opening (Fram Strait) etc not be annotated in here somewhere, with connected discussion? This is important since both have been connected the timing of onset of northern glaciation at various times. It would be good to mark these tectonic closures and uncertainty on Fig. 2.

Important **tectonic reconfigurations** (Drake Passage, Central American Gateway, Fram Strait, Bering Strait, Indonesian Throughflow) are added to Figure 2. In the manuscript text, these tectonic configurations are discussed in the context of inter-basinal d13C gradients.

Reviewer #2, Comment 16

What scale of glaciation and what should you call it? Herbert et al., 2016, a proxy-rich study of this same interval finds evidence for transient glaciations in the Northern Hemisphere and with a steepening of the pole-to-equator temperature gradient from 7-5.4 Ma. Vleeschouwer et al., call it Arctic glaciation? Some resolution on this please? What exactly is meant by this. Transient glaciations on Greenland? If you want to call it ‘Arctic glaciation’, how does the fossil data look for the rest of the Arctic?

This comment is addressed in our answers to Reviewer #2 Comment 3, 4 and 12.

Reviewer #2, Comment 17

What do the sedimentation rates look like for all the sites in the megasplice in relation to interpreting signals, and their potential blurring by bioturbation?

Sedimentation rates are of course a selection criterion for all sites in the megasplice, as sedimentation rate directly influences the ability of a specific site to record the shortest astronomical cycle, i.e. precession. This is now specified in the Methods section.

Reviewer #2, Comment 18

Fig. 1. Make it clear in the caption that panel A is the orbital template while B is the analysis of the mega splice. IODP does not need to be followed by the word 'program' since that is already in the acronym.

Done

Reviewer #2, Comment 19

Ext Data fig. 2. Clarify whether this is the Pacific corrected d13C or not. Show from where the Pacific data correction is applied, with bars/arrows or something.

The figure in question is now Extended Data Figure 4. This is the Pacific corrected data, and this information has been added to the caption. The need for arrows/bars has been alleviated by the introduction of Extended Data Figures 2 and 3, which are designed to clarify the basin correction process in detail.

Reviewer #2, Comment 20

Ext Data fig. 5: In the caption, explain what the different coloured symbols represent?

Color coding of the filled circles is identical to Figure 2. This information has been added to the figure caption of what is now Extended Data Figure 8.

Reviewer #3

As an overview, I admire the effort and ambition of producing the d13C megasplice and contrasting it with the evolution of climate as recorded by the d18O proxy. These compilations stimulate both further empirical research (testing hypotheses with new data) and conceptual work. They are typically widely cited for good reason. The team here has the right combination of stratigraphic skills and paleoclimatic expertise to produce a significant piece of work.

We want to thank Reviewer #3 for the confidence she/he has in our team.

I find the current version to be too immature to publish. It relies too much on a finding already made by Kirtland Turner (2014)- the phase change at ~ 6 Ma without adding enough solid material either in a numerical/statistical or conceptual framework to stand apart. I'd urge the authors to dig deeper. There are several obvious targets that fall out of their compilation that would add new fodder. For example, they could follow the evolution of variance in d18O and d13C in their 1.2 Myr windows. This should give information on whether variance scales similarly between the cryosphere and whatever is represented by d13C or not. They could also do moving window correlation analyses to assess coupling (correlation) and scaling (slope) between the proxies. It also seems to me that by focusing uniquely on ~100 kyr rhythms they are neglecting the role of tilt and precession, which are almost certainly expressed in the megasplice time series.

We appreciate the reviewers' suggestion to look at the evolution of variance, d13C vs. d18O correlation and slope. However, we decided to stick to the phase analysis for multiple reasons.

- (1) Despite the groundbreaking paper by Sandy Kirtland-Turner (2014), phase analysis remains undervalued as a tool to infer past carbon cycle – climate interactions.
- (2) The phase analysis presented in our manuscripts digs deeper into the Kirtland-Turner's original finding (simultaneous switch between Pacific & Atlantic, sliding-window approach pinpointing the switch at 6 Ma) and puts it into a context of NH glaciation inception.
- (3) The phase oscillations around the zero-phase in the Oligocene-Miocene have never been reported before. Their statistically-significant correlation with 2.4-Myr eccentricity is enigmatic.

That being said, we took the reviewer's critique about the immature discussion of the conceptual framework very seriously. We **delved into the question about how much ice it would take to generate the ~6 Ma switch** (see also reviewer #2), and we worked on **the clarification of our conceptual explanation for the oscillations around zero phase (35 – 6 Ma)**. Neither of these topics (NH glaciation & Oligocene – Miocene weathering regime shift) are settled in the scientific literature. We believe that, with phase analysis, we are able to shed new light on both topics.

We thank the reviewer's methodological suggestions. We decided that these are best explored in the context of a separate manuscript. We anticipate the megasplice to be used for a variety of follow-up analyses, both by ourselves as by independent research groups. We emphasize that, just as the d18O megasplice, the d13C megasplice is first and foremost **a tool for the paleoclimate & paleoceanography community** that can be used for a wide variety of statistic and stratigraphic purposes.

Further, the reviewer is correct in her/his statement that we only focus on eccentricity rhythms (~100 kyr in Figure 2; 405 kyr in Extended Data Figure 7). This is because the long residence time of carbon in the oceans amplify longer forcing periods in benthic

d13C records. This much complicates the interpretation of leads-and-lags between d18O and d13C on obliquity and precessional timescales.

I would urge a very complete rewrite and also omitting the final “Back to the Future” section, which is simply not compelling.

We significantly **edited the manuscript** to better situate the 6 Ma switch within the existing literature on the start of bipolar glaciation. In addition, we worked hard to comprehensively formulate the conceptual framework that explains the observed oscillations around the zero phase. As suggested by the reviewer, we **omitted the final “Back to the Future” section**.

Reviewer #3, Comment 1

Abstract:

Wouldn't it be simpler to say that the “2.4-Myr eccentricity modulates the phase relationship between...”- the use of “in-phase” is hard to understand because it stands for one phasing (e.g. zero phase lag), not a modulated phase.

Done

Reviewer #3, Comment 2

The statement “We hypothesize that Arctic glaciation and the emergence of bipolar ice sheets enabled eccentricity to exert a major influence on the size of continental carbon reservoirs.” Is problematic on 2 fronts:

First, (and this comment will continue through the manuscript), I would advise the authors not to attribute a direct role to Eccentricity. It is possible that a direct and important link between Eccentricity and climate will emerge, but no one has identified it yet based on any physical model of climate. What is more likely is that behaviors (frequency spectra, phasing) with “~Eccentricity” wavelengths observed in the past arise through non-linearities in response to the precessional cycle.

The reviewer's comment indirectly relates to the well-known **100,000-year problem**: The Earth's climate system and carbon cycle exhibit important 100-kyr rhythms (e.g. Late Pleistocene glacial-interglacial cycles), but there is negligible power in the eccentricity band of insolation time-series. As mentioned by the reviewer, non-linear responses to the precessional cycle are often invoked to explain the observed 100-kyr rhythms. Throughout the manuscript, we call upon the absence of seasonal extremes during eccentricity minima (with minimum precession amplitude) as a (non-linear) key-factor in the growth of the continental biosphere carbon reservoir. Our hypotheses thus consider the 100,000-year problem.

Though, we agree with the reviewer that the quoted sentence from the abstract can be misinterpreted, as if we were claiming a “direct forcing” of carbon reservoirs and/or ice volume by eccentricity.

A second problem is that most of the Pliocene (5.35-2.58 Ma) was above the threshold for Arctic glaciation so I would downplay the interpretation of bipolar ice sheets as a prerequisite for the ~6 Ma phase shift. I'm basing this on abundant Pliocene evidence from the circum-Arctic region, including palynology, the absence of ice rafting, as well as warm temperatures recorded in the North Atlantic and North Pacific, and the generally depleted benthic $\delta^{18}\text{O}$ values, which do not allow for much or any build-up of ice except for very short-lived events-too short lived to account for a general regime shift at 6 Ma.

See also relevant answers to Reviewer #2 Comments 3, 4, 12, 16.

In the revised version of the manuscript, we explain in much more detail **exactly what we mean with “Arctic glaciation”**. We explicitly refer to marine records of diamictites and Ice rafted debris in the North Atlantic and North Pacific of late Miocene and Pliocene age. ODP Site 918 is fundamental in our argument, as it displays a sedimentary sequence of alternating diamictites, ice-rafted debris and marine

sediments without glacial sediments all throughout the Pliocene. It provides support for transient glaciation throughout the late Miocene and Pliocene in the Northern Hemisphere. More specifically, we interpret our results in the context of the Larsen et al. (1994) hypothesis, which states that glaciation began in southeast Greenland in the Late Miocene because of the combination of high precipitation and high topography in the area.

We make the link with the modelling work by Rob DeConto and colleagues (2008), who show the appearance of small ice sheets in good agreement with the description / interpretation by Larsen et al. (1994) at 1.5 x PAL CO₂.

We also discuss the **Pliocene warming** that is explicitly mentioned by reviewer #3, by looking at the SST reconstructions of Herbert et al. (2016). The modest warming of the Pliocene in the NH high latitudes (>50°N) seems **insufficient to reverse the anti-phased dynamic that we observe worldwide since the late Miocene**.

Reviewer #3, Comment 3

A general problem is that “climate-carbon cycle interaction” is left very undefined. It would be nice if the authors can state some specific mechanisms (beyond simply invoking weathering changes, for example), as well as which way they would drive coupling of d18O and d13C observed. Granted, the authors do not have a full model and it’s very hard to uniquely interpret the d13C signal, but I think they should suggest some isotopic scenarios that could be modeled, rather than simply presenting statistical associations.

It is true that these two elements were missing from the original manuscript. We give two examples of well-known “**isotopic scenarios**” (**PETM and coal formation**) and discuss their consequences for the d13C composition of the deep-ocean C reservoir in the first paragraph of the revised manuscript. We also discuss the intense coupling of climate (temperature and precipitation) and carbon cycle (respiration, productivity, solubility).

Reviewer #3, Comment 4

Introduction:

It would be good to remind the reader of what the different components that could influence the d13C composition of the ocean are: changes in the terrestrial biosphere (less = negative signal to the ocean, citations to examples...?), burial of organic matter in the ocean (more = negative), ocean-atmosphere partitioning of CO₂, long-term changes in CO₂ sources and sinks to the atmosphere, etc. The 13C isotope system is way more complex than for d18O, which is why it’s harder to understand.

See answer to Reviewer #3, comment #3. We give two examples of well-known “isotopic scenarios” (PETM and coal formation) and discuss their consequences for the d13C composition of the deep-ocean C reservoir in the first paragraph of the revised manuscript.

Reviewer #3, Comment 5

I'd like the authors to provide the rationale for constructing the megasplice from single sites. I believe they do this to choose optimal records from a sampling/stratigraphic point of view but the reader may not understand this. And, unlike the case for d18O, we expect significant glacial-interglacial shifts in d13C at single sites that might reflect basin-basin fractionation rather than global DIC isotopic changes. We know that for the Pleistocene at least, the 13C gradients between the Atlantic and Pacific are dynamic. Can the authors satisfy us that, at the orbital scale, the isotopic changes they document are in fact global?

See also answer to Reviewer #2, comment 13.

We agree with the reviewer that it is fundamental to acknowledge that we start from the assumption that the d13C isotopic changes that we are documenting with the megasplice are global. The assumption of "**a global d13C heartbeat**" is now explicitly stated in the second paragraph of the manuscript.

We point the reader to Extended Data Figure 1 to illustrate the validity of this assumption. Indeed, this figure shows a consistent astronomical signature between the Pacific and the Atlantic for the Pleistocene. The fact that the assumption seems valid for the Pleistocene is of high relevance, as this is the Epoch of the Cenozoic with the highest ocean heterogeneity.

Reviewer #3, Comment 6

I would also think some comments are in order on age dating and sample resolution. I have a lot of confidence in this group, but a summary statement of how sample data were aligned to the geological time scale would be useful.

We added to the methods section: "*We first scrutinized and then adopted all of those original age-depth models, except for Site U1146. At that site, the published age-depth model is not compatible with the pre- and succeeding records.*"

Sampling resolution is explicitly stated in the methods chapter and the full megasplice dataset will of course be available on **PANGAEA**. The R code for megasplice construction and statistical analysis will be made available through **GitHub**.

We also want to emphasize the information on megasplice construction that is compiled in **Extended Data Table 1**. **This table provides the doi of the original datasets (PANGAEA or NOAA), contains the splicing points and lists the source of the age-depth model applied in the megasplice.** The latter is extremely significant as the chronostratigraphy of some legacy sites (e.g. 982 and 926) has been updated in recent publications. These most recent age-depth alignments are explicitly listed in Extended Data Table 1.

Reviewer #3, Comment 7

"Hence, the global climate and global carbon cycle response to astronomical forcing, and specifically the response to eccentricity strongly differ from that point onwards." As alluded to above, it is very unlikely that the climate and carbon cycle respond linearly and/or directly to eccentricity. "405 kyr forcing " is also a dangerous phrase in the same light, as "forcing" implies a direct cause-effect.

We rephrased the discussion of the d13C wavelet spectrogram, in response to this comment and the next.

Reviewer #3, Comment 8

The statement “This divergence in $\delta^{13}\text{C}_{\text{benthic}}$ and $\delta^{18}\text{O}_{\text{benthic}}$ response to 405-kyr eccentricity forcing has been attributed to a weakening in the coupling between cryosphere and carbon cycle from the mid-Miocene onwards, most likely in connection to the expansion of Antarctica to its near-modern extent”

Seems very naïve and neglects the complexity of what $d^{13}\text{C}$ actually means. We know from the late Pleistocene that the cryosphere and atmospheric CO_2 levels are EXTRAORDINARILY coupled. The point is that the $d^{13}\text{C}$ proxy represents multiple aspects of the carbon cycle, and not CO_2 level itself. I also think that a number of high-resolution studies, such as that of Holbourn and colleagues, would dispute the lack of $d^{18}\text{O}/d^{13}\text{C}$ coupling after the middle Miocene.

We admit that -in the previous version of the manuscript- we could have done a better job at describing another long-standing problem in paleoclimatology: “*Why does the 405-kyr imprint in $d^{18}\text{O}$ disappear after 14 Ma (but not the 100-kyr imprint!), whereas the eccentricity imprint remains strong in $d^{13}\text{C}$?*” We **rephrased the discussion of the $d^{13}\text{C}$ wavelet spectrogram**, considering the reviewer’s comments 7 and 8 and the literature we cite in this context.

Reviewer #3, Comment 9

I’d like to know more about how the authors do their phase analysis. Since the “100 kyr” cycle is in fact a composite of multiple terms with two main components at ~123 and 97 kyr, one worries about spurious phase assignment. The authors state that they “selected the phase from the frequency with the highest coherency”. This approach may be a bit arbitrary- it will depend to begin with on how high the coherencies are (is there good evidence for phase locking?) and then on the dispersion of coherencies within the 90-135 kyr band).

I would back up and look at how compelling the evidence is for continuous 100 kyr components matching the predicted eccentricity terms. I’ve seen cases where there is power at say 85 kyr or 145 kyr that is pseudo-100 kyr, but where there isn’t good evidence of the expected 123 and 97 kyr components when subjected to statistical tests. The phase analysis only makes sense if one has evidence for the underlying eccentricity-related components in each window of the analysis. Figure 1 supports the common presence of ~100 kyr components but more details would help substantiate the use of the 90-135 kyr band.

We added **Extended Data Figure 6 to complement the methods** chapter. In this figure, we illustrate *how* we do our phase analysis, and especially *how* we select the discrete frequency from which we take the phase result.

With her/his comment, Reviewer #3 encouraged us to dig deeper into the question: “*Is there good $d^{18}\text{O}$ - $d^{13}\text{C}$ coherence in the 100-kyr band all throughout the megasplite?*”. In addition to the wavelet spectrum of the $d^{13}\text{C}$ megasplite in Figure 1, we now also provide a visual sense for the $d^{18}\text{O}$ – $d^{13}\text{C}$ coherence in each analysis window, by making **symbol sizes bigger for phase-results with higher coherence** (Figure 2). This new Figure 2 shows high-coherence results (>0.6) all throughout the megasplite.

We then explored *to what extent* the correlation analyses between the oscillations around zero phase and 2.4 Myr eccentricity are impacted when low-coherence results are disregarded. The new Figure 3 (only considering phase results with coherence > 0.3) is similar to the original one, hence this figure is still underpinning our paleoclimate interpretations and conclusions. Admittedly, the choice for a cut-off at 0.3 coherence is rather arbitrary. Nevertheless, that specific cut-off level roughly corresponds with a confidence level of ~70% of non-zero coherence. The second arbitrary milestone, i.e. coherence = 0.6, corresponds to a confidence level of ~97.5% of non-zero coherence.

One can thus have rather good confidence that there effectively is a dependency between the $\delta^{13}\text{C}$ and $\delta^{18}\text{O}$ signals in the frequency band of 100-kyr eccentricity.

The reviewer seems to argue for a wider frequency range for conducting our phase analysis. However, **our arguments for choosing exactly the 90 – 135 kyr band are motivated in Extended Data Figure 5 and Extended Data Table 3**: Too narrow frequency ranges make that some eccentricity-related variability is not considered for phase analysis. Too wide frequency ranges allow for non-eccentricity-related variability to be considered during phase analysis. Extended Data Table 3 comprises a sensitivity analysis of correlation as a function of the frequency range for phase analysis. From this table, it becomes clear that widening the frequency range (as suggested by the reviewer) would result in a drop of correlation between long-term eccentricity, on the one hand, and the oscillations around the zero-phase, on the other hand. We interpret this drop in correlation as an indication of too much non-eccentricity-related variability interfering with the phase analysis. To conclude, we use the 90 – 135 kyr frequency window to calculate phase relationships between $\delta^{13}\text{C}$ and $\delta^{18}\text{O}$ on time-scales of ~100-kyr eccentricity because this frequency window includes the four main ~100-kyr terms [(g4-g5) with 94.9 kyr period, (g3-g5) with 98.9 kyr period, (g4-g2) with 123.9 kyr period, (g3-g2) with 130.7 kyr period] and accommodates minor (<5%) age-model inaccuracies.

Reviewer #3, Comment 10

A new result relative to the Kirtland Turner work is the look at oscillations around zero phase and the idea that these oscillations tie to the envelope of the long eccentricity cycles. I was glad to see that the authors do provide a statistical test to back up their proposed correlation.

Thank you. As mentioned in the previous answer, we now test the proposed correlation only with phase-estimates that correspond to coherence > 0.3. Hence, the slightly different numbers in comparison to the previous version of the manuscript.

I found that the section proposing a conceptual explanation for lead-lag relationships was hard to follow. In particular:

“But the response time of the climate system to eccentricity forcing increases when continental weathering becomes more important within the global carbon cycle. In such situations, global climate variability is strongly driven by the weathering CO_2 sink, in addition to direct Milankovitch forcing and other climate forcing factors³⁴. We therefore hypothesize that the carbon cycle expands its driving influence on climate during weathering maxima, thereby allowing $\delta^{13}\text{C}$ to lead $\delta^{18}\text{O}$.”

One would think that there would be first order relationships therefore between: $\delta^{18}\text{O}$ (a rough proxy for the temperature-weathering effect) and $\delta^{13}\text{C}$ (a recorder of the weathering cycle?). But none is proposed- we only get a set of assertions that do not link to the observables. In particular, the logic of how weathering fluxes would shift the marine $\delta^{13}\text{C}$ inventory is not made explicit. One can try to infer it from the Figure 3 caption, but I find it opaque. It's also hard to understand the reasoning behind linking the response time of the climate system to eccentricity and to weathering. A mechanism would need to be asymmetrical (non-linear) to generate long-term modulations of any importance, since the eccentricity cycles return to a similar baseline ~100 kyr (I realize they do not strictly return to the same baseline, but close to it).

We paid much attention to a better and clearer formulation of the **conceptual framework that could explain the observed $\delta^{13}\text{C}$ – $\delta^{18}\text{O}$ phase relationships and their modulation by long-term eccentricity**. This paragraph is now much expanded, and better connected to the Figure 3 caption and to the cartoon illustrations on that figure. Importantly, we now also explicitly mention the Himalayan orogen as a key

factor in the change from a zonal climate organization (Interval I) to the more monsoon dominated (north – south organized) climate system of Interval II.

Reviewer #3, Comment 11

As to what would drive changes in weathering intensity, it's not clear that the authors have a hypothesis. If weathering is to lead $d_{18}O$, then there needs to be a process to initiate it separate from the cryosphere. Figure 3 suggests that the mechanism lies in the hydrological cycle. If so, aren't there implications that one would see signals at precession, which controls low latitude hydrology? These signals might not be resolved well, but they should be looked for.

We are now more explicit on exactly **which boundary conditions are different between Interval I and Interval II**, and what impact that has on the monsoonal circulation (hence, the hydrological cycle), and ultimately on the weathering CO_2 sink.

Reviewer #3, Comment 12

I'm not sure I would characterize the interval from 26-6 Ma as one climatic state (Termed Interval II here). In fact, it encompasses significant late Oligocene/early Miocene ice ages, the partial deglaciation of the Antarctic (~17.5-14 Ma) and then the refrigeration of the Antarctic. Given these large climatic swings, the absence of phase changes is all the more enigmatic, and suggests that the authors interpretations are simplistic.

We were certainly not claiming that no significant climate changes occurred during Interval II. However, in the light of our phase analyses, we do not observe fundamental changes in the climate – carbon cycle response to orbital forcing. Not in Figure 1A (wavelet spectrum) nor in Figure 3 (oscillations around the zero-phase). Nevertheless, in the detailed description of the Interval II conceptual framework, we now **acknowledge the long duration and marked climate variability of this interval**.

Reviewer #3, Comment 13

I'm puzzled by the beginning of the "Back to the Future" section:

"Our analysis of leads and lags between the climate and the carbon cycle constitutes the first evidence for its regulation by continental ice volume and eccentricity." The authors must not mean what they wrote- there is a huge literature on both topics. I think they didn't clearly articulate what they intend here. Furthermore, their previous analysis has not made clear that ice volume "regulates" carbon, in fact they contended the opposite in some cases.

The intent of the cited sentence was to emphasize the "previously-unobserved" climate – carbon cycle interactions (oscillations around zero-phase) and to stress the proposed link between ice volume and the phase-switch. We recognize that this was not appropriately formulated in the previous version of the manuscript.

We **completely rewrote the concluding paragraph**, as suggested by the reviewer. We step away from the "warning message", that climate-carbon cycle interactions might change in the Anthropocene. Instead, we end the paper by stating that our work presents mere "qualitative hypotheses", which need to be quantified and tested in further research. Nevertheless, the results of this work represent an important step towards a well-constrained Earth System Sensitivity (ESS).

Reviewer #3, Comment 14

Extended Figure 1: change to a light colored from black font for site identification- very difficult to read

Extended Data Figure 1 has been changed, now to also include a Miocene example of the basin-correction. The color coding adopted in this figure follows the **color coding that has been consistently adopted throughout the paper**.

REVIEWER COMMENTS

Reviewer #2 (Remarks to the Author):

The authors have gone a long way to addressing the reviewers comments.

I am satisfied with their answers to my questions about the Megasplice construction and, thus, and satisfied that it is a solid archive on its own. This is well explained by the additional text and new extended figures.

The distinction compared to Turner et al., 2014 is also clearer now.

This amazingly well put together data compilation has a lot to offer and I think the authors are digging deep in this first interrogation of it and have justified this well in their response to other reviewers comments, also, which parts they are not dealing with here and why.

To me the question of when did NH glaciation begin to have a global carbon system-scale signal is massive, it is unresolved and paradigm shifting, and thus this contribution provides a novel and important perspective.

However, NH ice is also the part that I am still not 100% happy with, i.e. the framework of evidence presented that NH ice of sufficient size existed by 6 Ma to have the effects inferred. There are some key papers that the authors should refer to in this respect, that on one side I think will help their argument (Beirman et al., 2016), but other's they also need to reflect on. Refer to O'Regan et al., 2007 presents an important review of Arctic climate history evolution in this respect.

While there is evidence for perennial sea ice cover some times in the Arctic since 13 Ma, although potentially with ice free summers in the late Miocene (Stein et al., 2016) and Pliocene (Clotten et al., 2019), and ice rafting off Greenland (St. John work), also see Bierman et al., 2016, there is also extensive paleobotanical evidence for Arctic forests until the Pliocene.

For example, according to Matthews et al. (2019), macrofossils imply an 'Arctic tree line' extending up to 80°N (compared to 70°N today) into the middle Pliocene of northern Canada (Meighan Island) with the first modern-like Arctic tundra and forest-tundra occurring at ca. 3 Ma. Tundra appeared later in Beringia, which sits at ca. 70°N, probably in parallel with Pliocene/Pleistocene glacial intensification (Volkova, 2011; Matthews et al., 2019).

How does this match with considerable northern ice?

While CO₂ modelling of thresholds for ice sheet inception, to compare with coarse CO₂ proxy data, and ice rafting (could be alpine glacials) provide interesting theoretical constrains (which are model dependent) the floral constraints cannot be ignored.

Can you defend that the change in orbital phasing is not connected to changes in vegetation biomes in the northern high latitudes from 6 Ma? E.g. a significant slowing in forest growth as you switch to boreal forest, or development of tundra and permafrost. What scale/sign of δ¹³C signal might one expect if one of these reservoirs started to play a bigger role in from 6 Ma?

References

Bierman, P. R., Shakun, J. D., Corbett, L. B., Zimmerman, S. R., and Rood, D. H., 2016, A persistent and dynamic East Greenland Ice Sheet over the past 7.5 million years: *Nature*, v. 540, no. 7632, p. 256-260.

Matthews, J. V., Jr., Telka, A., and Kuzmina, S. A., 2019, Late neogene insect and other invertebrate fossils from alaska and arctic/subarctic Canada: *Invertebrate Zoology*, v. 16, no. 2, p. 126-153.

O'Regan, A. M., Williams, C. J., Frey, K. E., and Jakobsson, M., 2011, A Synthesis of the long-term paleoclimatic evolution of the Arctic: *Oceanography*, The changing Arctic Ocean, Special Issue on the International Polar Year (2007–2009), v. 24, no. 3, p. 66-80.

Volkova, V. S., 2011, Paleogene and Neogene stratigraphy and paleotemperature trend of West Siberia (from palynologic data): *Russian Geology and Geophysics*, v. 52, no. 7, p. 709-716.

Reviewer #3 (Remarks to the Author):

I appreciate that the authors have made a number of substantial changes in the manuscript to document the underlying splice construction (now with more extensive and useful material in the Supplement) and made their statistical methods read more clearly. The paper also develops potential scenarios to explain the phase shift change and relate it to a changing response to eccentricity-related climate changes (lines 144-239). The authors should be congratulated for pulling an important data set together and making intriguing observations.

My critique is that a presentation of such an important data set as the $\delta^{13}C$ megasplice should stand the test of time- not in being "right", but in accurately telling the reader what seems clear and what remains mysterious. The title of the paper represents a step away from the mark: "Ice volume and rock weathering determine climate – carbon cycle feedbacks on the rhythms of eccentricity" This would require that the authors a) demonstrate causality and b) substantiate that the major features that emerge from their analysis do indeed represent documented changes in ice volume and rock weathering. Instead, the title is their hypothesis, which can be disputed at present and does not uniquely fall out of the data analysis. If the hypotheses was compellingly sustained, this title would work, but I don't think that's the case. Note that the central role of NH ice is repeated explicitly on lines 176-179:

146). We relate this

147 change in response to the advent of continental ice in the northern hemisphere (NH) and the
148 resulting areal competition between ice and high-latitude continental carbon reservoirs (peat-
and wetlands, boreal forests).

Causality: it is quite possible that ice volume is the tail, not the head, of the dog, and that carbon cycle feedbacks drive ice volume. As for rock weathering, it too responds to climate and CO₂ forcing so that a chain of causation is not clear.

Supporting the hypothesis. The authors continue to see 6 Ma as a milestone in ice volume evolution. However the existing evidence (ice rafting, temperature, and $\delta^{18}O$) show that the late Miocene glaciations were transient, and did not involve a large permanent increase in ice volume. The early Pliocene was warmer, less glaciated (as supported by $\delta^{18}O$) and did not have ice rafting in the North Atlantic. Their phase shift at 6 Ma is indeed a significant finding, but to associate it with a permanent ice volume regime shift is simply not supported. Yet the authors continue this claim in the abstract

14 We hypothesize that

15 Arctic glaciation and the emergence of bipolar ice sheets created boundary conditions

16 under which continental ice sheets started to compete with the continental biosphere
for domains to expand.

Indeed if there was a compelling time to claim an ice-volume to phase relationship, one would have thought it would be at the end of the mid-Miocene Climatic Optimum, which saw a far larger increase in ice volume than anything at 6 Ma. This does emerge in the drop-out of 405 kyr cycling in $\delta^{18}O$ in the text, but the almost exclusive focus on the 6 Ma shift short changes an interesting observation in the paper.

Likewise, I do not see any proxy support for a clear role for rock weathering changing at 6 Ma-tracers such as ^{87}Sr etc.

My point isn't to take away from an interpretation of a major change in $\delta^{18}\text{O}$ - $\delta^{13}\text{C}$ relationships at ~6 Ma: rather it's whether the authors have put their finger on its cause. It's OK sometimes not to be too sure...

A next point is that the paper can help the reader better appreciate the complexity of the C isotope reservoir of the ocean. The authors have added material (lines 32 to 45) but they choose to focus on extreme sudden events such as the PETM, or coal formation, which are not the kind of processes they track with this paper. Instead, the authors should discuss the levers more appropriate to ~100 kyr and 2.4 Myr timescales central to this paper. On this timescale, we have a) variations in terrestrial biomass b) ocean-atmosphere partitioning c) weathering inputs of DIC and d) Corg burial (and probably a few more). The important point would be that the $\delta^{13}\text{C}$ signal represents a composite of many different processes. Perhaps that's why it can change its phase behaviour...

Can the authors build on their observations of the typical phase lag of $\delta^{13}\text{C}$ relative to $\delta^{18}\text{O}$ (lines 250-254):

149 This small but noticeable phase lag

150 of $\delta^{13}\text{C}$ has been observed before^{9,38,39} and explained by the long residence time of carbon in

151 the ocean⁴⁰. The large size of the deep-ocean carbon reservoir causes the response time of
152 the carbon cycle to eccentricity forcing (through the amplitude modulation of precession) to be
153 slightly slower than that of the climate-cryosphere system.

154 255

This explanation is incomplete. If there is a 100 kyr response time of the ocean carbon system, and it responds linearly to forcing, it would lead to a quarter wavelength phase lag relative to a forcing, or 25 kyr. Finding a 1-2 kyr phase lag instead implies that there are much more rapid processes operating, most likely on the precessional timescale (typical phase lags observed for precessional signals relative to orbital forcing lie in the range of 1-2 kyr). So the short phase lag actually supports the underlying precessional role- but the way this section is phrased is not accurate in thinking about the residence time of carbon and a phase lag.

I'm puzzled by the section on "weathering as a potential..." and would like the authors to make their logic more explicit. Can they identify exactly how/why changes in weathering would drive a marine $\delta^{13}\text{C}$ signal? Weathering of silicates by itself produces no $\delta^{13}\text{C}$ signal, and weathering of carbonate will not change the ocean's $\delta^{13}\text{C}$ of DIC much. Destruction of the terrestrial biosphere will affect DIC, but that is not the same as "weathering".

With the various discussion on mechanisms relating $\delta^{13}\text{C}$ to climate, I don't follow why the authors don't more explicitly consider the mechanism that we know dominates the late Pleistocene: an oscillating deep ocean CO_2 reservoir, where storage of metabolic CO_2 lowers the marine $\delta^{13}\text{C}$, and draws down CO_2 . An important question would be whether this Pleistocene behaviour applies to earlier periods of the Neogene.

Specific feedback on the author's responses:

Reviewer #3, Comment 2

The statement "We hypothesize that Arctic glaciation and the emergence of bipolar ice sheets enabled eccentricity to exert a major influence on the size of continental carbon reservoirs." Is problematic on 2 fronts:

First, (and this comment will continue through the manuscript), I would advise the authors not to attribute a direct role to Eccentricity. It is possible that a direct and important link between

Eccentricity and climate will emerge, but no one has identified it yet based on any physical model of climate. What is more likely is that behaviors (frequency spectra, phasing) with ~Eccentricity" wavelengths observed in the past arise through non-linearities in response to the precessional cycle.

The reviewer's comment indirectly relates to the well-known 100,000-year problem: The Earth's climate system and carbon cycle exhibit important 100-kyr rhythms (e.g. Late Pleistocene glacial-interglacial cycles), but there is negligible power in the eccentricity band of insolation time-series. As mentioned by the reviewer, non-linear responses to the precessional cycle are often invoked to explain the observed 100-kyr rhythms. Throughout the manuscript, we call upon the absence of seasonal extremes during eccentricity minima (with minimum precession amplitude) as a (non-linear) key-factor in the growth of the continental biosphere carbon reservoir. Our hypotheses thus consider the 100,000-year problem.

Though, we agree with the reviewer that the quoted sentence from the abstract can be misinterpreted, as if we were claiming a "direct forcing" of carbon reservoirs and/or ice volume by eccentricity.

The authors continue to use the fraught construction

146 Yet, the mechanistic pathways through which eccentricity could have influenced ocean ventilation before and after the phase-switch are not well understood.

Which at least leads the reader to think that there is a direct process that translates eccentricity into 100, 405, and 2400 kyr cycles. The answer is most certainly more complex. Using the phrase "not well understood" does not remedy the implicit claim of a direct eccentricity forcing.

Reviewer #3, Comment 9

I'd like to know more about how the authors do their phase analysis. Since the "100 kyr" cycle is in fact a composite of multiple terms with two main components at ~123 and 97 kyr, one worries about spurious phase assignment. The authors state that they "selected the phase from the frequency with the highest coherency". This approach may be a bit arbitrary- it will depend to begin with on how high the coherencies are (is there good evidence for phase locking?) and then on the dispersion of coherencies within the 90-135 kyr band).

I would back up and look at how compelling the evidence is for continuous 100 kyr components matching the predicted eccentricity terms. I've seen cases where there is power at say 85 kyr or 145 kyr that is pseudo-100 kyr, but where there isn't good evidence of the expected 123 and 97 kyr components when subjected to statistical tests. The phase analysis only makes sense if one has evidence for the underlying eccentricity-related components in each window of the analysis. Figure 1 supports the common presence of ~100 kyr components but more details would help substantiate the use of the 90-135 kyr band.

We added Extended Data Figure 6 to complement the methods chapter. In this figure, we illustrate how we do our phase analysis, and especially how we select the discrete frequency from which we take the phase result.

With her/his comment, Reviewer #3 encouraged us to dig deeper into the question: "Is there good d18O - d13C coherence in the 100-kyr band all throughout the megasplice?". In addition to the wavelet spectrum of the d13C megasplice in Figure 1, we now also provide a visual sense for the d18O - d13C coherence in each analysis window, by making symbol sizes bigger for phase-results with higher coherence (Figure 2). This new Figure 2 shows high-coherence results (>0.6) all throughout the megasplice.

We then explored to what extent the correlation analyses between the oscillations around zero phase and 2.4 Myr eccentricity are impacted when low-coherence results are disregarded. The new Figure 3 (only considering phase results with coherence > 0.3) is similar to the original one, hence this figure is still underpinning our paleoclimate interpretations and conclusions. Admittedly, the choice for a cut-off at 0.3 coherence is rather arbitrary. Nevertheless, that specific cut-off level roughly corresponds with a confidence level of ~70% of non-zero coherence. The second arbitrary

milestone, i.e. coherence = 0.6, corresponds to a confidence level of ~97.5% of non-zero coherence. One can thus have rather good confidence that there effectively is a dependency between the $\delta^{13}\text{C}$ and $\delta^{18}\text{O}$ signals in the frequency band of 100-kyr eccentricity.

The reviewer seems to argue for a wider frequency range for conducting our phase analysis. However, our arguments for choosing exactly the 90 – 135 kyr band are motivated in Extended Data Figure 5 and Extended Data Table 3: Too narrow frequency ranges make that some eccentricity-related variability is not considered for phase analysis. Too wide frequency ranges allow for non-eccentricity-related variability to be considered during phase analysis. Extended Data Table 3 comprises a sensitivity analysis of correlation as a function of the frequency range for phase analysis. From this table, it becomes clear that widening the frequency range (as suggested by the reviewer) would result in a drop of correlation between long-term eccentricity, on the one hand, and the oscillations around the zero-phase, on the other hand. We interpret this drop in correlation as an indication of too much non-eccentricity-related variability interfering with the phase analysis. To conclude, we use the 90 – 135 kyr frequency window to calculate phase relationships between $\delta^{13}\text{C}$ and $\delta^{18}\text{O}$ on time-scales of ~100-kyr eccentricity because this frequency window includes the four main ~100-kyr terms [(g4-g5) with 94.9 kyr period, (g3-g5) with 98.9 kyr period, (g4-g2) with 123.9 kyr period, (g3-g2) with 130.7 kyr period] and accommodates minor (<5%) age-model inaccuracies.

The authors have misconstrued my argument. It is not to widen the frequency window around a central ~100 kyr window. Rather, it to ask them to validate their assumption that ~ 100 kyr variance is phase-locked to an underlying eccentricity pacing. My point about 85 or 145 kyr bands was that I've seen a number of records that suggest 100 kyr cyclicity, but the interpretation falls apart if one looks rigorously at the spectra. The authors need to convince us first that they have a robust fingerprint of the ~123 and 97 kyr components of eccentricity continuously in the time series. Only then can they talk about a phase analysis of this signal in relation to an underlying (known) pacing by orbital eccentricity. I agree that the author's window is appropriate to following the coherence between $\delta^{13}\text{C}$ and $\delta^{18}\text{O}$ on ~100 kyr timescales- the question is whether the assumption that a 100 kyr pacing represents a fundamental coupling can be substantiated from the data analysis.

Tim Herbert

Answers to reviewers.

Reviewer #2

The authors have gone a long way to addressing the reviewers comments. I am satisfied with their answers to my questions about the MegasplICE construction and, thus, and satisfied that it is a solid archive on its own. This is well explained by the additional text and new extended figures. The distinction compared to Turner et al., 2014 is also clearer now.

This amazingly well put together data compilation has a lot to offer and I think the authors are digging deep in this first interrogation of it and have justified this well in their response to other reviewers' comments, also, which parts they are not dealing with here and why.

To me the question of when did NH glaciation begin to have a global carbon system-scale signal is massive, it is unresolved and paradigm shifting, and thus this contribution provides a novel and important perspective.

Thank you. It is of course of utmost importance that the “technical characteristics” of our $\delta^{13}\text{C}$ megasplICE and its analysis come across clearly. We are happy to read that the reviewer deems the megasplICE construction sufficiently well-explained to differentiate itself from the Zachos et al. (2008) compilation and the Lisiecki & Raymo (2005) stack.

Also, we are glad to read that we could make the difference in analytic approach with Turner 2014 clearer.

Reviewer #2, Comment 1.

Alternative hypotheses for the 6 Ma phase-switch.

However, NH ice is also the part that I am still not 100% happy with, i.e. the framework of evidence presented that NH ice of sufficient size existed by 6 Ma to have the effects inferred. There are some key papers that the authors should refer to in this respect, that on one side I think will help their argument (Beirman et al., 2016), but other's they also need to reflect on. Refer to O'Regan et al., 2007 presents an important review of Arctic climate history evolution in this respect.

First of all, we thank the reviewer for pointing us to the Bierman et al., 2016 reference, which was missing in the previous version of our manuscript. The inclusion of this reference indeed helped our argument.

A reference to O'Regan et al. (2007) is not included in our manuscript, as it is somewhat too general and mainly focusses on sea-ice for the late Miocene and early Pliocene, whereas the key to our hypothesis lies with continental ice.

While there is evidence for perennial sea ice cover some times in the Arctic since 13 Ma, although potentially with ice free summers in the late Miocene (Stein et al., 2016) and Pliocene (Clotten et al., 2019), and ice rafting off Greenland (St. John work), also see Bierman et al., 2016, there is also extensive paleobotanical evidence for Arctic forests until the Pliocene.

For example, according to Matthews et al. (2019), macrofossils imply an 'Arctic tree line' extending up to 80°N (compared to 70°N today) into the middle Pliocene of northern Canada (Meighan Island) with the first modern-like Arctic tundra and forest-tundra occurring at ca. 3 Ma. Tundra appeared later in Beringia, which sits at ca. 70°N, probably in parallel with Pliocene/Pleistocene glacial intensification (Volkova, 2011; Matthews et al., 2019).

How does this match with considerable northern ice?

While CO₂ modelling of thresholds for ice sheet inception, to compare with coarse CO₂ proxy data, and ice rafting (could be alpine glacials) provide interesting theoretical constraints (which are model dependent) the floral constraints cannot be ignored.

In the revised version of the manuscript, we now explicitly discuss the available paleobotanical data. In this section, we acknowledge that the boreal realm remained extensive (up to 80°N) throughout the late Miocene and most of the Pliocene. We use the reference of Matthews et al. (2019) suggested by the reviewer, as well as an Earth-Science-Reviews paper on this topic by Pound et al. (2012). We agree with the reviewer that the paleobotanical data provide strong indications for a (much) warmer Arctic compared to today, and we identify this issue as an important “open question”. Yet, we also emphasize that these paleobotanical data are not necessarily at odds with a dynamic East Greenlandic Ice Sheet as early as the Late Miocene, given the large longitudinal climate variability along the Arctic circle.

Can you defend that the change in orbital phasing is not connected to changes in vegetation biomes in the northern high latitudes from 6 Ma? E.g. a significant slowing in forest growth as you switch to boreal forest, or development of tundra and permafrost. What scale/sign of δ¹³C signal might one expect if one of these reservoirs started to play a bigger role in from 6 Ma?

We agree with the reviewer that the observed phase-switch is not the result of a mere replacement of vegetation by ice. Instead, our hypothesis assumes a shift in the areal distribution of high-latitude biomes. To avoid any misunderstanding with the readers, we changed the corresponding statement in the abstract to:

We hypothesize that this transition is consistent with Arctic cooling and the emergence of bipolar ice sheets: Prior to 6 Ma, low-latitude continental carbon reservoirs expanded during astronomically-forced cool spells. After 6 Ma, however, continental carbon reservoirs contract rather than expand during cold periods (i.e. glacials) due to competing effects between terrestrial ice and high-latitude biomes.

In the subsection “The anti-phase hypothesis” we elaborate on this point to avoid confusion.

Yet, the advent of continental ice and substantial cooling in the northern hemisphere (NH) could have initiated the areal competition between ice and high-latitude continental carbon reservoirs (peat- and wetlands, boreal forests, tundra), and intensified the mutual competition among these high-latitude biomes. The latter is relevant because tundra ecosystems, for example, have a significantly smaller carbon storage capacity compared to boreal forests.

Reviewer #3

I appreciate that the authors have made a number of substantial changes in the manuscript to document the underlying splice construction (now with more extensive and useful material in the Supplement) and made their statistical methods read more clearly. The paper also develops potential scenarios to explain the phase shift change and relate it to a changing response to eccentricity-related climate changes (lines 144-239). The authors should be congratulated for pulling an important data set together and making intriguing observations.

Reviewer #3, Comment 1. *It's OK sometimes not to be too sure.*

My critique is that a presentation of such an important data set as the d13C megasplice should stand the test of time- not in being “right”, but in accurately telling the reader what seems clear and what remains mysterious. The title of the paper represents a step away from the mark: “Ice volume and rock weathering determine climate – carbon cycle feedbacks on the rhythms of eccentricity” This would require that the authors a) demonstrate causality and b) substantiate that the major features that emerge from their analysis do indeed represent documented changes in ice volume and rock weathering. Instead, the title is their hypothesis, which can be disputed at present and does not uniquely fall out of the data analysis. If the hypothesis was compellingly sustained, this title would work, but I don't think that's the case.

The first version of the manuscript submitted to Nature Communications carried a more general title that didn't refer to our main hypotheses / conclusions. This was criticized by Reviewer #2 for being “*Too general and rather flat*” in the previous round of review. In response to this critique, we changed the title to focus on our interpretations and conclusions. We prefer the new title as it reflects our interpretation of the observed patterns in terms of ice volume and continental weathering.

We did however consider the underlying message of this comment, which is that we cannot be 100% sure to touch upon the “ultimate causes” of the observed phase switches (→ *it is quite possible that ice volume is the tail, not the head, of the dog*). It goes without saying that this is a fair point raised by the reviewer. Yet, our interpretations are based on a combination of extensive literature review and a careful data analysis of the d13C & d18O megasplices. For the latter reason, we would like to keep our main interpretations in the title. As a compromise between these two viewpoints, we replaced the verb “determine” by “mediate” in the title. The verb “mediating” does not necessarily imply that ice volume and weathering are at the very top of the causation chain, regulating Earth system response to astronomical forcing. The verb does however assign an important role for ice & weathering, as highlighted in our paper and based on the interpretation of our proxy-series analyses. Yet, “mediating” does not claim them to be the “ultimate causes”.

That being said, we remain open for editorial suggestions as to how to change the title of our manuscript.

Note that the central role of NH ice is repeated explicitly on lines 176-179:

We relate this change in response to the advent of continental ice in the northern hemisphere (NH) and the resulting areal competition between ice and high-latitude continental carbon reservoirs (peat- and wetlands, boreal forests).

Causality: **it is quite possible that ice volume is the tail, not the head, of the dog**, and that carbon cycle feedbacks drive ice volume. As for rock weathering, it too responds to climate and CO2 forcing so that a chain of causation is not clear.

Supporting the hypothesis. The authors continue to see 6 Ma as a milestone in ice volume evolution. However the existing evidence (ice rafting, temperature, and d18O) show that the late Miocene glaciations were transient, and did not involve a large permanent increase in ice volume. The early Pliocene was warmer, less glaciated (as supported by d18O) and did not have ice rafting in the North Atlantic. Their phase shift at 6 Ma is indeed a significant finding, but to associate it with a permanent ice volume regime shift is simply not supported. Yet the authors continue this claim in the abstract:

We hypothesize that Arctic glaciation and the emergence of bipolar ice sheets created boundary conditions under which continental ice sheets started to compete with the continental biosphere for domains to expand.

A similar comment was also raised by Reviewer #2 (Comment 1).

We nuanced this claim in the abstract to emphasize that the observed phase-switch is not the result of *just* the replacement of vegetation by ice. Instead, our hypothesis assumes a shift in the areal distribution of high-latitude biomes (terrestrial ice cover, tundra, boreal forest, peat lands, wet lands). To avoid misunderstanding, we changed this sentence to:

We hypothesize that this transition is consistent with Arctic cooling and the emergence of bipolar ice sheets: Prior to 6 Ma, low-latitude continental carbon reservoirs expanded during astronomically-forced cool spells. After 6 Ma, however, continental carbon reservoirs contract rather than expand during cold periods (i.e. glacials) due to competing effects between terrestrial ice and high-latitude biomes.

In the subsection “The anti-phase hypothesis” we also somewhat elaborated this point to avoid confusion.

Yet, the advent of continental ice and substantial cooling in the northern hemisphere (NH) could have initiated the areal competition between ice and high-latitude continental carbon reservoirs (peat- and wetlands, boreal forests, tundra), and intensified the mutual competition among these high-latitude biomes. The latter is relevant because tundra ecosystems, for example, have a significantly smaller carbon storage capacity compared to boreal forests.

Indeed if there was a compelling time to claim an ice-volume to phase relationship, one would have thought it would be at the end of the mid-Miocene Climatic Optimum, which saw a far larger increase in ice volume than anything at 6 Ma.

The Middle Miocene Climate Transition is indeed a time of large ice volume increase on Antarctica. However, this body of ice rapidly grew to a rather undynamic ice sheet. In De Vleeschouwer et al. (2017), we described this as “*Antarctica grew too big to pulse at the beat of precession*”. Such an undynamic ice sheet is not able to trigger significant changes in the areal distribution between ice cover and biomes, limited by the geography of the Antarctic continent. For that reason, it is not surprising that this large increase in ice volume did not trigger a phase-switch. The ice volume increase at ~6 Ma in the Arctic, on the other hand, marks the start of very dynamic Arctic ecosystems, with significant changes in areal distribution between (ephemeral or not, that remains an open question) ice cover and high-latitude biomes.

This does emerge in the drop-out of 405 kyr cycling in d18O in the text, but the almost exclusive focus on the 6 Ma shift short changes an interesting observation in the paper.

The drop-out of 405-kyr power in $\delta^{18}\text{O}$ is indeed intriguing and still unexplained. This problem is clearly identified as an “open question” in the manuscript. We do not focus on the problem because our $\delta^{13}\text{C}$ – $\delta^{18}\text{O}$ phase analyses do not provide any clues as to what might be the cause of this drop-out. The in-phase behaviour between $\delta^{13}\text{C}$ and $\delta^{18}\text{O}$ simply continues across the 405-kyr drop-out at ~14 Ma. This means that we do not have a good, feasible hypothesis for this event. Instead, we focus on our 6 and 26 Ma hypotheses, for which we have clear and unambiguous phase-analysis results.

Likewise, I do not see any proxy support for a clear role for rock weathering changing at 6 Ma-tracers such as ^{87}Sr etc.

We are not claiming a clear role for rock weathering at 6 Ma. Our ~6 Ma hypothesis chiefly pertains to the high-latitude continental biosphere.

A shift in the locus of rock weathering from the high- to the low-latitudes is claimed for 26 Ma. The phase change described for 26 Ma occurs just after the second period of rapid $\delta^{17}\text{Li}_{\text{sw}}$ increase, as described by Misra and Froehlich (2012, Science), which they associated with the Himalayan Orogen.

My point isn't to take away from an interpretation of a major change in $\delta^{18}\text{O}$ - $\delta^{13}\text{C}$ relationships at ~6 Ma: rather it's **whether the authors have put their finger on its cause**. It's OK sometimes not to be too sure...

During the revision of our manuscript, we paid special attention at making a clear distinction between what conclusions can be drawn unambiguously from the $\delta^{13}\text{C}$ - $\delta^{18}\text{O}$ phase shifts (basically, that there has been regime shifts in climate – carbon cycle interactions), and which interpretations are made with the help of additional lines of evidence from the literature. We clearly identify open questions (405-kyr problem, extensive boreal forests during the Pliocene, the role of ocean ventilation, a detailed history of Himalayan orogen) and we consider alternative plausible hypotheses (e.g. ocean ventilation).

We are happy to read that both reviewers are confident that the $\delta^{13}\text{C}$ megasplice, as an archive, will stand the test of time and will rapidly develop into a reference curve for many paleoclimate and paleoceanographic studies. By implementing their suggested changes and words of caution, we feel comfortable that our concepts for climate-carbon cycle interactions will be relevant for years to come, trigger multiple follow-up studies, and thus will also stand the test of time.

Reviewer #3, Comment 2. *Carbon cycle complexity*

A next point is that the paper can help the reader better appreciate the complexity of the C isotope reservoir of the ocean. The authors have added material (lines 32 to 45) but they choose to focus on extreme sudden events such as the PETM, or coal formation, which are not the kind of processes they track with this paper. Instead, the authors should discuss the levers more appropriate to ~100 kyr and 2.4 Myr timescales central to this paper. On this timescale, we have a) variations in terrestrial biomass b) ocean-atmosphere partitioning c) weathering inputs of DIC and d) Corg burial (and probably a few more). The important point would be that the $\delta^{13}\text{C}$ signal represents a composite of many different processes. Perhaps that's why it can change its phase behaviour...

An important point to make, indeed. We now provide a comprehensive overview of factors that influence the $\delta^{13}\text{C}$ of DIC of seawater (adding geological reservoirs, photosynthesis & respiration, ocean circulation).

Reviewer #3, Comment 3.

Phase lag

Can the authors build on their observations of the typical phase lag of $\delta^{13}\text{C}$ relative to $\delta^{18}\text{O}$ (lines 250-254):

This small but noticeable phase lag of $\delta^{13}\text{C}$ has been observed before and explained by the long residence time of carbon in the ocean. The large size of the deep-ocean carbon reservoir causes the response time of the carbon cycle to eccentricity forcing (through the amplitude modulation of precession) to be slightly slower than that of the climate-cryosphere system.

This explanation is incomplete. If there is a 100 kyr response time of the ocean carbon system, and it responds linearly to forcing, it would lead to a quarter wavelength phase lag relative to a forcing, or 25 kyr. Finding a 1-2 kyr phase lag instead implies that there are much more rapid processes operating, most likely on the precessional timescale (typical phase lags observed for precessional signals relative to orbital forcing lie in the range of 1-2 kyr). So the short phase lag actually supports the underlying precessional role- but the way this section is phrased is not accurate in thinking about the residence time of carbon and a phase lag.

First of all, we fully acknowledge that our statement is a simplification. That being said, we want to emphasize that this phase lag is the result of a **non-linear** response to eccentricity-modulated precession.

To illustrate this point, we first quote Pälike et al. (2006): *Phase estimates between astronomy and data determine a ~20 ky lag of $\delta^{13}\text{C}$ compared to $\delta^{18}\text{O}$ for long eccentricity cycles (405 ky), with close to zero lags for other astronomical terms. Phase estimates between $\delta^{13}\text{C}$ compared to $\delta^{18}\text{O}$ suggest that $\delta^{13}\text{C}$ shows increased lag times for correspondingly longer periods, a behavior that we are able to model as part of this study. A similar pattern has been recognized previously in late Oligocene records. This observation is compatible with the long residence time of carbon in the oceans [~0.1 My] that transfers energy from climatic precession into eccentricity bands through a nonlinear process, resulting in a frequency- dependent phase lag of $\delta^{13}\text{C}$.*

Second, we provide a simple “back of the envelope” example. The blue curve provides a hypothetical non-linear response to eccentricity-modulated precession. Note that we only put in a 100-kyr amplitude modulation (no 405-kyr modulation). The red curve provides the hypothetical response of the deep-sea carbon reservoir, calculated with a reservoir size of 39,000 GtC and a carbon flux of 0.6 GtC/year. The red line has most power in the 100-kyr band (because of the non-linear response in combination with the long residence time), and exhibits a phase lag with the forcing of a few thousand years.

We rephrased the explanation to emphasize the double requirement for the observed phase-lag on eccentricity timescales: a non-linear response to precession + the long residence time of C in the ocean.

Quote above from Pälike et al., The Heartbeat of the Oligocene Climate System, *Science* **Vol. 314**, Issue 5807, pp. 1894-1898 (2016). Reprinted with permission from AAAS.

Reviewer #3, Comment 4.

The weathering hypothesis

I'm puzzled by the section on "weathering as a potential..." and would like the authors to make their logic more explicit. Can they identify exactly how/why changes in weathering would drive a marine $\delta^{13}\text{C}$ signal? Weathering of silicates by itself produces no $\delta^{13}\text{C}$ signal, and weathering of carbonate will not change the ocean's $\delta^{13}\text{C}$ of DIC much. Destruction of the terrestrial biosphere will affect DIC, but that is not the same as "weathering".

We are not claiming that weathering drives the $\delta^{13}\text{C}$ signal directly. We are claiming that when a non-temperature-determined factor enhances rock weathering (e.g. tectonic uplift), an incited carbon cycle expands its control on climate and therewith reduces its time-lag relative to the climate system on orbital time-scales: Hence periodic $\delta^{13}\text{C}_{\text{benthic}}$ (carbon cycle) leads relative to $\delta^{18}\text{O}_{\text{benthic}}$ (climate). We made this explicitly clear in the appropriate section.

Reviewer #3, Comment 4.

Ocean ventilation as an alternative hypothesis

With the various discussion on mechanisms relating $\delta^{13}\text{C}$ to climate, I don't follow why the authors don't more explicitly consider the mechanism that we know dominates the late Pleistocene: an oscillating deep ocean CO_2 reservoir, where storage of metabolic CO_2 lowers the marine $\delta^{13}\text{C}$, and draws down CO_2 . An important question would be whether this Pleistocene behaviour applies to earlier periods of the Neogene.

Ocean ventilation is now discussed in more detail as an alternative hypothesis and open question for the phase-switch at 6 Ma. Importantly, according to research by Ann Holbourn et al. (2017), ocean ventilation did flip its response to astronomical forcing between the Middle Miocene and the Pleistocene. According to these authors, colder periods during the Miocene correspond to a more vigorous ocean circulation. Whereas during the Pleistocene, colder periods are characterized by less ocean ventilation. However, which mechanisms were decisive for reversing the response of ocean ventilation to astronomical forcing. That, in combination with the argument that the isotopic signal of continental biomass is likely to be significantly larger than that of ocean ventilation makes that we stay with our preferred continent-based hypothesis.

Specific feedback on the author's responses.

Reviewer #3, Comment 5.

The 100-kyr problem

Reviewer #3, Comment 2 and our answer from previous review round.

The statement "We hypothesize that Arctic glaciation and the emergence of bipolar ice sheets enabled eccentricity to exert a major influence on the size of continental carbon reservoirs." Is problematic on 2 fronts:

First, (and this comment will continue through the manuscript), I would advise the authors not to attribute a direct role to Eccentricity. It is possible that a direct and important link between Eccentricity and climate will emerge, but no one has identified it yet based on any physical model of climate. What is more likely is that behaviors (frequency spectra, phasing) with ~Eccentricity" wavelengths observed in the past arise through non-linearities in response to the precessional cycle.

Answer:

The reviewer's comment indirectly relates to the well-known 100,000-year problem: The Earth's climate system and carbon cycle exhibit important 100-kyr rhythms (e.g. Late Pleistocene glacial-interglacial cycles), but there is negligible power in the eccentricity band of insolation time-series. As mentioned by the reviewer, non-linear responses to the precessional cycle are often invoked to explain the observed 100-kyr rhythms. Throughout the manuscript, we call upon the absence of seasonal extremes during eccentricity minima (with minimum precession amplitude) as a (non-linear) key-factor in the growth of the continental biosphere carbon reservoir. Our hypotheses thus consider the 100,000-year problem. Though, we agree with the reviewer that the quoted sentence from the abstract can be misinterpreted, as if we were claiming a "direct forcing" of carbon reservoirs and/or ice volume by eccentricity.

The authors continue to use the fraught construction:

Yet, the mechanistic pathways through which eccentricity could have influenced ocean ventilation before and after the phase-switch are not well understood.

Which at least leads the reader to think that there is a direct process that translates eccentricity into 100, 405, and 2400 kyr cycles. The answer is most certainly more complex. Using the phrase "not well understood" does not remedy the implicit claim of a direct eccentricity forcing.

We agree that this sentence could be misleading, and could make some of our readers believe we are advocating a direct linear eccentricity forcing. To avoid such confusion, this sentence has been adjusted to:

A fundamental change in ocean ventilation response to astronomical forcing could thus constitute an alternative hypothesis for the observed phase-switch. It remains an open question, however, which modification(s) of Earth's boundary conditions has/have been responsible for reversing the response of ocean ventilation to astronomical forcing.

Reviewer #3, Comment 9 and our answer from previous review round.

I'd like to know more about how the authors do their phase analysis. Since the "100 kyr" cycle is in fact a composite of multiple terms with two main components at ~123 and 97 kyr, one worries about spurious phase assignment. The authors state that they "selected the phase from the frequency with the highest coherency". This approach may be a bit arbitrary- it will depend to begin with on how high the coherencies are (is there good evidence for phase locking?) and then on the dispersion of coherencies within the 90-135 kyr band).

I would back up and look at how compelling the evidence is for continuous 100 kyr components matching the predicted eccentricity terms. I've seen cases where there is power at say 85 kyr or 145 kyr that is pseudo-100 kyr, but where there isn't good evidence of the expected 123 and 97 kyr components when subjected to statistical tests. The phase analysis only makes sense if one has evidence for the underlying eccentricity-related components in each window of the analysis. Figure 1 supports the common presence of ~100 kyr components but more details would help substantiate the use of the 90-135 kyr band.

Answer:

We added Extended Data Figure 6 to complement the methods chapter. In this figure, we illustrate how we do our phase analysis, and especially how we select the discrete frequency from which we take the phase result.

With her/his comment, Reviewer #3 encouraged us to dig deeper into the question: "Is there good $\delta^{18}\text{O}$ - $\delta^{13}\text{C}$ coherence in the 100-kyr band all throughout the megasplice?". In addition to the wavelet spectrum of the $\delta^{13}\text{C}$ megasplice in Figure 1, we now also provide a visual sense for the $\delta^{18}\text{O}$ - $\delta^{13}\text{C}$ coherence in each analysis window, by making symbol sizes bigger for phase-results with higher coherence (Figure 2). This new Figure 2 shows high-coherence results (>0.6) all throughout the megasplice.

We then explored to what extent the correlation analyses between the oscillations around zero phase and 2.4 Myr eccentricity are impacted when low-coherence results are disregarded. The new Figure 3 (only considering phase results with coherence > 0.3) is similar to the original one, hence this figure is still underpinning our paleoclimate interpretations and conclusions. Admittedly, the choice for a cut-off at 0.3 coherence is rather arbitrary. Nevertheless, that specific cut-off level roughly corresponds with a confidence level of ~70% of non-zero coherence. The second arbitrary milestone, i.e. coherence = 0.6, corresponds to a confidence level of ~97.5% of non-zero coherence. One can thus have rather good confidence that there effectively is a dependency between the $\delta^{13}\text{C}$ and $\delta^{18}\text{O}$ signals in the frequency band of 100-kyr eccentricity.

The reviewer seems to argue for a wider frequency range for conducting our phase analysis. However, our arguments for choosing exactly the 90 – 135 kyr band are motivated in Extended Data Figure 5 and Extended Data Table 3: Too narrow frequency ranges make that some eccentricity-related variability is not considered for phase analysis. Too wide frequency ranges allow for non-eccentricity-related variability to be considered during phase analysis. Extended Data Table 3 comprises a sensitivity analysis of correlation as a function of the frequency range for phase analysis. From this table, it becomes clear that widening the frequency range (as suggested by the reviewer) would result in a drop of correlation between long-term eccentricity, on the one hand, and the oscillations around the zero-phase, on the other hand. We interpret this drop in correlation as an indication of too much non-eccentricity-related variability interfering with the phase analysis. To conclude, we use the 90 – 135 kyr frequency window to calculate phase relationships between $\delta^{13}\text{C}$ and $\delta^{18}\text{O}$ on time-scales of ~100-kyr eccentricity because this frequency window includes the four main ~100-kyr terms [(g4-g5) with 94.9 kyr period, (g3-g5) with 98.9 kyr period, (g4-g2) with 123.9 kyr period, (g3-g2) with 130.7 kyr period] and accommodates minor ($<5\%$) age-model inaccuracies.

The authors have misconstrued my argument. It is not to widen the frequency window around a central ~100 kyr window. Rather, it to ask them to validate their assumption that ~100 kyr

variance is phase-locked to an underlying eccentricity pacing. My point about 85 or 145 kyr bands was that I've seen a number of records that suggest 100 kyr cyclicality, but the interpretation falls apart if one looks rigorously at the spectra. The authors need to convince us first that they have a robust fingerprint of the ~123 and 97 kyr components of eccentricity continuously in the time series. Only then can they talk about a phase analysis of this signal in relation to an underlying (known) pacing by orbital eccentricity. I agree that the author's window is appropriate to following the coherence between $\delta^{13}\text{C}$ and $\delta^{18}\text{O}$ on ~100 kyr timescales- the question is whether the assumption that a 100 kyr pacing represents a fundamental coupling can be substantiated from the data analysis.

We apologize for having misunderstood the reviewer's original comment, but we are happy to read that the reviewer deems our frequency windows appropriate for the conducted phase analyses.

To answer his original question: We are sure that the reviewer will agree with us, that for the youngest part of the studied interval, the research by Lorraine Lisiecki (2010, *Nature Geoscience*) unambiguously demonstrates that the 100-kyr climate response is phase locked to eccentricity.

It is true that the wavelet analysis provided in Figure 1 does not provide the level of detail in the frequency domain that is needed to discern the 123 and 97 kyr components of eccentricity. To alleviate this issue, we added a simple MTM power spectrum of the $\delta^{13}\text{C}$ megasplite to Supplementary Figure 5. In this spectrum, all main eccentricity periodicities can be discerned, indicating the fundamental coupling between eccentricity and deep-sea carbon isotopes throughout the last 35 Ma.

REVIEWER COMMENTS

Reviewer #2 (Remarks to the Author):

The authors have addressed my previous comments in this version of the ms and I am largely content with that. Addition of the paleobotanical and additional Greenland ice constraints help a lot, and rephrasing to clarify some previously unclear statements is a definite improvement, within the framework of my questioning.

However, reviewer 3's comments certainly stirs things up and prompts some further questions.

Fundamentally, the method used here, seems to me a very thorough integration of the sub-orbital level isotopic relationships in this Cenozoic record, and as such, the authors have produced an impressive data set with revealing time series analysis. If rev. 3 had concerns about the methods, then the authors seem to have done their best to tackle these with new the additional analysis and SI figures etc.

The authors Northern Hemisphere cryosphere start-up explanation for the observed evolution of isotopic phasing is now quite compelling, although it has taken some work to improve the arguments (the additional references and more details on biome types really helps, plus the explanation of why the MMCT shouldn't show this). I have a few further questions on this (see below). In this sense and in my opinion, the study does represent a "conceptual step towards a process-led understanding of Icehouse Earth System Sensitivity", which is the final phrase of the abstract.

However, considering the exchange between the authors and reviewer 3, I have a further few questions that need addressing regarding the concepts of ice volume and weathering that are now upfront in the title:

1. Considering all the discussion leading to a focus on high-latitude biome shifts as agents/drivers of signal change, is it not more appropriate to focus on these 'biome' adjustments in general as the explanation for the shifts at 6 Ma, rather than explicitly saying 'ice volume', in the title? If the authors can defend that they think it is ice volume, its not a 'deal breaker', but I am interested to see a response to that questions.

2. The 'weathering' issue (also mentioned by Rev -3). Weathering is up there in the title, introduced as being important to carbon cycling on long time scales within the paper, discussed a fair bit, but we lose the connection between changes in weathering and terrestrial carbon sinks/biomes through the paper. A key conclusive statement is now: "Areal competition among continental ice and high-latitude biomes as the decisive factor for the late Miocene phase-switch."

As I see it, the high latitude 'biomes' under discussion as the agents of change are discussed in the context of centres for biological carbon pumping and storage, rather than chemical carbon pumping (weathering)? Unless the authors are arguing that the biome shift exposes weatherable materials in sensitive regions, strengthening the weathering feedback. This should be cleared up.

3. In Fig. 2, why is the d18O and d13C phasing signal much more variable in the Atlantic compared to the Pacific, especially in the mid to late Miocene, prior to 6 Ma (Fig 2). Does this not argue that the oceans are playing an important role in the d13C signal?

4. Beyond the doubt that NH ice from 6 Ma might explain the observed shift in isotopic phasing, reviewer 3's comments raise a philosophical question about how 'far' authors can go with putting interpretations upfront to make a provocative title, which rev. 3 questions.

First of all, I'd say that tackling carbon isotopes as de Vleeschouwer et al have done here is challenging and commendable. These systems are difficult and complicated. I agree with reviewer

3 that we continue to discover new things about deep marine $\delta^{13}\text{C}$ on Cenozoic time scales and will continue doing so but explanatory hypotheses should not be hidden, if that is the focus.

Earlier detailed 'stacked' Cenozoic analyses reporting on long-term $\delta^{13}\text{C}$ dynamics used rather generic titles, including:

Trends, rhythms, and aberrations in global climate 65 Ma to present: *Science*, v. 292, no. 5513, p. 686-693. Zachos, J. C., Pagani, M., Sloan, L. C., Thomas, E., and Billups, K., 2001,

Later, aspects of the Cenozoic $\delta^{13}\text{C}$ curve were discussed in relation to specific Earth systems, such as ocean overturning:

Ocean overturning since the Late Cretaceous: Inferences from a new benthic foraminiferal isotope compilation: *Paleoceanography*, v. 24, p. PA4216. Thirty-five million years of changing climate – carbon cycle dynamics. Cramer, B. S., Toggweiler, J. R. T., Wright, J. D., Katz, M. E., and Miller, G. H., 2009,

The first title used by Vleeschouwer et al., in the initial version of this ms was more of the general type of title, thus could have included any possible interpretation i.e.:

Thirty-five million years of changing climate – carbon cycle dynamics

However, the focus of the story in this paper, with the thorough interrogation of orbital forcing signal-evolution including evolution of the cycle phasing between benthic $\delta^{18}\text{O}$ and $\delta^{13}\text{C}$ moves us beyond the general observation of 'patterns in carbon cycle dynamics' with the purpose of leading to stronger conclusions, i.e.

"This analysis uncovers that 2.4-Myr eccentricity modulates the in-phase relationship between $\delta^{13}\text{C}$ and $\delta^{18}\text{O}$ during the Oligo-Miocene (34-6 Ma), potentially related to changes in continental weathering. At 6 Ma, a striking switch from in-phase to anti-phase behaviour occurs, signalling a threshold in the climate system. We hypothesize that Arctic glaciation and the emergence of bipolar ice sheets enabled eccentricity to exert a major influence on the size of continental carbon reservoirs."

This is the meat of the study such that a more specific title is justifiable.

"Ice volume and rock weathering mediate climate – carbon cycle feedbacks on eccentricity timescales", could work as an alternative, especially with 'mediate' as a compromise, although see my comments below regarding whether ice volume and weathering are exactly what should be referred to.

The bottom line is that I think this study will have a lasting impact in the field. Maybe not all the interpretations will stand the test of time (although they do/should within the framework of available analysis at this time). It surely takes a hard look at this complex syst

Reviewer #3 (Remarks to the Author):

I'm disappointed to review the current version of the manuscript, because the work has much to admire and the data set is very impressive. The revision goes some ways to giving an overview of the $\delta^{13}\text{C}$ cycle and acknowledging the fact that there can be multiple plausible interpretations of the factors underlying the $\delta^{13}\text{C}$ evolution. However, the response fails to address what to me are first-order reservations on whether this paper. Hence my most pointed comments unfortunately largely repeat my earlier review.

While the authors have gone some way toward acknowledging that the Arctic was (mostly) warm through the Pliocene, they continue to insist on the 6 Ma time as a milestone in Arctic cooling and the emergence of bipolar ice sheets. They are correct that we can't rule out ephemeral expansions of a Greenland ice cap- proving a negative is nearly impossible. But on the other hand, there is no POSITIVE evidence to support 6 Ma as a northern hemisphere milestone and a lot of evidence to suggest that it is NOT the case: the paleobotanical evidence cited, SST recorded in the North Atlantic and North Pacific, and the lack of ice rafted debris in the northern hemisphere prior to ~3.2 Ma, benthic $\delta^{18}O$ evidence.... There IS physical evidence of northern hemisphere glaciation in a time window of ~6.5-5.6 Ma, but then the climate shifted toward warmer and less glaciated conditions. Northern hemisphere SST, for example, was so warm in the Pliocene that the existence of tundra was improbable (and there are now orbitally-resolved SST for most of this time period and the Lake El record capturing relatively mild conditions in Siberia in the mid-late Pliocene).

As before, I feel strongly that they are staking an interesting finding in their $\delta^{13}C$ compilation to a very tenuous (not to say unlikely) climatic interpretation- as pointed out in some detail by Reviewer #2

"We hypothesize that this transition is consistent with Arctic cooling and the emergence of bipolar ice sheets: Prior to 6 Ma, low-latitude continental carbon reservoirs expanded during astronomically-forced cool spells. After 6 Ma, however, continental carbon reservoirs contract rather than expand during cold periods (i.e. glacials) due to competing effects between terrestrial ice and high-latitude biomes. "

And

"The ice volume increase at ~6 Ma in the Arctic, on the other hand, marks the start of very dynamic Arctic ecosystems, with significant changes in areal distribution between (ephemeral or not, that remains an open question) ice cover and high-latitude biomes. "

This revision thus continues to insist on bipolar glaciation as a consistent feature of post 6 Ma climate, in spite of many lines of evidence to the contrary.

Note that the proposed link between a fundamental climate change, carbon cycle changes (smaller terrestrial reservoir, change in rock weathering) is essential for the current paper (and title) to hold together- this is not a minor point. The authors do not have actual evidence for a change in rock weathering- as with many points, this is an inference that would help them interpret the $\delta^{13}C$ data.

"We are not claiming that weathering drives the $\delta^{13}C$ signal directly. We are claiming that when a non-temperature-determined factor enhances rock weathering (e.g. tectonic uplift), an incited carbon cycle expands its control on climate and therewith reduces its time-lag relative to the climate system on orbital time-scales: Hence periodic $\delta^{13}C_{\text{benthic}}$ (carbon cycle) leads relative to $\delta^{18}O_{\text{benthic}}$ (climate). We made this explicitly clear in the appropriate section. "

This response points out the tendency to make claims that cannot be supported by the underlying data. The rock weathering idea might be right- but it lacks evidence and/or a compelling model for PLAUSIBILITY.

I'm also not sure how to evaluate:

"Ocean ventilation is now discussed in more detail as an alternative hypothesis and open question for the phase-switch at 6 Ma. Importantly, according to research by Ann Holbourn et al. (2017), ocean ventilation did flip its response to astronomical forcing between the Middle Miocene and the Pleistocene. According to these authors, colder periods during the Miocene correspond to a more vigorous ocean circulation. Whereas during the Pleistocene, colder periods are characterized by less ocean ventilation. However, which mechanisms were decisive for reversing the response of

ocean ventilation to astronomical forcing. That, in combination with the argument that the isotopic signal of continental biomass is likely to be significantly larger than that of ocean ventilation makes that we stay with our preferred continent-based hypothesis. "

The whole-ocean $\delta^{13}\text{C}$ effect for the last glacial maximum, when there was a massive loss of terrestrial biomass, is on the order of 0.3-0.4 ‰. Do the authors believe that the Pliocene/Miocene world saw shifts this large or larger on a G-IG scale? On what basis do they scale proposed G-GI climate change in the Mio-Plio to changes in the terrestrial reservoir (e.g. does it take 4°C change to lead to a reservoir shift of X Gigatons or Y in $\delta^{13}\text{C}$...?)

Reviewer #2

The authors have addressed my previous comments in this version of the ms and I am largely content with that. Addition of the paleobotanical and additional Greenland ice constraints help a lot, and rephrasing to clarify some previously unclear statements is a definite improvement, within the framework of my questioning. However, reviewer 3's comments certainly stirs things up and prompts some further questions.

Fundamentally, the method used here, seems to me a very thorough integration of the sub-orbital level isotopic relationships in this Cenozoic record, and as such, the authors have produced an impressive data set with revealing time series analysis. If rev. 3 had concerns about the methods, then the authors seem to have done their best to tackle these with new the additional analysis and SI figures etc.

Thank you very much. We are pleased to read that on the methodological front, we have been able to clearly display the details of $\delta^{13}\text{C}$ megasplice construction, while at the same time convincing we managed to convince reviewer #2 of the robustness of our time-continuous time-series analyses.

The authors Northern Hemisphere cryosphere start-up explanation for the observed evolution of isotopic phasing is now quite compelling, although it has taken some work to improve the arguments (the additional references and more details on biome types really helps, plus the explanation of why the MMCT shouldn't show this). I have a few further questions on this (see below). In this sense and in my opinion, the study does represent a "conceptual step towards a process-led understanding of Icehouse Earth System Sensitivity", which is the final phrase of the abstract.

Thanks again. We are glad to hear that reviewer #2 now considers our in-phase vs. anti-phase hypotheses (i.e. the 6 Ma switch) as convincing, and that they now look at this work as an important conceptual step in the (paleo)climate community's quest for a mechanistic understanding of climate - carbon cycle interactions and their evolution through geologic time.

However, considering the exchange between the authors and reviewer 3, I have a further few questions that need addressing regarding the concepts of ice volume and weathering that are now upfront in the title:

Reviewer 2 Comment 1: “Ice volume” in the title?

1. Considering all the discussion leading to a focus on high-latitude biome shifts as agents/drivers of signal change, is it not more appropriate to focus on these ‘biome’ adjustments in general as the explanation for the shifts at 6 Ma, rather than explicitly saying ‘ice volume’, in the title? If the authors can defend that they think it is ice volume, its not a ‘deal breaker’, but I am interested to see a response to that questions.

The reviewer's is absolutely right. During the previous revision round, the focus of the 6 Ma hypothesis shifted from continental ice towards high-latitude biome dynamics (→ areal competition between ice, polar desert, tundra and taiga, each with significantly different carbon storage per unit area). We still believe that continental ice is a powerful part of the equation, given its virtually negligible carbon storage capacity. But it is no longer fundamental in our strongly nuanced interpretation. This refinement is now reflected in the title:

High-latitude biomes and rock weathering mediate climate – carbon cycle feedbacks on eccentricity timescales.

Reviewer 2 Comment 2: The role of rock weathering in the 6 Ma phase-switch?

2. The ‘weathering’ issue (also mentioned by Rev -3). Weathering is up there in the title, introduced as being important to carbon cycling on long time scales within the paper, discussed a fair bit, but we lose the connection between changes in weathering and terrestrial carbon sinks/biomes through the paper. A key conclusive statement is now: “Areal competition among continental ice and high-latitude biomes as the decisive factor for the late Miocene phase-switch.”

As I see it, the high latitude ‘biomes’ under discussion as the agents of change are discussed in the context of centres for biological carbon pumping and storage, rather than chemical carbon pumping (weathering)? Unless the authors are arguing that the biome shift exposes weatherable materials in sensitive regions, strengthening the weathering feedback. This should be cleared up.

The reviewer is correct: the in-phase vs. anti-phase hypotheses chiefly rely on high-latitude Arctic biomes as the agents of biological carbon pumping and storage. Chemical carbon pumping and weathering do not play a major role in these hypotheses.

The “Rock weathering” in the title refers to the second part of the manuscript, where we reveal oscillations around the zero-phase, at the rhythm of the 2.4-Myr eccentricity cycle. In this part of the manuscript, our hypotheses strongly rely on the locus of rock weathering: In the high-latitudes during the Oligocene, through glacial weathering, resulting in $\delta^{13}\text{C}$ leads during 2.4-Myr eccentricity minima. In the low latitudes during the Miocene, through chemical weathering, resulting in $\delta^{13}\text{C}$ leads during 2.4 Myr eccentricity maxima.

To make clear that each of the two terms in the title (high-latitude biomes and rock weathering) refer to its respective part of the manuscript, we have incorporated these terms in the subheadings of the Results & Discussion section.

1. High-latitude biome dynamics as phase-switch trigger (55 char.)
2. Weathering locus modulates Oligo-Miocene leads and lags (58 char.)

Reviewer 2 Comment 3: Scatter in phase results of Atlantic Sites 982 & 926

3. In Fig. 2, why is the d18O and d13C phasing signal much more variable in the Atlantic compared to the Pacific, especially in the mid to late Miocene, prior to 6 Ma (Fig 2). Does this not argue that the oceans are playing an important role in the d13C signal?

The reviewer refers to the relatively large scatter between roughly $-\pi/2$ and $\pi/2$ for the phase results of Atlantic Sites 982 and 926 between $\sim 15 - 6$ Ma. The scatter is especially large in comparison to the phase differences calculated for contemporary Pacific Sites 1146 and 1338.

The main cause for the scatter, however, are the numerous data gaps in the Site 926 and 982 time-series. Because of our 1.2-Myr-wide sliding window approach, data gaps can have a noticeable negative impact on the phase result as they reduce the amount of data that is available for analysis. The figures below show that -when the full range of the records is considered and thus phase results are not so much impacted by data gaps- phase results for the Atlantic and Pacific are not distinctly different: All four phase spectra indicate a slightly-positive phase-difference (between 0 and $+\pi/3$) between d18O and d13C at the frequency of 100-kyr eccentricity.

→ Atlantic Sites with data-gaps (Wilkins et al., 2017 and Andersson & Jansen, 2003)

→ Pacific Sites of high-resolution and good age-control (Holbourn et al., 2014, 2018)

Reviewer 2 Comment 4: How far can one go with explanatory hypotheses based on circumstantial evidence only? - A philosophical / ethical question.

4. Beyond the doubt that NH ice from 6 Ma might explain the observed shift in isotopic phasing, reviewer 3's comments raise a philosophical question about how 'far' authors can go with putting interpretations upfront to make a provocative title, which rev. 3 questions.

First of all, I'd say that tackling carbon isotopes as De Vleeschouwer et al have done here is challenging and commendable. These systems are difficult and complicated. I agree with reviewer 3 that we continue to discover new things about deep marine $\delta^{13}\text{C}$ on Cenozoic time scales and will continue doing so but explanatory hypotheses should not be hidden, if that is the focus.

Earlier detailed 'stacked' Cenozoic analyses reporting on long-term $\delta^{13}\text{C}$ dynamics used rather generic titles, including:

Trends, rhythms, and aberrations in global climate 65 Ma to present: *Science*, v. 292, no. 5513, p. 686-693. Zachos, J. C., Pagani, M., Sloan, L. C., Thomas, E., and Billups, K., 2001,

Later, aspects of the Cenozoic $\delta^{13}\text{C}$ curve were discussed in relation to specific Earth systems, such as ocean overturning:

Ocean overturning since the Late Cretaceous: Inferences from a new benthic foraminiferal isotope compilation: *Paleoceanography*, v. 24, p. PA4216. Thirty-five million years of changing climate – carbon cycle dynamics. Cramer, B. S., Toggweiler, J. R. T., Wright, J. D., Katz, M. E., and Miller, G. H., 2009,

The first title used by Vleeschouwer et al., in the initial version of this ms was more of the general type of title, thus could have included any possible interpretation i.e.:

Thirty-five million years of changing climate – carbon cycle dynamics

However, the focus of the story in this paper, with the thorough interrogation of orbital forcing signal-evolution including evolution of the cycle phasing between benthic $\delta^{18}\text{O}$ and $\delta^{13}\text{C}$ moves us beyond the general observation of 'patterns in carbon cycle dynamics' with the purpose of leading to stronger conclusions, i.e.

"This analysis uncovers that 2.4-Myr eccentricity modulates the in-phase relationship between $\delta^{13}\text{C}$ and $\delta^{18}\text{O}$ during the Oligo-Miocene (34-6 Ma), potentially related to changes in continental weathering. At 6 Ma, a striking switch from in-phase to anti-phase behaviour occurs, signalling a threshold in the climate system. We hypothesize that Arctic glaciation and the emergence of bipolar ice sheets enabled eccentricity to exert a major influence on the size of continental carbon reservoirs."

This is the meat of the study such that a more specific title is justifiable.

"Ice volume and rock weathering mediate climate – carbon cycle feedbacks on eccentricity timescales", could work as an alternative, especially with 'mediate' as a compromise, although see my comments below regarding whether ice volume and weathering are exactly what should be referred to.

We thank the reviewer for confirming our feeling that a more specific title is appropriate in this case.

As discussed above, we agree with the reviewer that "ice volume" may not be the best term to put forward in the title, and it has been replaced by "high-latitude biomes" to better reflect the beef of our 6-Ma hypothesis.

The key to our 26-Ma hypothesis however resides with the locus of rock weathering (high latitudes glacial weathering prior to 26 Ma; low latitudes monsoonal weathering after 26 Ma), hence we prefer to keep "rock weathering" up front in the title.

The bottom line is that I think this study will have a lasting impact in the field. Maybe not all the interpretations will stand the test of time (although they do/should within the framework of available analysis at this time). It surely takes a hard look at this complex system.

Reviewer #3

I'm disappointed to review the current version of the manuscript, because the work has much to admire and the data set is very impressive. The revision goes some ways to giving an overview of the d13C cycle and acknowledging the fact that there can be multiple plausible interpretations of the factors underlying the d13C evolution. However, the response fails to address what to me are first-order reservations on whether this paper. Hence my most pointed comments unfortunately largely repeat my earlier review.

Reviewer 3 Comment 1: Pliocene Northern Hemisphere Ice sheets - A hot topic.

While the authors have gone some way toward acknowledging that the Arctic was (mostly) warm through the Pliocene, they continue to insist on the 6 Ma time as a milestone in Arctic cooling and the emergence of bipolar ice sheets. They are correct that we can't rule out ephemeral expansions of a Greenland ice cap- proving a negative is nearly impossible. But on the other hand, there is no POSITIVE evidence to support 6 Ma as a northern hemisphere milestone and a lot of evidence to suggest that it is NOT the case: the paleobotanical evidence cited, SST recorded in the North Atlantic and North Pacific, and the lack of ice rafted debris in the northern hemisphere prior to ~3.2 Ma, benthic d1O evidence.... There IS physical evidence of northern hemisphere glaciation in a time window of ~6.5-5.6 Ma, but then the climate shifted toward warmer and less glaciated conditions. Northern hemisphere SST, for example, was so warm in the Pliocene that the existence of tundra was improbably (and there are now orbitally-resolved SST for most of this time period and the Lake El record capturing relatively mild conditions in Siberia in the mid-late Pliocene).

As before, I feel strongly that they are staking an interesting finding in their d13C compilation to a very tenuous (not to say unlikely) climatic interpretation- as pointed out in some detail by Reviewer #2.

"We hypothesize that this transition is consistent with Arctic cooling and the emergence of bipolar ice sheets: Prior to 6 Ma, low-latitude continental carbon reservoirs expanded during astronomically-forced cool spells. After 6 Ma, however, continental carbon reservoirs contract rather than expand during cold periods (i.e. glacials) due to competing effects between terrestrial ice and high-latitude biomes. "

And

"The ice volume increase at ~6 Ma in the Arctic, on the other hand, marks the start of very dynamic Arctic ecosystems, with significant changes in areal distribution between (ephemeral or not, that remains an open question) ice cover and high-latitude biomes. "

This revision thus continues to insist on bipolar glaciation as a consistent feature of post 6 Ma climate, in spite of many lines of evidence to the contrary.

We fully acknowledge the fact that our d13C megasplice solely provides circumstantial evidence for a 6-Ma start-up of Northern Hemisphere cryosphere. We have emphasized this even more in the revised version of the manuscript, and our interpretations remain deeply rooted in the available literature on the commencement of bi-polar glaciation.

During the previous revision round, we provided (1) additional references, (2) more details on high-latitude biome dynamics and (3) we explained why the MMCT cooling doesn't show a phase-switch. The d13C megasplice circumstantial evidence is thus intertwined with ample literature on the topic (multiple Mio-Pliocene ice rafted debris records¹⁻⁴, Be isotopes⁵, seismic analysis⁶, pollen studies^{7,8}, modelling of the pCO₂ threshold of bi-polar glaciation⁹, modelling of vegetation dynamics¹⁰, proxy-based

pCO_{211} and SST reconstructions¹², obliquity signatures in benthic $d18O$ records^{13,14}). **All these elements combined lead to our interpretation that the Pliocene warming was not sufficient to reverse the anti-phase dynamics that exist since 6 Ma** [→ i.e. coincident with late Miocene cooling and (ephemeral or not?) northern hemisphere glaciation]. The issue of Pliocene warming and its consequences for the NH cryosphere was and still is explicitly discussed in the manuscript.

For all those reasons, are interpretation should not be dismissed as tenuous and unlikely. Reviewer #2 writes: “*The Northern Hemisphere cryosphere start-up explanation for the observed evolution of isotopic phasing is now quite compelling*”.

We also take the opportunity to highlight two figures on the topic, from two highly-regarded papers.

1. De Schepper et al. (2014, Earth Science Reviews), in a global synthesis of Pliocene glaciation evidence (including NH ice rafted debris), identifies four distinct bi-polar glaciation events, with the Greenlandic ice sheet occasionally reaching tidewater. This series of clearly recognizable Pliocene glaciation events thus indicate that **a dynamic northern hemisphere cryosphere existed during the Pliocene, not least on eccentricity (10⁵ year) time scales**:
 - a. At ~4.9 Ma, local glaciation is suggested for Greenland, Iceland and Scandinavia.
 - b. At ~4.0 Ma, local glaciation is suggested for Greenland, Iceland, Scandinavia, the Barents Sea, Alaska and British Columbia.
 - c. At ~3.6 Ma, major glaciation is suggested for Greenland, Iceland and the Barents Sea. Local glaciation in Alaska, Canada and Scandinavia.
 - d. At ~3.3 Ma, major glaciation is suggested for Greenland, Iceland, the Barents Sea and Scandinavia. Local glaciation in Alaska. As mentioned by the reviewer, the Lake El’gygytyn record indeed indicates non-glacial conditions for the mid-late Pliocene. But, around ~3.3 Ma, the reconstructed mean temperature of the warmest month at El’gygytyn is comparable to that of the Holocene (Brigham-Grette et al., 2013, Science). Hence, a priori, not too hot for bi-polar ice caps.

Figure from De Schepper et al. (2014): Summary of the major Northern Hemisphere ice sheets during the Pliocene.

Reprinted from *Earth-Science Reviews*, **135**, 83-102, De Schepper et al., A global synthesis of the marine and terrestrial evidence for glaciation during the Pliocene Epoch, Copyright 2014, with permission from Elsevier

2. Brigham-Grette et al. (2013, Science) present coupled climate-vegetation model simulations with Pliocene boundary conditions: A $p\text{CO}_2$ range between 300 ppm and 400 ppm is generally accepted for the Pliocene (e.g. de la Vega et al., 2020, Scientific Reports). These model results show that **high-latitude biomes are very susceptible to $p\text{CO}_2$ and orbital forcing, with contingent areal proportions of tundra, boreal forest and grassland.**

[REDACTED - Figure 4 from Brigham-Grette, et al. (2013)]

However, it is in everyone's interest to break this deadlock, and **we are certainly willing to leave the door open for an ice-free northern hemisphere during the Pliocene.** In the previous version of the manuscript we already wrote:

“The modest warming of the Pliocene seems insufficient to reverse this dynamic, with Pliocene sea surface temperatures (SST) in the NH high latitudes (>50°N) failing to return the comparably warm Tortonian conditions (8 - 12 Ma in Figure 2a of Herbert et al.).”

We now added

“The warmest intervals of the Pliocene, even when they were characterized by little or no NH continental ice, thus stayed with anti-phased climate-carbon interactions. In our hypothesis, this anti-phased behaviour would then be primarily driven by areal competition between tundra and boreal forest.”

We adapted the title in response to the suggestion of Reviewer #2, and we toned down every allusion to continental ice since 6 Ma in the paper. **We now consistently refer to “high latitude biomes” or “Arctic continental carbon reservoirs”.** These terms encompass polar deserts, tundra, peat- and wetlands as well as boreal forest and thus **take the sting out of the debate on whether or not persistent bi-polar glaciation existed since 6 Ma.**

Reviewer 3 Comment 2: How far can one go with explanatory hypotheses based on circumstantial evidence only? - A philosophical / ethical question.

Note that the proposed link between a fundamental climate change, carbon cycle changes (smaller terrestrial reservoir, change in rock weathering) is essential for the current paper (and title) to hold together- this is not a minor point. The authors do not have actual evidence for a change in rock weathering- as with many points, this is an inference that would help them interpret the $\delta^{13}\text{C}$ data.

“We are not claiming that weathering drives the $\delta^{13}\text{C}$ signal directly. We are claiming that when a non-temperature-determined factor enhances rock weathering (e.g. tectonic uplift), an incited carbon cycle expands its control on climate and therewith reduces its time-lag relative to the climate system on orbital time-scales: Hence periodic $\delta^{13}\text{C}_{\text{benthic}}$ (carbon cycle) leads relative to $\delta^{18}\text{O}_{\text{benthic}}$ (climate). We made this explicitly clear in the appropriate section. “

This response points out the tendency to make claims that cannot be supported by the underlying data. The rock weathering idea might be right- but it lacks evidence and/or a compelling model for PLAUSIBILITY.

Reviewer #3 criticises that our paper does not present direct evidence for the proposed hypotheses. We fully attest to the reviewer’s remark that we are doing nothing more than presenting inferences that fit with the observed $\delta^{18}\text{O}$ - $\delta^{13}\text{C}$ patterns. But we certainly do no less than that, and our effort was kindly described by reviewer #2 as *challenging* and *commendable*. Not the less, the reviewer raises a philosophical question about how far we can go with paleoclimate interpretations based on circumstantial evidence and inductive reasoning.

The deep-ocean carbon isotope system is complicated. The vast majority of paleoceanographers will admit that all $\delta^{13}\text{C}_{\text{benthic}}$ -based conclusions are based on inductive reasoning. By definition, such conclusions are uncertain and may change in the future. It is therefore impossible to prove a $\delta^{13}\text{C}_{\text{benthic}}$ -based conclusion right or wrong. One can merely judge whether it is strong or weak, plausible or implausible.

My co-authors and I took an in-depth look at this complex system and we came up with explanatory hypotheses. These hypotheses are partly provocative, but in our opinion the most plausible explanation for the observed phase-relationships.

Reviewer #2 writes: “*This study will have a lasting impact in the field. Maybe not all the interpretations will stand the test of time. [...] They do/should within the framework of available analysis at this time*”.

In other words, it’s the best we can do with the presently-available data. At the end of the “*Weathering locus modulates Oligo-Miocene leads and lags*” subsection, we admit that further modelling work is needed to test our concepts (and thus to evaluate their plausibility) and we give some concrete ideas as how to such numerical modelling study could be conceived:

The concepts presented here will require testing with Earth System models that can simulate the temporal and spatial distribution of weathering, using time-varying orbits and under different boundary conditions: with and without a Himalayan mountain range and with different amounts of continental ice on Antarctica.

Reviewer 3 Comment 3: Ocean ventilation as an alternative hypothesis for the 6 Ma phase switch.

I'm also not sure how to evaluate:

“Ocean ventilation is now discussed in more detail as an alternative hypothesis and open question for the phase-switch at 6 Ma. Importantly, according to research by Ann Holbourn et al. (2017), ocean ventilation did flip its response to astronomical forcing between the Middle Miocene and the Pleistocene. According to these authors, colder periods during the Miocene correspond to a more vigorous ocean circulation. Whereas during the Pleistocene, colder periods are characterized by less ocean ventilation. However, which mechanisms were decisive for reversing the response of ocean ventilation to astronomical forcing. That, in combination with the argument that the isotopic signal of continental biomass is likely to be significantly larger than that of ocean ventilation makes that we stay with our preferred continent-based hypothesis. “

The whole-ocean $\delta^{13}\text{C}$ effect for the last glacial maximum, when there was a massive loss of terrestrial biomass, is on the order of 0.3-0.4 ‰. Do the authors believe that the Pliocene/Miocene world saw shifts this large or larger on a G-IG scale? On what basis do they scale proposed G-GI climate change in the Mio-Plio to changes in the terrestrial reservoir (e.g. does it take 4°C change to lead to a reservoir shift of X Gigatons or Y in $\delta^{13}\text{C}$...?).

We are not sure whether we fully understand the reviewer's point here.

If the reviewer intends to emphasize the fact that ocean ventilation can only account for 0.3 – 0.4 ‰ of $\delta^{13}\text{C}$ variability, whereas the $\delta^{13}\text{C}$ megasplice exhibits a far greater (~1 ‰) variability: That is exactly the point we want to make. The ocean ventilation hypothesis fits the sign of the change (in-phase behaviour >6 Ma; anti-phase behaviour <6 Ma). However, it doesn't fit the amplitude of the observed $\delta^{13}\text{C}_{\text{benthic}}$ variability.

We refer the reviewer to Keigwin and Boyle (1985). These authors wrote:

“Although down-core carbon isotope data in the Atlantic and Pacific reflect changes in deep-ocean circulation patterns, a larger fraction of $\delta^{13}\text{C}$ variability in both oceans is due to changes in the inventory of continental reduced carbon.”

In our manuscript, a similar message is phrased as:

“We note that our preferred areal competition hypothesis is compatible with an ocean ventilation mechanism. [...] However, we consider the areal competition among high-latitude biomes the most parsimonious explanation as the magnitude of the continental biomass isotopic signal is likely to be significantly larger than a mere ventilation signal¹⁵.”

This whole paragraph is conceived in such a way that we first propose an alternative hypothesis, after which arguments are presented as to why we still prefer our continental biomass hypothesis. In other words, we first openly consider an alternative hypothesis, and then scrutinize its compatibility, amplitude and plausibility.

References cited within this document.

- 1 Larsen, H. C. *et al.* Seven Million Years of Glaciation in Greenland. *Science* **264**, 952-955, doi:10.1126/science.264.5161.952 (1994).
- 2 De Schepper, S., Gibbard, P. L., Salzmann, U. & Ehlers, J. A global synthesis of the marine and terrestrial evidence for glaciation during the Pliocene Epoch. *Earth-Science Reviews* **135**, 83-102, doi:10.1016/j.earscirev.2014.04.003 (2014).
- 3 Denton, G. H. & Armstrong, R. L. Miocene-Pliocene glaciations in southern Alaska. *Am. J. Sci.* **267**, 1121-1142, doi:10.2475/ajs.267.10.1121 (1969).
- 4 John, K. E. K. S. & Krissek, L. A. The late Miocene to Pleistocene ice-rafting history of southeast Greenland. *Boreas* **31**, 28-35, doi:10.1111/j.1502-3885.2002.tb01053.x (2002).
- 5 Bierman, P. R., Shakun, J. D., Corbett, L. B., Zimmerman, S. R. & Rood, D. H. A persistent and dynamic East Greenland Ice Sheet over the past 7.5 million years. *Nature* **540**, 256-260, doi:10.1038/nature20147 (2016).
- 6 Pérez, L. F., Nielsen, T., Knutz, P. C., Kuijpers, A. & Damm, V. Large-scale evolution of the central-east Greenland margin: New insights to the North Atlantic glaciation history. *Global Planet. Change* **163**, 141-157, doi:https://doi.org/10.1016/j.gloplacha.2017.12.010 (2018).
- 7 Pound, M. J., Haywood, A. M., Salzmann, U. & Riding, J. B. Global vegetation dynamics and latitudinal temperature gradients during the Mid to Late Miocene (15.97–5.33Ma). *Earth-Science Reviews* **112**, 1-22, doi:https://doi.org/10.1016/j.earscirev.2012.02.005 (2012).
- 8 Matthews Jr, J., Telka, A. & Kuzmina, S. Late Neogene insect and other invertebrate fossils from Alaska and Arctic/Subarctic Canada. *Invertebrate Zoology* **16**, 126-153 (2019).
- 9 DeConto, R. M. *et al.* Thresholds for Cenozoic bipolar glaciation. *Nature* **455**, 652-656, doi:10.1038/nature07337 (2008).
- 10 Brigham-Grette, J. *et al.* Pliocene Warmth, Polar Amplification, and Stepped Pleistocene Cooling Recorded in NE Arctic Russia. *Science* **340**, 1421-1427, doi:10.1126/science.1233137 (2013).
- 11 de la Vega, E., Chalk, T. B., Wilson, P. A., Bysani, R. P. & Foster, G. L. Atmospheric CO₂ during the Mid-Piacenzian Warm Period and the M2 glaciation. *Scientific Reports* **10**, 11002, doi:10.1038/s41598-020-67154-8 (2020).
- 12 Herbert, T. D. *et al.* Late Miocene global cooling and the rise of modern ecosystems. *Nature Geosci* **9**, 843-847, doi:10.1038/ngeo2813 (2016).
- 13 Drury, A. J., John, C. M. & Shevenell, A. E. Evaluating climatic response to external radiative forcing during the late Miocene to early Pliocene: New perspectives from eastern equatorial Pacific (IODP U1338) and North Atlantic (ODP 982) locations. *Paleoceanography* **31**, 167-184, doi:10.1002/2015PA002881 (2016).
- 14 Drury, A. J. *et al.* Deciphering the State of the Late Miocene to Early Pliocene Equatorial Pacific. *Paleoceanography and Paleoclimatology* **33**, 246-263, doi:10.1002/2017pa003245 (2018).
- 15 Keigwin, L. & Boyle, E. in *The Carbon Cycle and Atmospheric CO₂: Natural Variations Archean to Present* Vol. 32 (eds Sundquist E. & W.S. Broecker) 319-328 (American Geophysical Union, 1985).

REVIEWERS' COMMENTS:

Reviewer #2 (Remarks to the Author):

The process of review here has been an interesting journey. From my perspective the authors have addressed the reviewers comments as well as they can and defended their position appropriately, thus I would be happy that this moves forward to publication after a few last things here.

I like the new title. I have recently been involved with a big review of the Miocene, submitted to *Paleoceanography and Palaeoclimatology*:

Steinthorsdottir, M., Coxall, H., Boer, A. D., Huber, M., Barbolini, N., Bradshaw, C., Burls, N., Feakins, S., Gasson, E., Henderiks, J., Holbourn, A., Kiel, S., Kohn, M., Knorr, G., Kürschner, W., Lear, C., Liebrand, D., Lunt, D., Mörs, T., Pearson, P., Pound, M., Stoll, H., and Strömberg, C., Submitted to *Paleoceanography and Palaeoclimatology*, 1 July 2020, *The Miocene: The Future of the Past: Paleoclimatology and Paleoclimatology*, p. 2020PA004037.

From reviewing and synthesising massive amounts of literature across the bread of palaeontology, climatology palaeoceanography and modelling there is one thing I have learned from this is how hugely significant the changes to terrestrial biomes were through the mid to late Miocene. This paper is consistent with this view.

Abstract:

Line 26-27: "We hypothesize that this transition is consistent with Arctic cooling and the emergence of bipolar ice sheets"

In line with the new title and general thread of the paper could this not be changed to 'cool-polar biomes'.

Just a suggestion. I think keeping ice sheets is ok, since it does represent one of the carbon system signal drivers discussed in the ms, and there is evidence for Greenland Ice from 6 Ma.

" We still believe that continental ice is a powerful part of the equation, given its virtually negligible carbon storage capacity."

This statement in the rebuttal is really nice and clear – can you fit this into the paper .

-The role of rock weathering in the 6 Ma phase-switch?

The authors have cleared up the mechanical ice erosion aspects in the rebuttal but I think it would help in the paper if they still plainly say there are two different carbon system subsystems involved, then we are prepared for these as they come up under the new subtitles (which help).

Inductive reasoning.

"The vast majority of paleoceanographers will admit that all $\delta^{13}\text{C}$ benthic-based conclusions are based on inductive reasoning."

True. Aspects of a paleoceanographer's work has to involve inductive reasoning that relies upon existing background-theory, - you can investigate some new things and test some new ideas in a study but you can't test and push the boundaries of all of the theory at once. Here some basic assumptions regarding controls on benthic $\delta^{13}\text{C}$ are embedded.

The reason that interpretation of $\delta^{13}\text{C}$ in benthic forams is complicated is that there are an array of possible causal explanations for changes on orbital time scales including:

-changes in terrestrial carbon burial (which ends up being the favoured explanatory hypotheses here),

- changes in carbon burial in deep sea sediments,
- changes in where erosion is happening cross latitudes
- and changes in ocean ventilation changing the $\delta^{13}\text{C}$ of bottom water where the benthic foram signal carriers live.

...and these could all work as interpretations in this case to some extent within the existing background theoretical framework. The authors now argue pretty well why they favour some of these over others.

I believe that 6 Ma is a threshold for many subsystems. The early history of northern hemisphere ice is still a tough egg to crack, since the massively dynamic ice sheets have bulldozed so much of the evidence on land or covered it km deep in debris on the seafloor. Thus the snippets of data we do have, which are only gradually coming to light and being synthesized (citations in the current ms do a nice job of bringing these snippets together) are crucial.

I know that the authors argue that its northern biomes that are important, rather than Antarctic ones but it is my understanding that Antarctic glacial extent also increased around 8-6 Ma, at least since 8 Ma (although that's not shown in the AIS depiction in fig 1- see comment below). It would be good to just clarify that.

-A quick comment here, in your Fig. 1, I was interested to see your representation of Antarctic Ice volume and notice that the references you use for this picture are oldish (Oerlemans et al, 2004 and Berger et al., 1999) and predate more recent work that has targeted Antarctic Neogene records. Can you update this, and check that this is still the paradigm as it is my understanding that more of a jump in ice volume occurred in Antarctica between 11 and 5 Ma. E.g.

Shakun, J. D., Corbett, L. B., Bierman, P. R., Underwood, K., Rizzo, D. M., Zimmerman, S. R., Caffee, M. W., Naish, T., Golledge, N. R., and Hay, C. C., 2018, Minimal East Antarctic Ice Sheet retreat onto land during the past eight million years: *Nature*, v. 558, no. 7709, p. 284-287.

Also cross check recent IODP Wilkesland drilling results.

The line of arguments bringing us to the conclusion that terrestrial biome shifts, rather than ocean overturning, are the most plausible causal explanations for the global (benthic foram) $\delta^{13}\text{C}$ signals, is to me convincing at this time.

Inductive reasoning can never guarantee the truth of the conclusion, but in Paleooceanography, and like most complicated multivariate process operating in the geological past on long time scales, that's often what we must lean on. Look at debates over the cause of the PETM.

Other comments

Lines 102-104: something missing in this sentence now, a typo I think.

Lines 129-130: "The abrupt shift from in-phase to anti-phase behaviour around 6 Ma in the Pacific and shortly after in the Atlantic Ocean. ",

What's the delay time? Looks like its about 1 myrs later in the Atlantic...so around 5 Ma. What caused that change? Does this not suggest that the oceans are involved in this some how? Or since its just one or 2 sites that give us the records (comparing Site 926 in the Atlantic and U1338 in the Pacific) could this be within the age model error? Or do we trust these age models?

Throughout: "high-latitude biomes" maybe this needs to be 'cool' high latitude biomes since what is today 'boreal forest might once have been the high-latitude biome 'normal'. – i.e. forests , which is a biome, use to grow in the polar circle.

So if its 5 Ma in the Atlantic that the switch to anti phase behaviour happens does this make a difference regarding other correlations? Should you be saying 5-6 Ma, which with the end of the Miocene at 5.3 Ma is basically very close the M/P epoch boundary (massive changes, hence new geological epoch – which stratigraphers/palaeontologists knew about decades ago!).

Lines 264-267:

“Hence, the observed mid-Miocene in-phase $\delta^{13}\text{C}$ - $\delta^{18}\text{O}$ behaviour could also be explained through fluctuations in ocean ventilation. However, exactly why ocean ventilation was vigorous during cold periods in the mid-Miocene, but sluggish during cold periods in the Pleistocene is poorly understood”

I agree with this doubt. Plus contourite (deep sea drift) constraints on NADW behaviour through the Miocene don't always match perspectives from benthic foram $\delta^{13}\text{C}$

Uenzelmann-Neben, G., and Gruetzner, J., 2018, Chronology of Greenland Scotland Ridge overflow: What do we really know?: *Marine Geology*, v. 406, p. 109-118.

Reviewer #3 (Remarks to the Author):

I continue to disagree with the author's interpretation that the change in phasing of the $\delta^{13}\text{C}$ signal must be associated uniquely with a northern hemisphere and perhaps cryosperic onset. In their response, they point to evidence for short-lived NH glaciations in the Pliocene that I do not dispute at all- but the claim of a persistent NH glaciation after 6 Ma continues to be unsupported.

However, the paper has evolved to consider many more possibilities than the first two drafts and now reads in a much more nuanced way. The replies to the latest reviews are extensive and on-going. I think the balance has now tipped in favor of accepting the paper and I will be glad to see it join the literature.

Reviewer #2:

The process of review here has been an interesting journey. From my perspective the authors have addressed the reviewers comments as well as they can and defended their position appropriately, thus I would be happy that this moves forward to publication after a few last things here.

I like the new title. I have recently been involved with a big review of the Miocene, submitted to *Paleoceanography and Palaeoclimatology*:

Steinthorsdottir, M., Coxall, H., Boer, A. D., Huber, M., Barbolini, N., Bradshaw, C., Burls, N., Feakins, S., Gasson, E., Henderiks, J., Holbourn, A., Kiel, S., Kohn, M., Knorr, G., Kürschner, W., Lear, C., Liebrand, D., Lunt, D., Mörs, T., Pearson, P., Pound, M., Stoll, H., and Strömberg, C., Submitted to *Paleoceanography and Paleoclimatology*, 1 July 2020, The Miocene: The Future of the Past: *Paleoceanography and Paleoclimatology*, p. 2020PA004037.

From reviewing and synthesising massive amounts of literature across the bread of palaeontology, climatology palaeoceanography and modelling there is one thing I have learned from this is how hugely significant the changes to terrestrial biomes were through the mid to late Miocene. This paper is consistent with this view.

Abstract:

Line 26-27: "We hypothesize that this transition is consistent with Arctic cooling and the emergence of bipolar ice sheets"

In line with the new title and general thread of the paper could this not be changed to 'cool-polar biomes'.

Just a suggestion. I think keeping ice sheets is ok, since it does represent one of the carbon system signal drivers discussed in the ms, and there is evidence for Greenland Ice from 6 Ma.

This line in the abstract has been adapted according to the suggestion of the editor, and thus only refers to Arctic cooling (and thus no longer to Greenland Ice).

“ We still believe that continental ice is a powerful part of the equation, given its virtually negligible carbon storage capacity.”

This statement in the rebuttal is really nice and clear – can you fit this into the paper .

This statement has been built into the concluding paragraph, in a sentence that underlines that with ice, our hypothesis works very efficiently.

-The role of rock weathering in the 6 Ma phase-switch?

The authors have cleared up the mechanical ice erosion aspects in the rebuttal but I think it would help in the paper if they still plainly say there are two different carbon system subsystems involved, then we are prepared for these as they come up under the new subtitles (which help).

Unfortunately, we had to remove the tertiary subtitles, in line with the Nature Communications style guidelines.

However, we used the secondary subtitles in the Results and Discussion section to clearly associate “High-latitude biome dynamics” with the phase-switch, and “Weathering locus” with leads and lags in the climate-carbon cycle system.

Inductive reasoning.

“The vast majority of paleoceanographers will admit that all $\delta^{13}\text{C}$ benthic-based conclusions are based on inductive reasoning.”

True. Aspects of a paleoceanographer’s work has to involve inductive reasoning that relies upon existing background-theory, - you can investigate some new things and test some new ideas in a study but you can’t test and push the boundaries of all of the theory at once. Here some basic assumptions regarding controls on benthic $\delta^{13}\text{C}$ are embedded.

The reason that interpretation of $\delta^{13}\text{C}$ in benthic forams is complicated is that there are an array of possible causal explanations for changes on orbital time scales including:

- changes in terrestrial carbon burial (which ends up being the favoured explanatory hypotheses here),
- changes in carbon burial in deep sea sediments,
- changes in where erosion is happening cross latitudes
- and changes in ocean ventilation changing the $\delta^{13}\text{C}$ of bottom water where the benthic foram signal carriers live.

...and these could all work as interpretations in this case to some extent within the existing background theoretical framework. The authors now argue pretty well why they favour some of these over others.

Thank you.

I believe that 6 Ma is a threshold for many subsystems. The early history of northern hemisphere ice is still a tough egg to crack, since the massively dynamic ice sheets have bulldozed so much of the evidence on land or covered it km deep in debris on the seafloor. Thus the snippets of data we do have, which are only gradually coming to light and being synthesized (citations in the current ms do a nice job of bringing these snippets together) are crucial.

I know that the authors argue that its northern biomes that are important, rather than Antarctic ones but it is my understanding that Antarctic glacial extent also increased around 8-6 Ma, at least since 8 Ma (although that’s not shown in the AIS depiction in fig 1- see comment below). It would be good to just clarify that.

-A quick comment here, in your Fig. 1, I was interested to see your representation of Antarctic Ice volume and notice that the references you use for this picture are oldish (Oerlemans et al, 2004 and Berger et al., 1999) and predate more recent work that has targeted Antarctic Neogene records. Can you update this, and check that this is still the paradigm as it is my understanding that more of a jump in ice volume occurred in Antarctica between 11 and 5 Ma. E.g.

Shakun, J. D., Corbett, L. B., Bierman, P. R., Underwood, K., Rizzo, D. M., Zimmerman, S. R., Caffee, M. W., Naish, T., Golledge, N. R., and Hay, C. C., 2018, Minimal East Antarctic Ice Sheet retreat onto land during the past eight million years: *Nature*, v. 558, no. 7709, p. 284-287.

Also cross check recent IODP Wilkesland drilling results.

The Oerlemans et al. 2004 is the only quantitative reconstruction of Antarctic ice volume that is time-continuous for the last 35 million years. The more recent Shakun et al. (2018) paper concludes that “warming during the past eight million years was insufficient to cause widespread or long-lasting meltback of the EAIS margin onto land”. At least, this conclusion is in line with the Oerlemans et al. (2004) reconstruction, which suggests a rather stable Antarctic ice volume [$15 - 20 \cdot 10^6 \text{ km}^3$] over the last 8 million years.

The line of arguments bringing us to the conclusion that terrestrial biome shifts, rather than ocean overturning, are the most plausible causal explanations for the global (benthic foram) $\delta^{13}\text{C}$ signals, is to me convincing at this time.

Thank you.

Inductive reasoning can never guarantee the truth of the conclusion, but in Paleoceanography, and like most complicated multivariate process operating in the geological past on long time scales, that's often what we must lean on. Look at debates over the cause of the PETM.

Other comments

Lines 102-104: something missing in this sentence now, a typo I think.

Checked and corrected

Lines 129-130: “The abrupt shift from in-phase to anti-phase behaviour around 6 Ma in the Pacific and shortly after in the Atlantic Ocean.”

What's the delay time? Looks like its about 1 myrs later in the Atlantic...so around 5 Ma. What caused that change? Does this not suggest that the oceans are involved in this some how? Or since its just one or 2 sites that give us the records (comparing Site 926 in the Atlantic and U1338 in the Pacific) could this be within the age model error? Or do we trust these age models?

The time-control on these records is solid. Age-model discrepancies are not the origin of the small delay.

The reason for the delay between the different sites in question (1146, U1337, U1338, 926) remains an open question. But it is clear that even within one basin, time-delays between sites can be discerned (e.g. Site 1146 vs. U1338/U1337 in the Pacific). To me, this suggest that the oceans are involved in determining the speed at which the isotopic signal of climate-carbon cycle perturbations (occurring in the biosphere & atmosphere @ 6 Ma, according to our hypothesis) reaches the abyssal depths of the ocean basins. A comprehensive explanation of this issue is however beyond the scope of our manuscript.

Throughout: “high-latitude biomes” maybe this needs to be ‘cool’ high latitude biomes since what is today ‘boreal forest might once have been the high-latitude biome ‘normal’. – i.e. forests , which is a biome, use to grow in the polar circle.

Already in the abstract, we define what we understand under Arctic biomes: ice, tundra, taiga. For that reason, we decided against adding the adjective “cool” throughout the manuscript.

So if its 5 Ma in the Atlantic that the switch to anti phase behaviour happens does this make a difference regarding other correlations? Should you be saying 5-6 Ma, which with the end of the Miocene at 5.3 Ma is basically very close the M/P epoch boundary (massive changes, hence new geological epoch – which stratigraphers/palaeontologists knew about decades ago!).

No: we should not be saying 5 – 6 Ma, but we should stick to the 6 Ma label.

This is because the deep-marine sedimentary archives studied here are recording a fundamental change in Earth System dynamics. And as it goes with recorders, some are more sensitive than others, and some record a change more readily than others. As I explained above, I think it is reasonable to assume that the time-delays between sites (1146, U1337, U1338, 926) are related to the speed at which isotopic signals of reach the deep-ocean. It reaches Site 1146 first, at 6 Ma, so by that time the “phase-switch” in the atmosphere/biosphere system had already taken place.

Lines 264-267:

“Hence, the observed mid-Miocene in-phase $\delta^{13}\text{C}$ - $\delta^{18}\text{O}$ behaviour could also be explained through fluctuations in ocean ventilation. However, exactly why ocean ventilation was vigorous during cold periods in the mid-Miocene, but sluggish during cold periods in the Pleistocene is poorly understood”

I agree with this doubt. Plus contourite (deep sea drift) constraints on NADW behaviour through the Miocene don't always match perspectives from benthic forams $\delta^{13}\text{C}$

Uenzelmann-Neben, G., and Gruetzner, J., 2018, Chronology of Greenland Scotland Ridge overflow: What do we really know?: Marine Geology, v. 406, p. 109-118.

Thank you for this interesting nuance.

Reviewer #3 (Remarks to the Author):

I continue to disagree with the author's interpretation that the change in phasing of the $\delta^{13}\text{C}$ signal must be associated uniquely with a northern hemisphere and perhaps cryospheric onset. In their response, they point to evidence for short-lived NH glaciations in the Pliocene that I do not dispute at all- but the claim of a persistent NH glaciation after 6 Ma continues to be unsupported.

However, the paper has evolved to consider many more possibilities than the first two drafts and now reads in a much more nuanced way. The replies to the latest reviews are extensive and on-going. I think the balance has now tipped in favor of accepting the paper and I will be glad to see it join the literature.

We thank reviewer #3 for his challenging attitude throughout this review process, both in terms of methodology, as in terms of interpretations. It sure made us dig deeper and ultimately made the paper stronger.